# Incorporating Interventional Independence Improves Robustness against Interventional Distribution Shift

**Gautam Sreekumar**                                                                 *sreekum1@msu.edu*
*Department of Computer Science and Engineering, Michigan State University*

**Vishnu Naresh Boddeti**                                                            *vishnu@msu.edu*
*Department of Computer Science and Engineering, Michigan State University*

**Reviewed on OpenReview:** *https://openreview.net/forum?id=kXfcEyNIrf*

## Abstract

We consider the problem of learning robust discriminative representations of causally related latent variables given the underlying directed causal graph and a training set comprising passively collected observational data and interventional data obtained through targeted interventions on some of these latent variables. We desire to learn representations that are robust against the resulting interventional distribution shifts. Existing approaches treat interventional data like observational data and ignore the independence relations that arise from these interventions, even when the underlying causal model is known. As a result, their representations lead to large disparities in predictive performance between observational and interventional data. This performance disparity worsens when interventional training samples are scarce. In this paper, (1) we first identify a strong correlation between this performance disparity and the representations' violation of statistical independence induced during interventions. (2) For linear models, we derive sufficient conditions on the proportion of interventional training data, for which enforcing statistical independence between representations of the intervened node and its non-descendants during interventions lowers the test-time error on interventional data. Combining these insights, (3) we propose RepLIn, a training algorithm that explicitly enforces this statistical independence between representations during interventions. We demonstrate the utility of RepLIn on a synthetic dataset, and on real image and text datasets on facial attribute classification and toxicity detection, respectively, with semi-synthetic causal structures. Our experiments show that RepLIn is scalable with the number of nodes in the causal graph and is suitable to improve robustness against interventional distribution shifts of both continuous and discrete latent variables compared to the ERM baselines.

## 1 Introduction

We consider the problem of learning robust discriminative representations corresponding to latent variables for downstream prediction tasks. These latent variables usually correspond to semantic concepts such as the color of an object, the level of glucose in the blood, and a person's age. The relationship between these latent variables can be modeled using directed acyclic graphs (DAGs) called *causal graphs*. Causal modeling allows manually altering the causal graph and observing its effects on the data. E.g., intervene on the amount of insulin (parent variable) in the blood by consuming an insulin inhibitor and then measure the blood glucose level (child variable). This procedure is known as a *causal intervention*, and the data collected through this procedure is called *interventional data*. In contrast, data passively collected without intervention is known as *observational data*. Of the several types of interventions possible on a causal graph, we are interested in

*hard interventions* where we manually set the value of one or more variables. Intervening on a node renders it statistically independent of its *parent nodes* in the causal graph[1]. See (Peters et al., 2017, Chapter 6).

Consider a causal graph with latent variables $A$ and $B$. During observation, $A$ causes $B$ ($A \rightarrow B$). An attribute-specific representation of $A$, denoted by $\boldsymbol{F}_A$, learned from only observational data, may contain information about its child node $B$ due to the association between $A$ and $B$, although this association may be broken during interventions on $B$. **Example**: Consider a computer-aided diagnosis system that inputs a chest X-ray image and outputs representations corresponding to $A =$ "air sac inflammation" to predict pneumonia and $B =$ "fluid accumulation around lungs" to check for pleural effusion. These representations will be used by their corresponding diagnosis-specific predictors. This design makes the system modular and interpretable. Here, pneumonia is one of the many unrelated causes of excess fluid accumulation. *E.g.*, excess fluid accumulation can happen as a side effect of some medication (intervention) unrelated to pneumonia. The representation, $\boldsymbol{F}_A$, used to predict pneumonia may incorporate information about fluid accumulation ($B$) to aid pneumonia diagnosis, although excess fluid accumulation does not guarantee pneumonia. To avoid misdiagnosis, these representations must include only the information necessary for their diagnostic purpose, and must be robust against distribution shifts due to any interventions on causally downstream variables.

To learn representations that are robust against interventional distribution shifts, interventional data samples are included in the training set. For example, in (Sauer & Geiger, 2021; Gao et al., 2023), interventional data was generated to train image classification models invariant to texture and background. When interventional training data is available, existing discriminative learning approaches treat interventional data merely as data sourced from a different domain or *environment*, ignoring the explicit statistical independence relations arising from interventions[2] (Arjovsky et al., 2019; Sagawa et al., 2020; Heinze-Deml & Meinshausen, 2021). As we demonstrate, ignoring these independence relations may lead to representations that are still susceptible to interventional distribution shifts during inference. Moreover, performing interventions in the real world is often challenging and expensive. This limits the amount of interventional data available for training and thus demands a causally motivated learning strategy that leverages the limited interventional training data.

In this work, we first consider a simple case study where we observe that models that do not learn independent representations during interventions show a performance drop on interventional data. We then derive sufficient conditions on the proportion of interventional data during training, under which enforcing linear independence between interventional features of linear models during training can reduce test-time error on interventional data. Motivated by these theoretical insights, we propose "Representation Learning from Interventional Data" (**RepLIn**), an algorithm to learn representations with improved robustness against interventional distribution shifts. We confirm the utility of RepLIn on a variety of synthetic (Sec. 5.1) and real datasets (Secs. 5.2 and 5.3) on various modalities with semi-synthetic causal structures, and demonstrate its scalability to the number of nodes (Sec. 6.2).

**Our Contributions**:

- **Observation**: We demonstrate a positive correlation between the accuracy drop during interventional distribution shift and the dependence between representations corresponding to the label node and its children. We refer to this as "interventional feature dependence" (Sec. 3.3).

- **Theory**: We theoretically explain why linear ERM models are susceptible to interventional distribution shifts in the regime of linear causal models. In the same setting, we theoretically and empirically show that enforcing linear independence between interventional features improves robustness when sufficient interventional data is available during training and establish the sufficient condition (Sec. 3.4).

- **Approach**: We propose a novel training algorithm that combines these insights and demonstrates that this model minimizes the drop in accuracy under interventional distribution shifts by explicitly enforcing independence between interventional features (Sec. 4).

---

[1]For ease, we refer to "statistical independence" as "independence," and "hard interventions" as "interventions." We will also use "features" and "representations" interchangeably to denote the vector representations of the data learned by a model.
[2]The distribution shift due to differing environments is more general than interventional distribution shift. See App. D.

## 2 Related Works

**Identifiable Causal Representation Learning (ICRL)** seeks to learn representations of the underlying causal model under certain assumptions (Locatello et al., 2019; Schölkopf et al., 2021; Hyvärinen et al., 2024), and are important to interpretable representation learning. Several ICRL works also use interventional data (Lippe et al., 2022b; 2023; Ahuja et al., 2023; Squires et al., 2023; von Kügelgen et al., 2023; Zhang et al., 2023; Jiang & Aragam, 2023; Buchholz et al., 2023; Varıcı et al., 2024b; Bing et al., 2024; Lachapelle et al., 2024), where known interventional targets are identifiably learned up to permutation ambiguity. In contrast, we are interested in learning discriminative representations when some underlying causal relations are known. Instead of learning the entire causal model, we seek to exploit the known independence relations from interventions to learn discriminative representations that are robust against these interventions, and hence do not require fully identifiable models. A detailed review of ICRL is in App. C.

**Interventional data** is key in causal discovery (Eberhardt et al., 2005; Yu et al., 2019; Ke et al., 2019; Lippe et al., 2022a; Wang et al., 2022b) as one can only retrieve causal relations up to a Markov equivalent graph without interventions or assumptions on the causal model. For example, known interventional targets have been used for unsupervised causal discovery of linear causal models (Subramanian et al., 2022), interventional and observational data have been leveraged for training a supervised model for causal discovery (Ke et al., 2022), and interventions with unknown targets were used for differentiable causal discovery (Brouillard et al., 2020). Interventional data also find applications in reinforcement learning (Gasse et al., 2021; Ding et al., 2022a) and recommendation systems (Zhang et al., 2021; Krauth et al., 2025; Luo et al., 2024). While this body of work focuses on discovering causal relations in the data, our work considers how to leverage known causal relations to learn data representations that are robust to distribution shifts induced by interventions.

**Domain Generalization (DG)**: In DG, the learning objective is a predictor for an attribute of interest that is robust/invariant to changes in the domain/environment (Mahajan et al., 2021; Wang et al., 2022a; Ding et al., 2022b). Here, there is no interest in learning representations for the domain, and multiple factors could be jointly treated as a single domain. Moreover, there is also no requirement that the learned predictor for the attribute of interest is free of domain information (Rosenfeld et al., 2022). Therefore, the learned representations obtained from domain generalization may not be trustworthy for modular applications such as the medical diagnosis system described in Sec. 1. A more detailed discussion is provided in App. D.

## 3 The Learning from Interventional Data Problem

**Notations**: Random variables and random vectors are denoted by regular (e.g., $A$) and bold (e.g., $\boldsymbol{a}$) serif characters, respectively. The distribution of a random variable $A$ is denoted by $P_A$.

**Setup**: We now formally define the problem of interest in this paper, namely *learning attribute-specific discriminative representations that are robust against known interventional distribution shifts*[3], in general terms, and examine a specific case study in Sec. 3.1. The learning problem is characterized by a DAG $\mathcal{G}$ that causally relates the attributes of interest $A_1, \ldots, A_m$, and $B$. Let $\mathcal{P}a(B) = \{A_1, \ldots, A_m\}$ denote the parents of the attribute $B$. These attributes along with other unobserved exogenous variables $\boldsymbol{U}$, generate the observable data $\boldsymbol{X}$ as $\boldsymbol{X} =$

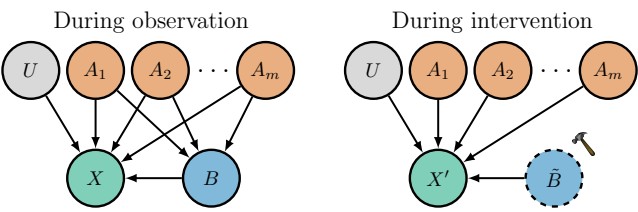

Figure 1: **Causal graph modification due to intervention:** During observation, $B$ is the effect of its parent variables $\mathcal{P}a(B) = \{A_1, \ldots, A_m\}$. When we intervene on $B$, it becomes statistically independent of its parents.

$g_{\boldsymbol{X}}(B, A_1, \ldots, A_m, \boldsymbol{U})$. During interventions, the variable $B$ is set to values drawn from a known distribution independent of $\mathcal{P}a(B)$. Therefore, the post-intervention variable $B$ (denoted by $\tilde{B}$) is statistically independent of its parents, i.e., $\tilde{B} \perp\!\!\!\perp \mathcal{P}a(B)$, as shown in Fig. 1. Although $g_{\boldsymbol{X}}$ is not affected by this intervention, the distribution of $\boldsymbol{X}$ (now denoted by $\boldsymbol{X}'$) will change since it is a function of $B$.

---

[3]We use "discriminative" to explicitly state that the purpose of these representations is robust prediction and not data generation. Information loss with improved robustness is therefore acceptable.

**Assumptions**: To learn representations that are robust against distribution shift due to intervention on $B$, our setting only provides us information about $B$ and its parents in the causal graph, and not of any causal relations between $A_1, \ldots, A_m$. We also do not place restrictions on the functional form of causal relations between $A_1, \ldots, A_m, B$, and $\boldsymbol{X}$, or on their marginal distributions. Our training set comprises both observational and interventional samples, i.e., $\mathcal{D}^{\text{train}} = \mathcal{D}^{\text{obs}} \cup \mathcal{D}^{\text{int}}$, where $\mathcal{D}^{\text{obs}} \sim P(\boldsymbol{X}, B, A_1, \ldots, A_m)$ and $\mathcal{D}^{\text{int}} \sim P(\boldsymbol{X}', \tilde{B}, A_1, \ldots, A_m)$. However, the number of interventional training samples is much less compared to the number of observational training samples, i.e., $|\mathcal{D}^{\text{int}}| \ll |\mathcal{D}^{\text{obs}}|$. Given $\mathcal{D}^{\text{train}}$ and $\mathcal{G}$, the goal is to learn attribute-specific discriminative representations $\boldsymbol{F}_B = h_B(\boldsymbol{X})$ and $\boldsymbol{F}_{A_i} = h_{A_i}(\boldsymbol{X})$, where $\boldsymbol{F}_{A_i}$ are robust against distribution shifts due to intervention on $B$.

### 3.1 Does Accuracy Drop during Interventions Correlate with Interventional Feature Dependence?

In this section, we will design a synthetic dataset for a case study to establish a correlation between the accuracy drop on interventional data and the statistical dependence between the attribute representations under intervention.

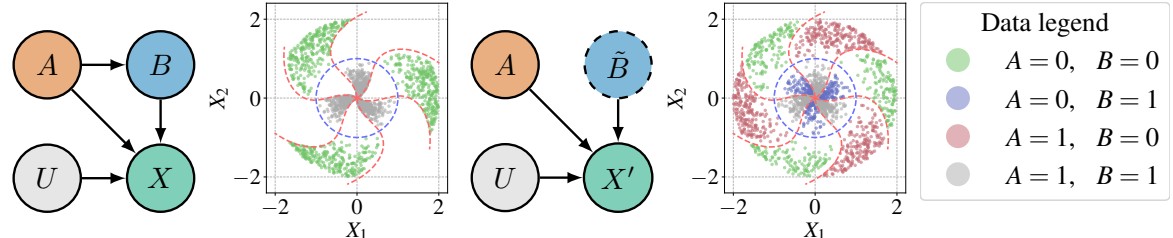

(a) Observational graph and data      (b) Interventional graph and data

Figure 2: **An illustration of Windmill Dataset:** $A$ and $B$ are binary random variables that are causally linked to each other and $\boldsymbol{X}$, as shown in (a). By intervening on $B$ as shown in (b), we make $A \perp\!\!\!\perp \tilde{B}$. $\boldsymbol{X} = g_{\boldsymbol{X}}(A, B, \boldsymbol{U})$ where $\boldsymbol{U}$ denotes unobserved noise variables. The true decision boundaries for predicting $A$ and $B$ from $\boldsymbol{X}$ are shown in red and blue dashed lines, respectively. See App. I for a detailed description.

**Windmill Dataset**: In the causal graph shown in Fig. 2a, $A$ and $B$ are binary random variables that generate the observed data $\boldsymbol{X} \in \mathbb{R}^2$. $\boldsymbol{X}$ is also affected by an unobserved noise variable $\boldsymbol{U}$. Functionally, $\boldsymbol{X} = g_{\boldsymbol{X}}(A, B, \boldsymbol{U})$. $A$ itself could be a function of unobserved random factors that are of no predictive interest to us. Therefore, we model $A \sim \text{Bernoulli}(0.6)$. The distribution of $B$ is only affected by $A$, as denoted by the arrow between them. Analytically, $B := A$, where $:=$ is the causal assignment operator, following (Peters et al., 2017). Visually, the observed data looks like a windmill. The value of $A$ determines the windmill's blade, and $B$ determines the radial distance. The precise angle and radial distance of the data samples are determined by noise samples independent of $A$ and $B$. The windmill blades are also sheared as a sinusoidal function of the radial distance. In Fig. 2b, we intervene on $B$, modeled as $\tilde{B} \sim \text{Bernoulli}(0.5)$. This induces a change in the distribution of $B$, and subsequently in that of $\boldsymbol{X}$. Since the intervention is independent of $A$, $\tilde{B}$ is also independent of $A$, denoted by removing the arrow between $A$ and $\tilde{B}$.

Note that $g_{\boldsymbol{X}}$ is a one-to-one mapping by construction and unaffected by this intervention. Therefore, a single model can predict $A$ and $B$ from $\boldsymbol{X}$ accurately during both observation and intervention. However, the true decision boundary for $A$ is more *complex* than that of $B$[4]. Therefore, models may use information from $B$ to predict $A$ due to their association during observation, similar to the concept of simplicity bias (Shah et al., 2020). The exact mathematical formulation of the data-generating process is provided in App. I.

**The learning task** is to accurately predict $A$ and $B$ from $\boldsymbol{X}$ at test time. We have $N$ samples for training, where $\beta N$ are interventional and $(1-\beta)N$ are observational with $0 < \beta \ll 1$. For this demonstration, we set $N = 40,000$, $\beta = 0.01$ to get 39,600 observational and 400 interventional samples. We train a feed-forward network with two hidden layers to learn representations $\boldsymbol{F}_A$ and $\boldsymbol{F}_B$ corresponding to $A$ and $B$, respectively. We normalize them by dividing them by their corresponding $L_2$ norm. Separate linear classifiers predict $A$

---

[4]We informally define "complexity" as the minimum polynomial degree required to approximate the decision boundary.

and $B$ from $\boldsymbol{F}_A$ and $\boldsymbol{F}_B$ respectively. As a result of WINDMILL's construction, $\boldsymbol{F}_A$ may contain information about $B$ even during interventions when $A \perp\!\!\!\perp B$. Following the standard ERM framework, the cross-entropy errors in predicting $A$ and $B$ from $\boldsymbol{F}_A$ and $\boldsymbol{F}_B$, respectively, provide the training signal. The statistical loss function can be written as $\mathcal{L}_{\text{total}}(f) = \mathbb{E}_{P_{\text{train}}}\left[\mathcal{L}_{\text{pred}}(f, \boldsymbol{X})\right]$. The training distribution is a mixture of observational and interventional distributions with $(1-\beta)$ and $\beta$ acting as the corresponding mixture weights. Thus, $\mathcal{L}_{\text{total}}(f) = (1-\beta)\mathbb{E}_{P_{\text{obs}}}\left[\mathcal{L}_{\text{pred}}(f, \boldsymbol{X}^{\text{obs}})\right] + \beta\mathbb{E}_{P_{\text{int}}}\left[\mathcal{L}_{\text{pred}}(f, \boldsymbol{X}^{\text{int}})\right]$.

| ERM version | Accuracy in predicting $A$ | | | Accuracy in predicting $B$ | | | NHSIC |
|---|---|---|---|---|---|---|---|
| | Observation | Intervention | Relative drop | Observation | Intervention | Relative drop | |
| Vanilla | $99.98 \pm 0.01$ | $60.15 \pm 3.12$ | $0.40 \pm 0.03$ | $100.00 \pm 0.00$ | $99.99 \pm 0.01$ | $0$ | $0.72 \pm 0.06$ |
| w/ Resampling | $94.53 \pm 1.14$ | $70.20 \pm 3.73$ | $0.26 \pm 0.03$ | $100.00 \pm 0.00$ | $99.99 \pm 0.01$ | $0$ | $0.64 \pm 0.08$ |

Table 1: The relative drop in accuracy in predicting $A$ correlates well with a gap in the measure of dependence between the learned representations on interventional data.

**Observations**: Tab. 1 shows the accuracy of ERM in predicting $A$ and $B$ on observational and interventional data during validation. *We expect no drop in accuracy from observation to intervention* if the learned representations are robust against interventional distribution shift. However, we observe that ERM performs only slightly better than random chance in predicting $A$ on interventional data. As a remedy, we modify the vanilla ERM method to sample observational and interventional data in separate batches, and thus prevent the gradients from interventional samples being obfuscated by those from observational samples, which are likely to be more in number in a given batch. This is equivalent to sampling interventional data $\left(\frac{1-\beta}{\beta}\right)$ times as observational data. We refer to this version as "ERM-Resampled." The equivalent training loss for a model $f$ in ERM-Resampled is $\mathcal{L}_{\text{total}}(f) = \mathbb{E}_{P_{\text{obs}}}\left[\mathcal{L}_{\text{pred}}(f, \boldsymbol{X}^{\text{obs}})\right] + \mathbb{E}_{P_{\text{int}}}\left[\mathcal{L}_{\text{pred}}(f, \boldsymbol{X}^{\text{int}})\right]$. Note that $\beta$ does not appear in $\mathcal{L}_{\text{total}}(f)$ due to resampling. Although ERM-Resampled performs better than vanilla ERM, ERM-Resampled still exhibits a large drop in predictive accuracy between observational and interventional data during inference. We also observe a drop in ERM-Resampled's observational accuracy of predicting $A$ as it improves its interventional accuracy. As we will show in Sec. 3.4, this drop in observational accuracy is due to the removal of spurious information previously exploited to boost its observational accuracy.

## 3.2 Measuring Statistical Dependence Between Interventional Features

A key consequence of hard interventions in causal graphs is that the variable being intervened upon becomes independent of all its non-descendants. Since the predictive accuracy on the parent node is affected by intervention, we hypothesize that the parent node's representation remains dependent on that of the child node during intervention, even when their underlying latent variables in the causal graph become independent. To verify our hypothesis, we measure the dependence between the representations. We choose to measure the dependence between the representations instead of between the representations and the latent attributes to align with our goal of learning robust attribute-specific representations.

**To measure dependence** between a pair of high-dimensional continuous random variables $\boldsymbol{X}$ and $\boldsymbol{Y}$, we use HSIC (Gretton et al., 2005). Empirical HSIC between $N$ i.i.d. samples $\mathcal{X} = \{\boldsymbol{x}_i\}_{i=1}^N$ and $\mathcal{Y} = \{\boldsymbol{y}_i\}_{i=1}^N$ from $\boldsymbol{X}$ and $\boldsymbol{Y}$, respectively, is $\text{HSIC}(\mathcal{X}, \mathcal{Y}) = \frac{1}{(N-1)^2}\text{Trace}\left[\boldsymbol{K}_X \boldsymbol{H} \boldsymbol{K}_Y \boldsymbol{H}\right]$, where $\boldsymbol{H}$ is the $N \times N$ centering matrix, and $\boldsymbol{K}_X, \boldsymbol{K}_Y \in \mathbb{R}^{N \times N}$ are Gram matrices whose $(i, j)^{\text{th}}$ entries are $k_X(\boldsymbol{x}_i, \boldsymbol{x}_j)$ and $k_Y(\boldsymbol{y}_i, \boldsymbol{y}_j)$, respectively. Here, $k_X$ and $k_Y$ are Mercer kernels (Schölkopf & Smola, 2002, Chapter 2). Since HSIC is unbounded, we normalize it as $\text{NHSIC}(\mathcal{X}, \mathcal{Y}) = \frac{\text{HSIC}(\mathcal{X}, \mathcal{Y})}{\sqrt{\text{HSIC}(\mathcal{X}, \mathcal{X})\,\text{HSIC}(\mathcal{Y}, \mathcal{Y})}}$, following (Cortes et al., 2012; Cristianini et al., 2001). To improve computational efficiency, we use random Fourier features (Rahimi & Recht, 2007).

**Observations:** Tab. 1 compares NHSIC values between the features $\boldsymbol{F}_A$ and $\boldsymbol{F}_B$ learned by ERM and ERM-Resampled on interventional data from WINDMILL dataset. We observe that features learned by ERM had more statistical dependence during interventions than those by vanilla ERM, indicating a larger violation of the underlying statistical independence relations in the causal graph during interventions. Interestingly, the relative drop in accuracy also increases with the statistical dependence between interventional features.

### 3.3 Strength of Correlation between Drop in Accuracy and Interventional Features Dependence

How strong is the observed correlation between the dependence of features and the drop in accuracy? For a given combination of predictive task and dataset, does it hold for a variety of hyperparameter settings? To answer these questions, we train several models under the ERM-Resampled setting described in Sec. 3.1. The learned models are feed-forward networks, each with one to six hidden layers and with 20 to 200 hidden units. We also use early-stopping in our training, as it has been noted as an effective regularizer (Sagawa et al., 2020). Early-stopping is executed in our experiments by choosing an arbitrary number of training epochs for each run. We measure the robustness of a model to interventional distribution shift using the relative drop in accuracy between observational and interventional data: $\text{Rel.}\Delta = \frac{\text{Obs acc.} - \text{Int acc.}}{\text{Obs acc.}}$. Similar experiments were reported in (Sreekumar & Boddeti, 2023), although their primary research question concerned the effect of data and model complexities on spurious correlations. In the following experiment, we expand their setting to deeper models and more variety in hyperparameters while foregoing the variation in data complexity.

In Fig. 3, we plot the relative drop in accuracy against the interventional feature dependence. In addition to NHSIC, we also use kernel canonical correlation (KCC) (Bach & Jordan, 2002) to measure the dependence. The strength of the correlation between the relative drop in accuracy and interventional feature dependence is quantified using Spearman's rank correlation coefficient ($\rho$) (Spearman, 1904) and Kendall's rank correlation coefficient ($\tau$) (Kendall, 1938). In Fig. 3a, $\rho = 0.81$ and $\tau = 0.61$ when the dependence is measured using NHSIC, indicating that the correlation noted in Sec. 3.2 can be observed for a wide range of hyperparameters. When KCC is used for measuring interventional feature dependence, $\rho = 0.75$ and $\tau = 0.56$ as shown in Fig. 3b. We observe that all models with a high relative drop in accuracy also have a large in-

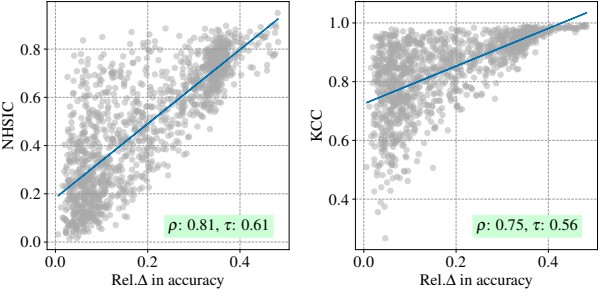

(a) Rel.$\Delta$ against NHSIC  (b) Rel.$\Delta$ against KCC

Figure 3: Across various model capacities and hyperparameter settings, a relative drop in accuracy is always accompanied by interventional feature dependence. However, the corollary does not hold. Feature dependence is measured using NHSIC and KCC.

terventional feature dependence (see top-right regions in the plots). However, the corollary is not true – a large interventional feature dependence does not mean a relative drop in accuracy. Therefore, we conclude for this case study that *a relative drop in accuracy is always accompanied by interventional feature dependence.*

**Note on Choice of Measure of Dependence**: The correlation strength in Fig. 3 was affected by our choice of measure of dependence. A popular measure of information between two random variables is Shannon mutual information (MI). However, computing MI requires density estimation as the first step, which is challenging for high-dimensional data (Paninski, 2003; McAllester & Stratos, 2020). For the same reason, MI is also not suitable for training representations. We note that a variational upper bound for MI can be obtained (and minimized to enforce independence between representations) if the conditional density of one random variable w.r.t. the other is known (Barber & Agakov, 2004; Alemi et al., 2018; Poole et al., 2019), although computing this bound still requires a tractable density.

In contrast, **kernel-based measures of dependence**, such as HSIC and KCC, are computationally efficient and well-suited for optimization. Both NHSIC and KCC satisfy the postulates for an appropriate measure of dependence in (Rényi, 1959) and measure dependence from the spectrum of the cross-covariance operator between reproducing kernel Hilbert spaces (RKHSs). However, HSIC measures the Hilbert-Schmidt norm of the cross-covariance operator while KCC measures its spectral norm (largest singular value). As a result, KCC is more suited for independence tests where the presence of dependence is more important than its overall strength. Informally, KCC is a "harsher" measure of dependence compared to NHSIC. HSIC between two random variables is equivalent to maximum mean discrepancy (MMD) (Gretton et al., 2012) between the joint distribution and the product of marginals of these variables (Schrab, 2025). MMD was originally proposed to check whether two samples came from the same distribution or not. They have similar computational costs. HSIC is also a lower bound on MI (Sriperumbudur et al., 2012; Xu et al., 2024). For the remainder of this work, we will use NHSIC for training and analysis, and reserve KCC for evaluation.

### 3.4 Will Minimizing Dependence between Interventional Features Improve Robustness?

Our case study in Sec. 3.3 showed that a large relative drop in accuracy is always accompanied by strong interventional feature dependence. Based on this observation, we ask the following question: Will minimizing interventional feature dependence improve the robustness to interventional distribution shifts? We answer this question theoretically using a linear causal model. The detailed proof of each step is provided in App. B.

**Causal Model**: We use the causal model shown in Fig. 2a with $A$ and $B$ being continuous random variables. $A$ and $B$ are causally related during observation as $B := w_{AB}A$. Although our analysis is valid if an external noise was added to $B$, we will skip such a noise term in the proof for conciseness. The observed data signal $\boldsymbol{X}$ is generated from $A$ and $B$ as $\boldsymbol{X} := \begin{bmatrix} X_A \\ X_B \end{bmatrix} + \boldsymbol{U}$, where $X_A := w_A A$ and $X_B := w_B B$. $\boldsymbol{U} := \begin{bmatrix} U_A \\ U_B \end{bmatrix}$ is exogenous noise. $U_A$ and $U_B$ are independent of $A$ and $B$ respectively. We intervene on $B$ as shown in Fig. 2b, severing the causal relation between $A$ and $B$. The intervened variable is denoted as $B'$ and $B' \perp\!\!\!\perp A$.

**Learning Model**: Similar to Sec. 3.3, the task is to predict the latent variables $A$ and $B$ from observed data signal $\boldsymbol{X}$. The training data is sampled from a training distribution $P_{\text{train}}$, which we model as a mixture of observation distribution, $P_{\text{obs}}$, and interventional distribution, $P_{\text{int}}$, with mixture weights $(1 - \beta)$ and $\beta$, respectively: $P_{\text{train}} = (1 - \beta)P_{\text{obs}} + \beta P_{\text{int}}$. We use linear models to learn attribute-specific representations $\boldsymbol{F}_A$ and $\boldsymbol{F}_B$, from which predictions $\hat{A}$ and $\hat{B}$, respectively, are made using corresponding classifiers. The linear models are parameterized by $\boldsymbol{\Theta}^{(A)}$ and $\boldsymbol{\Theta}^{(B)}$, and the classifiers are parameterized by $\boldsymbol{c}^{(A)}$ and $\boldsymbol{c}^{(B)}$.

**Statistical Risk**: The parameter matrix of the linear feature extractor for $A$ can be written in terms of its constituent parameter vectors as $\boldsymbol{\Theta}^{(A)} = \begin{bmatrix} \boldsymbol{\theta}_A^{(A)\top} \\ \boldsymbol{\theta}_B^{(A)\top} \end{bmatrix}$. Assuming zero mean for all latent variables[5], the statistical squared error of an arbitrary model in predicting $A$ from an interventional test sample $\boldsymbol{X}$ is,

$$E_A = \underbrace{\left(1 - w_A \boldsymbol{c}^{(A)\top} \boldsymbol{\theta}_A^{(A)}\right)^2 \rho_A^2 + \left(\boldsymbol{c}^{(A)\top} \boldsymbol{\theta}_A^{(A)}\right)^2 \rho_{\boldsymbol{U}_A}^2}_{E_A^{(1)}} + \underbrace{\left(w_B \boldsymbol{c}^{(A)\top} \boldsymbol{\theta}_B^{(A)}\right)^2 \rho_{B'}^2 + \left(\boldsymbol{c}^{(A)\top} \boldsymbol{\theta}_B^{(A)}\right)^2 \rho_{\boldsymbol{U}_B}^2}_{E_A^{(2)}} \quad (1)$$

where $\rho_A^2 = \mathbb{E}_{P_{\text{int}}}\left[A^2\right]$, $\rho_{B'}^2 = \mathbb{E}_{P_{\text{int}}}\left[B'^2\right]$, $\rho_{\boldsymbol{U}_A}^2 = \mathbb{E}_{P_{\text{int}}}\left[U_A^2\right]$, and $\rho_{\boldsymbol{U}_B}^2 = \mathbb{E}_{P_{\text{int}}}\left[U_B^2\right]$. The statistical risk can be split into two components: (1) $E_A^{(1)}$ in terms of $A$ and $U_A$, and (2) $E_A^{(2)}$ in terms of $B$ and $U_B$. $E_A^{(2)} \neq 0$ when $\boldsymbol{\theta}_B^{(A)} \neq \boldsymbol{0}$. A non-zero $\boldsymbol{\theta}_B^{(A)}$ indicates that the representation $\boldsymbol{F}_A$ is a function of $X_B$, i.e., it learns a spurious correlation with $B$. Thus, the prediction $\hat{A}$ is susceptible to interventions on $B$. In contrast, a robust model will have $\boldsymbol{\theta}_B^{(A)} = \boldsymbol{0}$, and thus achieves $E_A^{(2)} = 0$. Derivation of Eq. (1) is provided in App. B.1.

**An optimal ERM model** minimizes the expected training risk in predicting the latent attributes. Since we are interested in the accuracy drop in predicting $A$ from interventional data, we consider the optimization of parameters for predicting $A$ by minimizing the expected mean squared error over the training distribution.

$$\boldsymbol{\Theta}^{(A)*}, \boldsymbol{c}^{(A)*} = \underset{\boldsymbol{\Theta}^{(A)}, \boldsymbol{c}^{(A)}}{\operatorname{argmin}} \mathbb{E}_{P_{\text{train}}}\left[\left(A - \boldsymbol{c}^{(A)\top} \boldsymbol{\Theta}^{(A)\top} \boldsymbol{X}\right)^2\right] \quad (2)$$

Since $\boldsymbol{\Theta}^{(A)}$ and $\boldsymbol{c}^{(A)}$ can be optimized only up to a scaling factor, we can equivalently optimize $\boldsymbol{\psi}_A = \boldsymbol{c}^{(A)\top} \boldsymbol{\Theta}^{(A)\top} = \begin{bmatrix} \psi_1 \\ \psi_2 \end{bmatrix}$, where $\psi_1 = \boldsymbol{c}^{(A)\top} \boldsymbol{\theta}_A^{(A)}$ and $\psi_2 = \boldsymbol{c}^{(A)\top} \boldsymbol{\theta}_B^{(A)}$. Thus, Eq. (2) becomes

$$\boldsymbol{\psi}_A^* = \underset{\boldsymbol{\psi}_A}{\operatorname{argmin}} \mathbb{E}_{P_{\text{train}}}\left[(A - \boldsymbol{\psi}_A \boldsymbol{X})^2\right]. \quad (3)$$

To check whether the optimal ERM model is robust, we can verify if $\psi_2^* = 0$, since a robust model will have $\boldsymbol{\theta}_B^{(A)} = \boldsymbol{0}$. Solving Eq. (3) by setting the gradients of the optimization objective to zero, we get:

$$\psi_2^* = \frac{-(1 - \beta)w_B w_{AB} \sigma_A^2 \sigma_{U_A}^2}{T} \neq 0, \quad (4)$$

---

[5]The zero mean assumption is to make the calculations easier. This will not affect our conclusion from the proof, as we can always learn the mean of the data separately.

where $T$ is a non-zero scalar. If there was an added noise term in the causal relation $A \rightarrow B$, Eq. (4) would have taken a different form, but would have still been non-zero. Eq. (4) implies that $\boldsymbol{\theta}_B^{(A)} \neq \mathbf{0}$, and consequently implies $E_A^{(2)} \neq 0$ in optimal ERM models. Therefore, optimal ERM models are not robust against interventional distribution shift. The detailed derivation is provided in App. B.2.

Note that a robust model cannot be a minimizer of training loss in Eq. (3), as the minimizer requires $\boldsymbol{\theta}_B^{(A)} \neq \mathbf{0}$. This means improving robustness by minimizing spurious information from $B$ in predicting $A$ (through $\boldsymbol{\theta}_B^{(A)} \rightarrow \mathbf{0}$) may lead to higher prediction loss over observational data. This explains the drop in observational accuracy of ERM-Resampled when its interventional accuracy in predicting $A$ improved in Sec. 3.1. This phenomenon is also illustrated in App. H.

**Minimizing linear dependence**: In Sec. 3.3, we showed that interventional feature dependence correlated positively with the drop in accuracy on interventional data. We will now verify if minimizing dependence between interventional features $\boldsymbol{F}_A$ and $\boldsymbol{F}_B'$ can reduce the accuracy drop in a linear setting.

$$\boldsymbol{F}_A = \boldsymbol{\Theta}^{(A)\top}\boldsymbol{X} = X_A\boldsymbol{\theta}_A^{(A)} + X_B\boldsymbol{\theta}_B^{(A)} \qquad \text{(Interventional feature for } A)$$

$$\boldsymbol{F}_B' = \boldsymbol{\Theta}^{(B)\top}\boldsymbol{X} = X_A\boldsymbol{\theta}_A^{(B)} + X_B\boldsymbol{\theta}_B^{(B)} \qquad \text{(Interventional feature for } B')$$

Since the data generation process and the learned models are linear, it is sufficient to minimize the linear interventional dependence between representations instead of the full statistical dependence. Following the definition of HSIC (Gretton et al., 2005), the linear dependence in interventional features can be defined as[6],

$$\text{Dep}\left(\boldsymbol{F}_A, \boldsymbol{F}_B'\right) = \left\|\mathbb{E}_{P_{\text{int}}}\left[\boldsymbol{F}_A\boldsymbol{F}_B'^{\top}\right]\right\|_F^2. \tag{5}$$

Leveraging the independence relations during interventions, we can expand Eq. (5) as,

$$\left\|\mathbb{E}_{P_{\text{int}}}\left[\boldsymbol{F}_A\boldsymbol{F}_B'^{\top}\right]\right\|_F^2 = \left\|(w_A^2\rho_A^2 + \rho_{\boldsymbol{U}_A}^2)\boldsymbol{\theta}_A^{(A)}\boldsymbol{\theta}_A^{(B)\top} + (w_B^2\rho_{B'}^2 + \rho_{\boldsymbol{U}_B}^2)\boldsymbol{\theta}_B^{(A)}\boldsymbol{\theta}_B^{(B)\top}\right\|_F^2 \tag{6}$$

The dependence loss is thus the Frobenius norm of a sum of rank-one matrices. Three classes of solutions minimize Eq. (6): (1) $\boldsymbol{\theta}_A^{(A)} = \boldsymbol{\theta}_B^{(A)} = \boldsymbol{\theta}_A^{(B)} = \boldsymbol{\theta}_B^{(B)} = \mathbf{0}$, (2) $\boldsymbol{\theta}_A^{(B)} = \pm\gamma\boldsymbol{\theta}_A^{(A)}$ and $\gamma\boldsymbol{\theta}_B^{(B)} = \mp\boldsymbol{\theta}_B^{(A)}$ for some scalar $\gamma \neq 0$, and (3) $\boldsymbol{\theta}_A^{(A)} = \mathbf{0}$ or $\boldsymbol{\theta}_A^{(B)} = \mathbf{0}$, and $\boldsymbol{\theta}_B^{(A)} = \mathbf{0}$ or $\boldsymbol{\theta}_B^{(B)} = \mathbf{0}$. However, all solutions produce trivial features and increase the classification error, except two non-degenerate solutions: (S1) $\boldsymbol{\theta}_A^{(A)} = \mathbf{0}, \boldsymbol{\theta}_B^{(B)} = \mathbf{0}$, and (S2) $\boldsymbol{\theta}_B^{(A)} = \mathbf{0}, \boldsymbol{\theta}_A^{(B)} = \mathbf{0}$, where (S2) corresponds to a robust model. Since both (S1) and (S2) minimize Eq. (6), the solution with lower prediction error over $A$ and $B$ will prevail during training.

**Proposition 1.** *The total training error for (S1) is strictly greater than that of (S2) when the following conditions are satisfied: (1)* $\beta \geq 1 - \frac{1}{|w_{AB}|}$*, (2)* $\beta \geq \min\left(\frac{\rho_A^2}{2\rho_{B'}^2 + \rho_A^2}, \frac{\rho_{\boldsymbol{U}_A}^2}{w_A^2 w_{AB}^2 \rho_A^2}\right)$*.*

Proposition 1 conveys that, given a certain proportion of interventional data in the training set, explicitly enforcing independence between learned representations can provably improve robustness against interventional distributional shifts. Note that Proposition 1 describes sufficient conditions for (S1) to have a larger training error than (S2), and *does not* imply the contrary for smaller values of $\beta$. In practice, $\beta$ could be much smaller. For instance, in Tab. 2, explicitly enforcing independence using our proposed approach improves robustness even for $\beta = 1\%$. Refer to App. B.3 for derivation and empirical verification of Proposition 1.

**To experimentally verify the theoretical results**, we construct the causal model with $w_A = w_B = w_{AB} = 1$. The random variables $A$, $B$, $U_A$, and $U_B$ are sampled from independent normal distributions with zero mean and unit variance. We generate $N = 50000$ data points for training with $\beta = 0.5$. The classifiers use 2-dimensional features learned by linear feature extractors to predict $A$ and $B$. The experiment is repeated with 50 seeds. $E_A^{(1)}$ and $E_A^{(2)}$, the components of the statistical risk in Eq. (1), are plotted in Figs. 4a and 4b, respectively. As a reminder, a robust model will achieve $E_A^{(2)} = 0$. As expected, both models have similar $E_A^{(1)}$. However, linear independence models achieve lower $E_A^{(2)}$ ($E_A^{(2)} \approx 0$, similar to a robust model) than ERM models, and thus obtain a lower total error $E_A$ (Fig. 4c).

---

[6]For a complete definition of the dependence, refer to App. B.3.

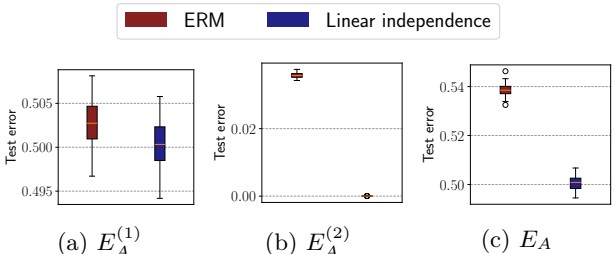

(a) $E_A^{(1)}$  (b) $E_A^{(2)}$  (c) $E_A$

Figure 4: Robust models achieve $E_A^{(2)} = 0$ in Eq. (1). ERM models have a non-zero $\boldsymbol{\theta}_B^{(A)}$ resulting in $E_A^{(2)} \neq 0$. Minimizing linear independence on interventional features results in orthogonal interventional feature spaces where $\boldsymbol{\theta}_B^{(A)} = \boldsymbol{\theta}_A^{(B)} = \mathbf{0}$. Thus, they result in robust models with $E_A^{(2)} = 0$.

## 4 RepLIn: Enforcing Statistical Independence between Interventional Features

The previous section demonstrated a strong correlation between the accuracy drop during interventions and interventional feature dependence. We also showed theoretically that minimizing linear dependence between interventional features can improve test-time error on interventional data for linear models. Based on this observation, we propose "Representation Learning from Interventional data" (RepLIn), a training method to learn discriminative representations that are robust against known interventional distribution shifts.

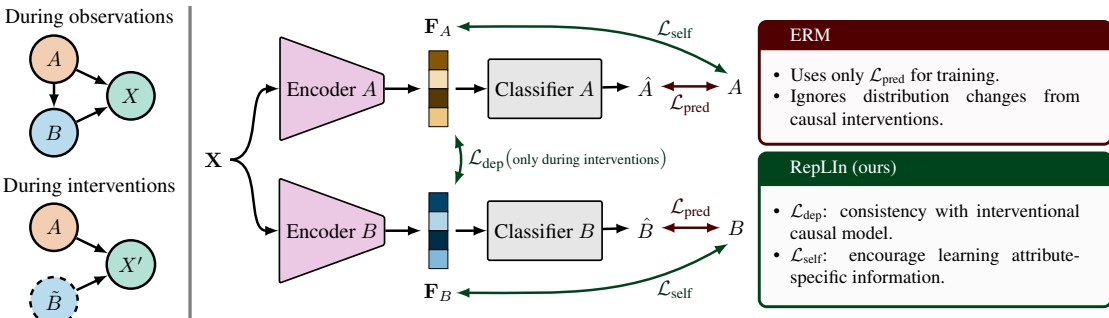

Figure 5: **Schematic illustration of RepLIn** for a two-variable causal graph $(A \rightarrow B)$ where $\boldsymbol{X} = g_{\boldsymbol{X}}(A, B, \boldsymbol{U})$. Encoders for $A$ and $B$ learn corresponding representations $\boldsymbol{F}_A$ and $\boldsymbol{F}_B$, which are then used by their corresponding classifiers to predict $\hat{A}$ and $\hat{B}$, respectively. On interventional samples, we minimize $\mathcal{L}_{\text{dep}}$ between the features to ensure their independence. On all samples, we minimize $\mathcal{L}_{\text{self}}$ to improve attribute-specific information in the representations.

**To enforce independence between interventional features**, we propose to use dependence-guided regularization, denoted as $\mathcal{L}_{\text{dep}}$, in addition to the prediction loss (e.g., cross-entropy for classification tasks) used in ERM. We refer to this regularization as "dependence loss," and is defined for the general case in Sec. 3 as $\mathcal{L}_{\text{dep}} = \sum_{i=1}^n \text{NHSIC}(\boldsymbol{F}_{A_i}^{\text{int}}, \boldsymbol{F}_B^{\text{int}})$. We minimize the dependence loss *only* for the interventional training samples since congruent statistical independence between latent variables occurs only during interventions.

However, $\mathcal{L}_{\text{dep}}$ may encourage learning irrelevant information to lower feature dependence. Therefore, **to encourage learning relevant attribute-specific information**, we use the "self-dependence loss," denoted by $\mathcal{L}_{\text{self}}$, to maximize the dependence between a feature and its corresponding label on both observational and interventional training data. We define $\mathcal{L}_{\text{self}}$ as $\mathcal{L}_{\text{self}} = 1 - \frac{\text{NHSIC}(\boldsymbol{F}_B, B) + \sum_{i=1}^n \text{NHSIC}(\boldsymbol{F}_{A_i}, A_i)}{2(n+1)}$. Using $\mathcal{L}_{\text{self}}$ in addition to $\mathcal{L}_{\text{pred}}$ ensures that the representations contain as much information about the modeled latent variables as possible, rather than just the information required for the train-time prediction task.

**Kernel choice for $\mathcal{L}_{\text{dep}}$ and $\mathcal{L}_{\text{self}}$:** The nature of dependence captured by $\mathcal{L}_{\text{dep}}$ and $\mathcal{L}_{\text{self}}$ depends on the richness of the underlying RKHS induced by the kernel used in NHSIC. To maximize a lower estimate of the dependence, we will use linear kernels, $k_X(\boldsymbol{x}_i, \boldsymbol{x}_j) = \boldsymbol{x}_i^\top \boldsymbol{x}_j$, in $\mathcal{L}_{\text{self}}$. Likewise, to remove all non-linear dependence between the representations, we will use RBF kernels in $\mathcal{L}_{\text{dep}}$. In general, universal kernels, such as RBF and polynomial kernels, capture non-linear dependence between variables up to the resolution decided by the kernel's parameters (Micchelli et al., 2006; Fukumizu et al., 2007). The empirical estimates of

HSIC in our loss functions have a bias of $\mathcal{O}(n^{-1})$ against their population estimates, where $n$ is the number of independent samples used to estimate HSIC (Gretton et al., 2005). The empirical HSIC estimates thus converge to their population estimates when the sample size is large. For computationally efficient estimation of NHSIC, we use random Fourier features (Rahimi & Recht, 2007).

In summary, RepLIn optimizes the following total loss: $\boxed{\mathcal{L}_{\text{total}} = \mathcal{L}_{\text{pred}} + \lambda_{\text{dep}}\mathcal{L}_{\text{dep}} + \lambda_{\text{self}}\mathcal{L}_{\text{self}}}$, where $\lambda_{\text{dep}}$ and $\lambda_{\text{self}}$ are weights that control the contribution of the respective losses. The impact of the choice of these hyperparameters is discussed in App. H. A pictorial overview of the RepLIn pipeline is shown in Fig. 5.

## 5 Experimental Evaluation

In this section, we verify the effectiveness of RepLIn compared to the baselines using the WINDMILL dataset introduced in Sec. 3.1, and evaluate its broader applicability to practical scenarios through the facial attribute prediction task on the CelebA dataset (Liu et al., 2015) and toxicity prediction on the CivilComments dataset (Borkan et al., 2019). Since the true causal models are not known for real datasets, we design plausible causal models for these datasets based on the variables of interest, and sample data points according to these semi-synthetic causal models. Through our experiments, we validate the following hypothesis: Does explicitly minimizing the interventional feature dependence reduce accuracy drop during interventions?

**Training Setup and Baselines**: We consider vanilla **ERM** and **ERM-Resampled** (Chawla et al., 2002; Cateni et al., 2014) as our primary baselines since they are the most commonly used training algorithms. Additionally, ERM-Resampled has been shown to be a strong baseline for group-imbalanced training and domain generalization (Idrissi et al., 2022; Gulrajani & Lopez-Paz, 2021). On WINDMILL dataset, we also consider the following domain generalization algorithms: **IRMv1** (Arjovsky et al., 2019), **Fish** (Shi et al., 2022), **GroupDRO** (Sagawa et al., 2020), **SAGM** (Wang et al., 2023), **DiWA** (Rame et al., 2022), and **TEP** (Qiao & Peng, 2024). The latter two are weight-averaging methods, for which we train 20 independent models per seed. We do not include ICRL baselines, as they learn attribute-identifiable representations up to permutation invariance, which cannot be used with attribute-specific classifiers. We compare two variants of our method against these baselines: RepLIn and RepLIn-Resampled. The latter variant uses the resampling strategy from ERM-Resampled. In each method, attribute-specific representations are extracted from the input data and fed into the corresponding classifiers to get the final prediction. All baselines use the same architecture to learn representations and linear layers to make the final prediction from these representations. The values of $\lambda_{\text{dep}}$ and $\lambda_{\text{self}}$ in RepLIn variants are tuned and kept fixed for all values of $\beta$. A detailed description of the datasets and the training settings is provided in App. A.

**Evaluation Metrics**: Our primary interest is in investigating the accuracy drop during interventions for the variables unaffected by these interventions. Ideally, if the learned features respect causal relations during interventions, we expect no change in the prediction accuracy of parent variables of the intervened variable between observational and interventional distributions. To measure the change, we use the relative drop in accuracy defined in Sec. 3.3: $\text{Rel.}\Delta = \frac{\text{Obs acc.} - \text{Int acc.}}{\text{Obs acc.}}$. Since we optimize NHSIC during training, we use KCC from Sec. 3.3 to evaluate the interventional feature dependence during testing. We repeat each experiment with five different random seeds and report the mean and standard deviation.

### 5.1 Windmill dataset

We first evaluate our method on WINDMILL dataset that helped us identify the relation between the performance gap between observational and interventional data in predicting $A$ in Sec. 3.1. As a reminder, the causal graph consists of two binary random variables, $A$ and $B$, where $A \rightarrow B$ during observations. We intervene by setting $B \sim \text{Bernoulli}(0.5)$, breaking the dependence between $A$ and $B$. The proportion of interventional samples in the training data varies from $\beta = 0.01$ to $\beta = 0.5$.

Tab. 2 compares the interventional accuracy in predicting $A$ for various amounts of interventional training data. We make the following observations: **(1)** RepLIn outperforms every baseline in interventional accuracy for all values of $\beta$, clearly demonstrating the advantage of exploiting the underlying causal relations when learning from interventional data, in contrast to treating it as a separate domain. **(2)** Comparing ERM and

RepLIn with their resampling variants, we observe the usefulness of resampling through its large gains when $\beta$ is very small (*e.g.*, $\beta \leq 0.05$). We are also interested in the relative drop in accuracy between observational and interventional data (Rel.$\Delta$). From Tab. 2, we observe that GroupDRO has the lowest Rel.$\Delta$ among the considered methods for $\beta \geq 0.05$, and achieves its best results when more interventional data is available during training. However, this improvement comes at the cost of lower interventional accuracy – over 7 percentage points difference compared to RepLIn. Meanwhile, Rel.$\Delta$ of RepLIn is comparable to GroupDRO at larger values of $\beta$ and has the least Rel.$\Delta$ at lower values of $\beta$. DiWA and TEP were provided with the same pool of models trained with minor variations in their hyperparameters. We do not consider the negative Rel.$\Delta$ of TEP since (1) TEP achieves very low interventional accuracy, performing barely above random chance, and (2) due to the high variance of its Rel.$\Delta$. We provide the results on observational data in App. E. As mentioned in Sec. 3.1, interventional robustness may be at odds with observational accuracy, as removing spurious information from representations may hurt performance on observational data. Our analysis in Sec. 6.1 shows that the representations learned by RepLIn are less affected by interventional shifts.

Accuracy on interventional data. The relative drop in accuracy is shown in parentheses.

| Method | $\beta = 0.5$ | | $\beta = 0.3$ | | $\beta = 0.1$ | | $\beta = 0.05$ | | $\beta = 0.01$ | |
|---|---|---|---|---|---|---|---|---|---|---|
| ERM | $76.87_{\pm 1.08}$ | $(0.18_{\pm 0.01})$ | $69.86_{\pm 3.19}$ | $(0.29_{\pm 0.04})$ | $62.78_{\pm 1.77}$ | $(0.37_{\pm 0.02})$ | $59.52_{\pm 1.30}$ | $(0.40_{\pm 0.01})$ | $60.15_{\pm 3.12}$ | $(0.40_{\pm 0.03})$ |
| ERM-Res. | $73.70_{\pm 3.19}$ | $(0.22_{\pm 0.04})$ | $71.19_{\pm 3.23}$ | $(0.24_{\pm 0.03})$ | $73.62_{\pm 1.54}$ | $(0.22_{\pm 0.02})$ | $71.03_{\pm 2.83}$ | $(0.25_{\pm 0.03})$ | $70.20_{\pm 3.73}$ | $(0.26_{\pm 0.03})$ |
| IRMv1 | $78.24_{\pm 0.79}$ | $(0.16_{\pm 0.01})$ | $74.83_{\pm 1.74}$ | $(0.20_{\pm 0.02})$ | $78.61_{\pm 2.24}$ | $(0.16_{\pm 0.02})$ | $76.28_{\pm 1.87}$ | $(0.18_{\pm 0.02})$ | $71.75_{\pm 2.03}$ | $(0.24_{\pm 0.02})$ |
| Fish | $77.23_{\pm 2.24}$ | $(0.19_{\pm 0.02})$ | $77.23_{\pm 1.32}$ | $(0.19_{\pm 0.01})$ | $78.24_{\pm 2.09}$ | $(0.18_{\pm 0.02})$ | $76.42_{\pm 1.95}$ | $(0.20_{\pm 0.02})$ | $\mathbf{73.92_{\pm 2.53}}$ | $(0.23_{\pm 0.03})$ |
| GroupDRO | $80.10_{\pm 1.66}$ | $(\mathbf{0.02_{\pm 0.01}})$ | $80.96_{\pm 1.33}$ | $(\mathbf{0.04_{\pm 0.02}})$ | $80.35_{\pm 1.01}$ | $(\mathbf{0.06_{\pm 0.02}})$ | $\mathbf{77.40_{\pm 1.16}}$ | $(\mathbf{0.08_{\pm 0.01}})$ | $71.86_{\pm 1.60}$ | $(\mathbf{0.22_{\pm 0.02}})$ |
| SAGM | $76.43_{\pm 2.37}$ | $(0.19_{\pm 0.02})$ | $79.05_{\pm 2.23}$ | $(0.17_{\pm 0.02})$ | $76.96_{\pm 4.36}$ | $(0.18_{\pm 0.03})$ | $79.86_{\pm 1.81}$ | $(0.16_{\pm 0.02})$ | $72.81_{\pm 3.10}$ | $(0.23_{\pm 0.03})$ |
| DiWA | $76.61_{\pm 2.15}$ | $(0.19_{\pm 0.02})$ | $76.71_{\pm 0.59}$ | $(0.19_{\pm 0.01})$ | $76.09_{\pm 0.69}$ | $(0.20_{\pm 0.01})$ | $75.83_{\pm 1.83}$ | $(0.20_{\pm 0.02})$ | $73.39_{\pm 1.31}$ | $(\mathbf{0.22_{\pm 0.01}})$ |
| TEP | $58.68_{\pm 4.72}$ | $(0.06_{\pm 0.19})$ | $60.42_{\pm 1.30}$ | $(0.09_{\pm 0.06})$ | $56.07_{\pm 3.35}$ | $(-0.04_{\pm 0.42})$ | $58.52_{\pm 4.36}$ | $(0.01_{\pm 0.25})$ | $59.23_{\pm 1.13}$ | $(0.18_{\pm 0.11})$ |
| RepLIn | $\mathbf{87.94_{\pm 1.46}}$ | $(0.08_{\pm 0.02})$ | $\mathbf{87.76_{\pm 2.30}}$ | $(0.10_{\pm 0.02})$ | $83.23_{\pm 2.67}$ | $(0.16_{\pm 0.03})$ | $73.63_{\pm 2.43}$ | $(0.25_{\pm 0.02})$ | $67.52_{\pm 2.30}$ | $(0.32_{\pm 0.03})$ |
| RepLIn-Res. | $\mathbf{88.46_{\pm 0.96}}$ | $(\mathbf{0.07_{\pm 0.01}})$ | $\mathbf{88.05_{\pm 1.04}}$ | $(\mathbf{0.08_{\pm 0.01}})$ | $\mathbf{87.91_{\pm 1.36}}$ | $(\mathbf{0.08_{\pm 0.01}})$ | $\mathbf{86.38_{\pm 0.85}}$ | $(\mathbf{0.10_{\pm 0.01}})$ | $\mathbf{78.41_{\pm 1.27}}$ | $(\mathbf{0.18_{\pm 0.02}})$ |

Table 2: **Results on Windmill dataset**: We evaluate the variants of RepLIn (highlighted in green) against the baselines on two metrics: interventional accuracy and relative accuracy drop on interventional data compared to observational. Compared to the baselines, RepLIn maintains its interventional accuracy as the proportion of interventional data during training ($\beta$) decreases. A similar trend is observed in the relative accuracy drop, where RepLIn remains significantly more robust than most baselines. The **best** and the **second-best** results are shown in different colors. "Res." stands for "Resampled".

## 5.2 Facial Attribute Prediction

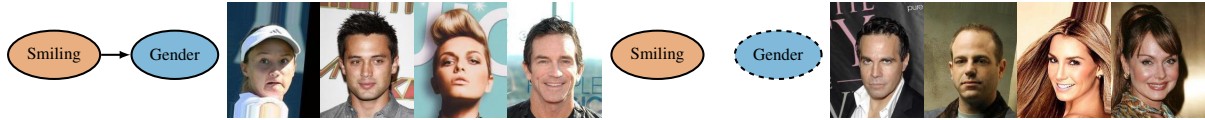

(a) Observational causal graph and samples  (b) Interventional causal graph and samples

Figure 7: Causal models for CelebA before and after intervention, along with images sampled from them.

We verify the utility of RepLIn for predicting facial attributes on the CelebA dataset (Liu et al., 2015). Among the 40 binary attributes that the images in the CelebA dataset are annotated with, we consider `smiling` and `gender` attributes for our prediction task. Since the true underlying relation between smile and gender is unknown, we adopt the resampling procedure from (Wang & Boddeti, 2022) to induce a desired causal relation between the attributes (`smiling` → `gender`) and obtain samples. Specifically, to simulate this causal relation, we sample `smiling` from Bernoulli(0.6) first, and then sample `gender` according to a probability distribution conditioned on the sampled `smiling` variable. We then sample a face image whose attribute labels match the sampled values from the semi-synthetic causal model. The diversity in the images is modeled as the result of unobserved noise variables. Note that, unlike in WINDMILL, these noise variables *may be* causally related to the attributes that we wish to predict, adding to the challenges in the dataset. The semi-synthetic causal model for this experiment and some sample images are shown in Fig. 7.

We first extract features from the face images using a ResNet-50 (He et al., 2016) model pre-trained on the ImageNet dataset (Deng et al., 2009). Then, similar to the architecture used for the WINDMILL experiments,

we employ a shallow MLP to act on these features, followed by a linear classifier to predict the attributes. Our loss functions act upon the features of the MLP. We use 30,000 samples for training and 15,000 for testing. We use the relative drop in interventional accuracy as the primary metric for comparing RepLIn-Resampled against ERM-Resampled. We also verify if the correlation between interventional feature dependence and the relative drop in accuracy observed in Sec. 3.3 on WINDMILL experiments holds in a more practical scenario.

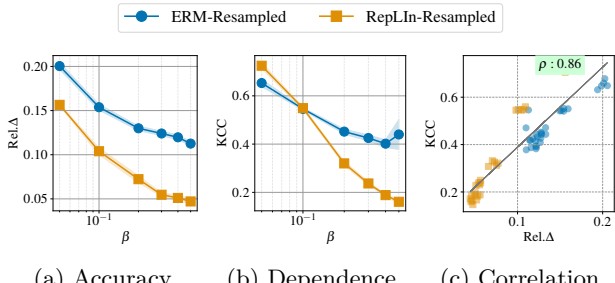

(a) Accuracy    (b) Dependence    (c) Correlation

Figure 8: **Facial Attribute Prediction:** (a) RepLIn has a lower relative drop in accuracy (Rel.$\Delta$) compared to ERM-Resampled. (b) Minimizing interventional feature dependence during training generalizes to testing. (c) Interventional feature dependence correlates positively with the relative drop in accuracy ($\rho = 0.86$).

Fig. 8 reports the experimental results on facial attribute prediction for various amounts of interventional training data. We make the following observations: **(1)** as the proportion of interventional data increases, the relative drop in accuracy (Rel.$\Delta$) in all methods decreases, **(2)** across all proportions of interventional data, RepLIn consistently outperforms the baseline in Rel.$\Delta$ by 4 to 7 percentage points, despite the potential challenges due to noise variable being causally related to the attributes of interest, **(3)** relative drop in accuracy and interventional feature dependence show strong positive correlation ($\rho = 0.86$), and **(4)** the interventional feature dependence of RepLIn steadily decreases as the amount of interventional data increases.

## 5.3 Toxicity Prediction in Text

We evaluate RepLIn on a toxicity prediction task on the CivilComments dataset (Borkan et al., 2019), which contains comments from online forums. We use a subset of this dataset labeled with identity attributes (such as `male`, `white`, `LGBTQ`, etc.) and toxicity scores by humans. The task is to classify each comment as toxic or not. Since prior works have identified gender bias in toxicity classifier models (Dixon et al., 2018; Park et al., 2018; Nozza et al., 2019), we will simulate a semi-synthetic causal model between the attribute `female` and toxicity: `female`$\rightarrow$ toxicity. During observation, both attributes assume identical binary values. During intervention, toxicity becomes independent of `female`. Input text comments are then sampled according to the attributes obtained from this semi-synthetic causal model. Similar to Sec. 5.2, we first extract features from the comments using BERT (Devlin et al., 2019) and train the baselines on these features. The baseline architecture consists of a linear layer to learn representations and a linear classifier layer to predict toxicity.

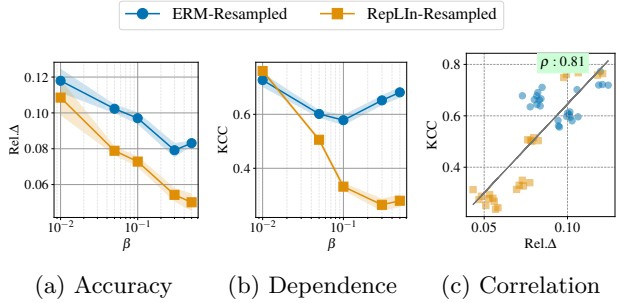

(a) Accuracy    (b) Dependence    (c) Correlation

Figure 9: **Toxicity Prediction in Text:** (a) RepLIn has lower interventional accuracy drop compared to ERM-Resampled; (b) Minimizing $\mathcal{L}_{\text{dep}}$ during training gives us representations that are independent during interventions; (c) The strong correlation between accuracy drop and interventional feature dependence further validates our hypothesis in Sec. 3.2.

Fig. 9 compares the performance of RepLIn against ERM-Resampled. Fig. 9b shows that enforcing independence between interventional features minimizes the interventional feature dependence during testing, although its effectiveness drops as $\beta$ approaches 0.01. RepLIn still outperforms the baseline in terms of the accuracy drop during interventions (Fig. 9a). We also note that RepLIn becomes more efficient in minimizing the interventional feature dependence as $\beta$ increases. The correlation between interventional feature dependence and Rel.$\Delta$ in Fig. 9c further validates our hypothesis in Sec. 3.2.

# 6 Discussion

## 6.1 How are representations learned by RepLIn different from those by ERM?

In this section, we qualitatively and quantitatively compare the interventional features learned by RepLIn and baselines to understand how RepLIn improves robustness against interventional distribution shift.

| Method | ERM | ERM-Resampled | IRMv1 | Fish | GroupDRO | RepLIn | RepLIn-Resampled |
|---|---|---|---|---|---|---|---|
| When $A = 0$ | $0.45 \pm 0.058$ | $0.423 \pm 0.105$ | $0.333 \pm 0.122$ | $0.341 \pm 0.111$ | $0.365 \pm 0.066$ | $\mathbf{0.15 \pm 0.03}$ | $\mathbf{0.188 \pm 0.032}$ |
| When $A = 1$ | $0.499 \pm 0.07$ | $0.456 \pm 0.11$ | $0.405 \pm 0.111$ | $0.37 \pm 0.116$ | $0.431 \pm 0.048$ | $\mathbf{0.183 \pm 0.058}$ | $\mathbf{0.168 \pm 0.047}$ |
| Average | $0.475 \pm 0.063$ | $0.439 \pm 0.105$ | $0.369 \pm 0.116$ | $0.355 \pm 0.113$ | $0.398 \pm 0.055$ | $\mathbf{0.166 \pm 0.035}$ | $\mathbf{0.178 \pm 0.036}$ |

Table 3: We compare the **Jensen-Shannon (JS) divergence** between interventional features from the baselines for WINDMILL with $\beta = 0.5$. Since the distribution of $\boldsymbol{F}_A^{\mathrm{int}}$ from a robust model is invariant to the value assumed by $B$ since $A \perp\!\!\!\perp B$ during interventions, JS divergence between $P(\boldsymbol{F}_A^{\mathrm{int}}|B = 0, A = a)$ and $P(\boldsymbol{F}_A^{\mathrm{int}}|B = 1, A = a)$ will be zero for a robust model, for $a \in \{0, 1\}$. Among the baselines, RepLIn achieves the lowest values of Jensen-Shannon divergence. The lowest and the second lowest scores are colored.

**Windmill dataset**: Robust representations of $A$ should change with $A$, but not $B$. We quantify their change with $B$ using the Jensen-Shannon (JS) divergence between the distributions of $\boldsymbol{F}_A^{\mathrm{int}}$ for a fixed value of $A$ and changing values of $B$. Tab. 3 shows the JS divergence between $P(\boldsymbol{F}_A^{\mathrm{int}}|B = 0, A = a)$ and $P(\boldsymbol{F}_A^{\mathrm{int}}|B = 1, A = a)$ (obtained through binning) for multiple baselines trained on WINDMILL dataset. JS divergence for a robust model will be zero. We observe that $\boldsymbol{F}_A^{\mathrm{int}}$ learned by RepLIn achieves the lowest JS divergence, meaning $\boldsymbol{F}_A^{\mathrm{int}}$ learned by RepLIn varies distributionally the least with $B$ among the baselines.

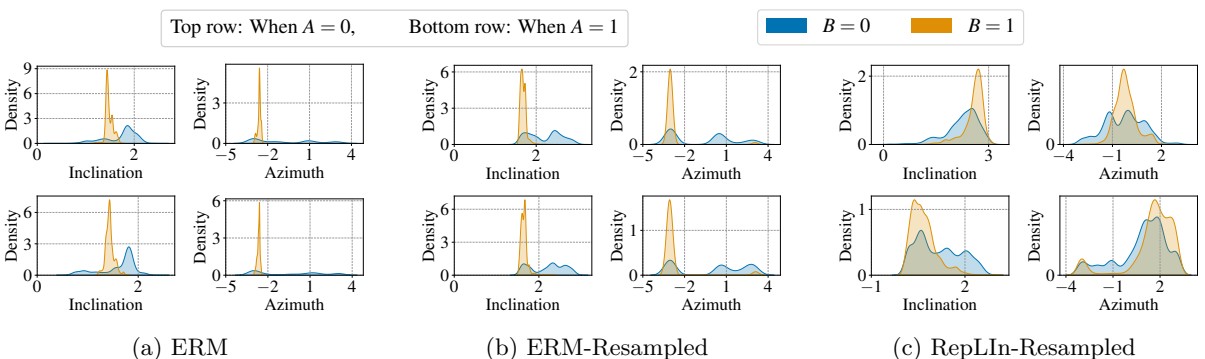

(a) ERM        (b) ERM-Resampled        (c) RepLIn-Resampled

Figure 10: Visualization of interventional features learned by various methods on WINDMILL dataset.

Since the learned representations were normalized to be on a unit sphere during training, we can examine them qualitatively through the histograms of the inclination and azimuth angles they subtend at the origin. We compare these histograms of $\boldsymbol{F}_A^{\mathrm{int}}$ learned by RepLIn-Resampled against the ERM baselines in Fig. 10. Each row shows the histograms for different values of $A$. Histograms for different values of $B$ are shown in different colors. Remember that the feature distributions for a robust model must change with $A$, but not $B$. We observed from the figure that the feature distributions of the baselines are affected by $B$ more than by $A$ due to the dependence between $\boldsymbol{F}_A^{\mathrm{int}}$ and $B$. However, the feature distributions learned by RepLIn change with $A$ and overlap significantly for different values of $B$. Thus, representations learned by RepLIn are more similar to those of a robust model. Similar visualizations for other baselines are provided in App. G.

**CelebA dataset**: We inspect the high-dimensional features learned on CelebA through their output attention maps obtained using Grad-CAM (Selvaraju et al., 2017). Fig. 11 shows the attention maps from models trained on the CelebA dataset with $\beta = 0.5$ for some samples with `smiling = 1` that were misclassified by ERM-Resampled but were correctly classified by RepLIn-Resampled. A robust model would attend to regions around the lips to make predictions about smiling. Observe that RepLIn-Resampled tends to focus more on the region around the lips (associated with smiling), while ERM-Resampled attends to other regions

of the image, such as hair and cap. This supports the trustworthiness of representations learned by RepLIn. GradCAM visualizations on the samples accurately classified by ERM, but not RepLIn, are shown in App. F.

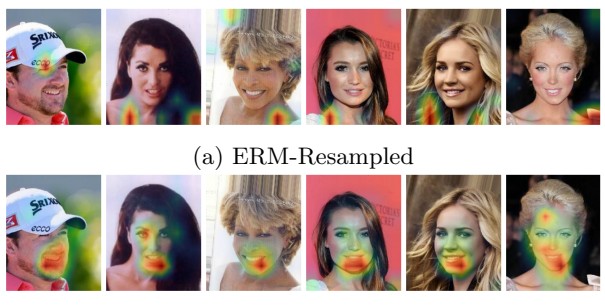

(a) ERM-Resampled

(b) RepLIn-Resampled

Figure 11: Consider these sample face images where the subjects are smiling. The ERM baseline misclassified these samples as not smiling, while RepLIn classified them correctly. We use GradCAM visualizations to identify the regions in the input images that the models used to make their predictions. The ERM model relied on factors such as hair and the presence of a hat that may correlate with gender to predict whether the subjects are smiling. In contrast, RepLIn attended to the lip regions to make predictions.

## 6.2 Scalability with number of nodes

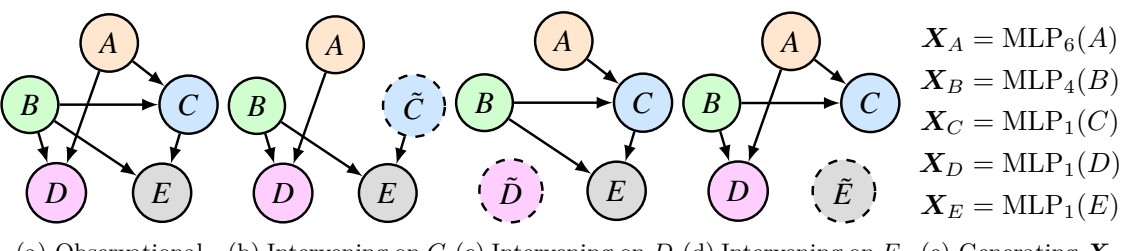

(a) Observational  (b) Intervening on $C$  (c) Intervening on $D$  (d) Intervening on $E$  (e) Generating $\boldsymbol{X}$

Figure 12: We construct a **5-variable causal graph** to demonstrate the scalability of our method with the number of nodes. To collect interventional data, we intervene on $C$, $D$, and $E$ separately and measure the performance drop in predicting $A$ and $B$ during these interventions. Intervened nodes in the graph have dashed borders. Note that we do not intervene on multiple targets at a time. The input data signal $\boldsymbol{X}$ is constructed as a concatenation of individual input signals, each being a function of a latent variable, i.e., $\boldsymbol{X} = \begin{bmatrix} \boldsymbol{X}_A^\top & \boldsymbol{X}_B^\top & \boldsymbol{X}_C^\top & \boldsymbol{X}_D^\top & \boldsymbol{X}_E^\top \end{bmatrix}^\top$ Here, $\mathrm{MLP}_l$ indicates a randomly initialized MLP with $l$ linear layers, each followed by a ReLU. We also add Gaussian noise sampled from $\mathcal{N}(0, 0.01)$ to the output of the MLP.

**Task**: In practice, the variable for which we wish to learn robust representations may have multiple child nodes. Therefore, it is imperative that RepLIn is scalable with both the number of intervened nodes and their parents. To verify this scalability, we use the causal graph shown in Fig. 12a with five latent variables. It consists of two binary source nodes, $A$ and $B$, and three binary derived nodes, $C$, $D$, and $E$. The observed signal $\boldsymbol{X}$ is a concatenation of attribute-specific observable components from each latent variable. The attribute-specific observable component $\boldsymbol{X}_p$ for the latent variable $p \in \{A, B, C, D, E\}$ is obtained by passing $p$ through a corresponding randomly initialized MLP and adding noise to its output (Fig. 12e). The MLPs used for $A$ and $B$ have more layers than those for $C$, $D$, and $E$ to entice the model to learn spurious correlations between parent and child nodes. The learning task is to predict the latent variables from $\boldsymbol{X}$.

**Training Setup**: During observations, $A$ and $B$ are sampled from independent Bernoulli(0.5) distributions, and the remaining nodes are obtained through the following logical expressions: $C := A \lor B$, $D := A \land B$, and $E := \neg B \land C$. Interventional training samples are collected by intervening on nodes $C$, $D$, and $E$ one at a time, as shown in Figs. 12b to 12d. The intervened variable assumes values from a Bernoulli(0.5) distribution independent of their parents. Each training batch has either observational or interventional samples, identical to the resampled setting. No batch will contain interventional data with different intervention targets. Thus, our method enforces the independence relations from at most one interventional target in each batch. The validation and test sets consist of samples collected during interventions on $C$, $D$, or $E$. Our primary metric will be the predictive accuracy for $A$ and $B$ during separate interventions on $C$, $D$, and $E$.

| Interventional target | Method | Predictive accuracy on $A$ | | | | Predictive accuracy on $B$ | | | |
|---|---|---|---|---|---|---|---|---|---|
| | | $\beta = 0.5$ | $\beta = 0.3$ | $\beta = 0.1$ | $\beta = 0.05$ | $\beta = 0.5$ | $\beta = 0.3$ | $\beta = 0.1$ | $\beta = 0.05$ |
| $C$ | ERM-Resampled | $79.71 \pm 0.30$ | $76.22 \pm 0.42$ | $\mathbf{73.97 \pm 0.39}$ | $73.56 \pm 0.36$ | $87.60 \pm 0.06$ | $85.45 \pm 0.23$ | $\mathbf{83.89 \pm 0.33}$ | $83.71 \pm 0.40$ |
| | RepLIn-Resampled | $\mathbf{95.37 \pm 0.97}$ | $\mathbf{78.77 \pm 0.54}$ | $72.15 \pm 0.31$ | $\mathbf{73.74 \pm 0.36}$ | $\mathbf{96.72 \pm 0.81}$ | $\mathbf{86.16 \pm 0.63}$ | $82.35 \pm 0.95$ | $82.43 \pm 0.65$ |
| $D$ | ERM-Resampled | $79.65 \pm 0.43$ | $75.47 \pm 0.64$ | $\mathbf{71.76 \pm 0.35}$ | $\mathbf{70.27 \pm 0.34}$ | $91.05 \pm 0.29$ | $90.21 \pm 0.27$ | $90.36 \pm 0.58$ | $90.55 \pm 0.74$ |
| | RepLIn-Resampled | $\mathbf{95.49 \pm 1.01}$ | $\mathbf{77.76 \pm 0.82}$ | $71.20 \pm 0.82$ | $68.80 \pm 0.79$ | $\mathbf{97.87 \pm 0.31}$ | $\mathbf{92.21 \pm 0.48}$ | $\mathbf{91.40 \pm 0.79}$ | $\mathbf{90.88 \pm 0.89}$ |
| $E$ | ERM-Resampled | $86.63 \pm 0.33$ | $81.90 \pm 0.26$ | $\mathbf{76.20 \pm 0.84}$ | $\mathbf{73.46 \pm 0.37}$ | $81.12 \pm 0.22$ | $78.00 \pm 0.48$ | $\mathbf{74.02 \pm 0.38}$ | $\mathbf{72.97 \pm 0.38}$ |
| | RepLIn-Resampled | $\mathbf{96.71 \pm 0.49}$ | $\mathbf{84.68 \pm 0.36}$ | $75.01 \pm 0.53$ | $71.52 \pm 0.87$ | $\mathbf{96.89 \pm 0.68}$ | $\mathbf{80.88 \pm 0.57}$ | $72.81 \pm 1.13$ | $71.60 \pm 0.59$ |

Table 4: **Results on 5-variable causal graph:** We compare the accuracy of RepLIn in predicting the source nodes $A$ and $B$ during interventions on non-source nodes $C$, $D$, and $E$ against that of ERM-Resampled. Our approach outperforms the baselines with sufficient interventional data.

**Observations**: The predictive test accuracies in Table 4 show that RepLIn significantly improves over the baseline with sufficient interventional training data ($\beta > 0.1$) for all intervention targets. When $\beta \leq 0.1$, RepLIn is comparable with the baseline. Our results demonstrate that the benefits of enforcing independence between interventional features extend to larger causal graphs with multiple intervention targets.

## 6.3 Limitations

Since RepLIn requires knowledge of the intervened node and its parent variables, RepLIn is affected by causal graph misspecification involving the intervened node. For example, if a parent of the intervened node is not known during training, RepLIn will not enforce the associated independence constraint during training and may lead to lower robustness against similar interventions at test time. RepLIn is also sensitive to imperfect interventions, where the intervened variable would be partially dependent on its parents, albeit with lowered dependence strength. Our experiments on imperfect interventions in App. B.4 showed that RepLIn's performance deteriorated when the interventional training data contained imperfect interventions, compared to a vanilla predictor trained with observational and perfect interventional data. However, it still outperformed vanilla predictors trained on the same dataset, especially at lower values of $\beta$. These results indicate that RepLIn works best when perfect interventional samples are available, and is more useful when interventional data is scarce and sample-efficient methods are required to improve robustness. RepLIn also requires tuning of the hyperparameters $\lambda_{\text{dep}}$ and $\lambda_{\text{self}}$, whose values typically increase with the complexity of the data generation process (App. H.1). The presence of additional hyperparameters also leaves RepLIn less attractive compared to intervention-specific predictors in those scenarios where inference samples carry information about the underlying intervention, such as with genetic data, where gene-specific interventions are usually known at inference time. Unlike RepLIn, these intervention-specific predictors can be trained using only ERM loss, provided sufficient interventional training samples are available. RepLIn is more suitable when inference-time intervention markers are unavailable.

## 7 Conclusion

This paper considered the problem of learning discriminative representations that are robust against known interventional distribution shifts and proposed a training algorithm for this objective that exploits the statistical independence induced by interventions in the underlying data generation process. First, we established a strong correlation between the accuracy drop during interventions and the statistical dependence between representations on interventional data. We then showed theoretically that minimizing linear dependence between interventional representations can improve the robustness of a linear model against interventional distribution shifts. Building on this result, we proposed RepLIn to learn representations that are robust against interventional distribution shift by explicitly enforcing statistical independence between learned representations on interventional data. Experimental evaluation of RepLIn across different causal graphs on both synthetic and real datasets on image and text modalities with semi-synthetic causal structures showed that RepLIn can improve predictive accuracy during interventions, even for low proportions of interventional data. RepLIn is also scalable to the number of causal attributes and can be used with continuous and discrete latent variables. We showed qualitatively and quantitatively that RepLIn is more successful in learning interventional representations that are unaffected by changes in their child nodes during interventions.

**Acknowledgments**

The authors thank the anonymous reviewers and the action editor for their constructive criticism and general feedback. GS also thanks Ramin Akbari for the useful discussions. This work was supported in part by the National Science Foundation (awards #2147116 and #2500983) and the Office of Naval Research (award #N00014-23-1-2417). Any opinions, findings, and conclusions or recommendations expressed in this material are those of the authors and do not necessarily reflect the views of NSF or ONR.

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

# Appendix

## A  Implementation details

We implement our models using PyTorch (Paszke et al., 2019) and use Adam (Kingma & Ba, 2015) as our optimizer with its default settings. Training hyperparameters for each dataset (such as the number of data points, training epochs, etc.) are shown in Tab. 5. For training stability, we warm up $\lambda_{\text{dep}}$ from 0 to its set value between $sN$ and $eN$ epochs where $N$ is the total number of epochs, and $s$ and $e$ are fractions shown in Tab. 5.

Table 5: List of hyperparameters used for each dataset.

| Dataset | #Training samples | Epochs | Batchsize | Initial LR | Scheduler | $\lambda_{\text{dep}}$ | $\lambda_{\text{self}}$ | Start $(s)$ | End $(e)$ |
|---|---|---|---|---|---|---|---|---|---|
| WINDMILL | 40,000 | 5000 | 1000 | 2e-3 | MultiStepLR(milestones=[1000], gamma=0.5) | 1 | 1 | 0.66 | 0.99 |
| CelebA | 30,000 | 2000 | 1000 | 1e-3 | MultiStepLR(milestones=[1000], gamma=0.1) | 20 | 2 | 0.01 | 0.99 |

For all methods, we first extract label-specific features from the inputs and pass them through a corresponding classifier to predict the label. The architecture of the feature extractor is the same for all methods on a given dataset, except on the WINDMILL dataset. The classification layer is a linear layer mapping from feature dimensions to the number of classes. The specific details for each dataset are provided below.

**Windmill dataset:** For ERM baselines, we use an MLP with two layers of size 40 and 1, with a ReLU activation after each layer (except the last) to extract the features. However, we observed that enforcing independence using 1-dimensional features was difficult. Therefore, we used 2-dimensional features for RepLIn, which were then normalized to lie on a sphere.

**CelebA dataset:** We first extract features from the raw image using a ResNet-50 (He et al., 2016) pre-trained on ImageNet (Deng et al., 2009). Although these features are not optimal for face attribute prediction, they are useful for face verification (Sharif Razavian et al., 2014). Additionally, it makes the binary attribute prediction task more challenging. We extract attribute-specific features from this input using a linear layer that maps it to a 500-dimensional space.

## B  Theoretical Motivation for RepLIn

In Sec. 3.4, we theoretically motivated RepLIn. This section explains the motivation with detailed proof.

**Sketch of proof:** First, we estimate the statistical risk in predicting the latent variables from interventional data from representations learned by arbitrary linear feature extractors and classifiers. In this statistical risk, we will identify a term that is the source of performance drop during interventions. We will then show that the optimal ERM models will suffer from this performance drop when trained on a dataset comprising observational and interventional data. Finally, we show that minimizing linear dependence between interventional features can lead to robust linear feature extractors.

| Entity | Notation | Examples |
|---|---|---|
| Scalar | Regular lowercase characters | $a, \gamma$ |
| Random variable | Regular serif uppercase characters | $A$ |
| Random vector | Bold serif uppercase characters | $\boldsymbol{A}$ |
| Distribution of a random variable $A$ | $P$ with subscript | $P_A$ |

Table 6: Mathematical notation used in the proof.

**Setup:** We follow the same mathematical notation as the main paper, shown in Tab. 6. The input data $\boldsymbol{X}$ is generated as a function of two latent variables of interest, $A$ and $B$. There are noise variables collectively denoted by $\boldsymbol{U}$ that participate in the data generation but are not of learning interest. Our task is to predict $A$ and $B$ from $\boldsymbol{X}$. $A$ and $B$ are causally related during observation. For ease of exposition, we will consider a simple linear relation $B \coloneqq w_{AB}A$. This causal relation breaks when we intervene on $B$. The intervened variable is denoted with an added apostrophe (i.e., $B'$). The data generation process can be written in the form of a structural causal model as follows:

$$A \sim P_A \qquad\qquad X_A := w_A A + U_A$$
$$B' \sim P_{B'} \qquad\qquad X_B := w_B B + U_B$$
$$B := w_{AB} A \text{ (during observations)}$$
$$B := B' \text{ (during interventions)} \qquad \boldsymbol{X} = \begin{bmatrix} X_A \\ X_B \end{bmatrix}$$
$$U_A, U_B \sim P_{\mathsf{U}}$$

**Training:** The distribution from which training data is sampled is denoted by $P_{\text{train}}$. The training data consists of both observational and interventional samples, which themselves come from distributions $P_{\text{obs}}$ and $P_{\text{int}}$. We are interested in the scenario where $(1 - \beta)$ proportion of the training data is observational, while the remaining $\beta$ proportion is interventional, where $0 < \beta < 1$. The training distribution can be represented as a mixture of observational and interventional distributions as follows:

$$P_{\text{train}}(\boldsymbol{X}, A, B) = (1 - \beta)P_{\text{obs}}(\boldsymbol{X}, A, B) + \beta P_{\text{int}}(\boldsymbol{X}, A, B)$$

Typically, we assume $\beta \ll 1$. We will also assume that $A$, $B$, $\boldsymbol{U}$, and $\boldsymbol{X}$ have zero mean, so that we may use linear models without bias terms to extract representations corresponding to the variables of interest and train linear classifiers on these representations. The corresponding classifiers are parameterized by $\boldsymbol{c}^{(A)}$ and $\boldsymbol{c}^{(B)}$. The predictions are made by the classifiers from the learned representations as $\hat{A} = \boldsymbol{c}^{(A)\top}\boldsymbol{\Theta}^{(A)\top}\boldsymbol{X}$ and $\hat{B} = \boldsymbol{c}^{(B)\top}\boldsymbol{\Theta}^{(B)\top}\boldsymbol{X}$. The models are trained by minimizing the mean squared error on the training data,
$\mathcal{L}_{\text{MSE}} = \mathbb{E}_{P_{\text{train}}}\left[ \left( \left\| A - \hat{A} \right\|_2^2 + \left\| B - \hat{B} \right\|_2^2 \right) \right]$.

## B.1 Statistical Risk in Predicting Interventional Latent Samples

The model predicts $\hat{A}$ and $\hat{B}$ from $\boldsymbol{X}$ during inference. The statistical squared error in predicting $A$ from interventional samples can be written as,

$$E_A = \mathbb{E}_{P_{\text{int}}}\left[ \left( A - \hat{A} \right)^2 \right] = \mathbb{E}_{P_{\text{int}}}\left[ \left( A - \boldsymbol{c}^{(A)\top}\boldsymbol{\Theta}^{(A)\top}\boldsymbol{X} \right)^2 \right] \tag{7}$$

The expectation is taken over the interventional distribution over $\boldsymbol{X}, A, B, \boldsymbol{U}$ denoted by $P_{\text{int}}$. $\boldsymbol{\Theta}^{(A)}$ can be written in terms of constituent parameter vectors as $\boldsymbol{\Theta}^{(A)} = \begin{bmatrix} \boldsymbol{\theta}_A^{(A)\top} \\ \boldsymbol{\theta}_B^{(A)\top} \end{bmatrix}$. The predicted latent $\hat{A}$ can hence be written in terms of these vectors as,

$$\hat{A} = \boldsymbol{c}^{(A)\top}\boldsymbol{\Theta}^{(A)\top}\boldsymbol{X} = \boldsymbol{c}^{(A)\top}\left( X_A\boldsymbol{\theta}_A^{(A)} + X_{B'}\boldsymbol{\theta}_B^{(A)} + \boldsymbol{\Theta}^{(A)\top}\boldsymbol{U} \right)$$

$$= w_A A \boldsymbol{c}^{(A)\top}\boldsymbol{\theta}_A^{(A)} + w_B B' \boldsymbol{c}^{(A)\top}\boldsymbol{\theta}_B^{(A)} + \boldsymbol{c}^{(A)\top}\boldsymbol{\Theta}^{(A)\top}\boldsymbol{U}$$

$$\therefore \left( A - \boldsymbol{c}^{(A)\top}\boldsymbol{\Theta}^{(A)\top}\boldsymbol{X} \right)^2 = \left( \left( 1 - w_A\boldsymbol{c}^{(A)\top}\boldsymbol{\theta}_A^{(A)} \right) A + w_B B' \boldsymbol{c}^{(A)\top}\boldsymbol{\theta}_B^{(A)} + \boldsymbol{c}^{(A)\top}\boldsymbol{\Theta}^{(A)\top}\boldsymbol{U} \right)^2$$

$$= \left( 1 - w_A\boldsymbol{c}^{(A)\top}\boldsymbol{\theta}_A^{(A)} \right)^2 A^2 + \left( w_B\boldsymbol{c}^{(A)\top}\boldsymbol{\theta}_B^{(A)} \right)^2 B'^2 + \tilde{U}^2$$

$$+ 2\left( 1 - w_A\boldsymbol{c}^{(A)\top}\boldsymbol{\theta}_A^{(A)} \right)\left( w_B\boldsymbol{c}^{(A)\top}\boldsymbol{\theta}_B^{(A)} \right) AB'$$

$$+ 2\left( 1 - w_A\boldsymbol{c}^{(A)\top}\boldsymbol{\theta}_A^{(A)} \right)\tilde{U}A + 2\left( w_B\boldsymbol{c}^{(A)\top}\boldsymbol{\theta}_B^{(A)} \right)\tilde{U}B' \tag{8}$$

$$\therefore E_A = \mathbb{E}_{P_{\text{int}}}\left[ \left( 1 - w_A\boldsymbol{c}^{(A)\top}\boldsymbol{\theta}_A^{(A)} \right)^2 A^2 + \left( w_B\boldsymbol{c}^{(A)\top}\boldsymbol{\theta}_B^{(A)} \right)^2 B'^2 + \tilde{U}^2 \right]$$

$$+ 2\mathbb{E}_{P_{\text{int}}}\left[ \left( 1 - w_A\boldsymbol{c}^{(A)\top}\boldsymbol{\theta}_A^{(A)} \right)\left( w_B\boldsymbol{c}^{(A)\top}\boldsymbol{\theta}_B^{(A)} \right) AB' \right]$$

$$+ 2\mathbb{E}_{P_{\text{int}}}\left[ \left( 1 - w_A\boldsymbol{c}^{(A)\top}\boldsymbol{\theta}_A^{(A)} \right)\tilde{U}A + 2\left( w_B\boldsymbol{c}^{(A)\top}\boldsymbol{\theta}_B^{(A)} \right)\tilde{U}B' \right]$$

where $\tilde{U} = \boldsymbol{c}^{(A)\top}\boldsymbol{\Theta}^{(A)\top}\boldsymbol{U} = \boldsymbol{c}^{(A)\top}\boldsymbol{\theta}_A^{(A)}U_A + \boldsymbol{c}^{(A)\top}\boldsymbol{\theta}_B^{(A)}U_B$. $\boldsymbol{U}$ denotes exogenous variables that are independent of $A$ and $B$. Due to interventions, we also have $A \perp\!\!\!\perp B$. The expectation of $AB'$ will be zero since

they are independent and have zero means marginally. Similarly, the expectation of the products of $\tilde{U}$ with $A$ and $B$ will be zero. Therefore,

$$E_A = \underbrace{\left(1 - w_A \boldsymbol{c}^{(A)\top} \boldsymbol{\theta}_A^{(A)}\right)^2 \rho_A^2 + \left(\boldsymbol{c}^{(A)\top} \boldsymbol{\theta}_A^{(A)}\right)^2 \rho_{\boldsymbol{U}_A}^2}_{E_A^{(1)}} + \underbrace{\left(w_B \boldsymbol{c}^{(A)\top} \boldsymbol{\theta}_B^{(A)}\right)^2 \rho_{B'}^2 + \left(\boldsymbol{c}^{(A)\top} \boldsymbol{\theta}_B^{(A)}\right)^2 \rho_{\boldsymbol{U}_B}^2}_{E_A^{(2)}} \quad (9)$$

where $\rho_A^2 = \mathbb{E}_{P_{\text{int}}}\left[A^2\right]$, $\rho_{B'}^2 = \mathbb{E}_{P_{\text{int}}}\left[B'^2\right]$, $\rho_{\boldsymbol{U}_A}^2 = \mathbb{E}_{P_{\text{int}}}\left[U_A^2\right]$, and $\rho_{\boldsymbol{U}_B}^2 = \mathbb{E}_{P_{\text{int}}}\left[U_B^2\right]$.

**Statistical risk for a robust model:** We are interested in robustness against interventional distribution shifts. The predictions of $A$ by a robust model are unaffected by interventions on its child variable $B$. If $\hat{A}$ must not depend on $B'$, then the corresponding representation $\boldsymbol{F}_A$ must not depend on it either, i.e., $\boldsymbol{\theta}_B^{(A)}$ must be a zero vector. Eq. (9) has two terms: $E_A^{(1)}$ and $E_A^{(2)}$. Therefore, a robust model will have $E_A^{(2)} = 0$ since $\boldsymbol{\theta}_B^{(A)} = \boldsymbol{0}$. Therefore, showing that an optimal ERM model has a non-zero $\boldsymbol{\theta}_B^{(A)}$ is sufficient to show that the model is not robust.

## B.2 Optimal ERM model

The optimal ERM model can be obtained by minimizing the expected risk in predicting the latent attributes. Since parameters are not shared between the prediction of $\boldsymbol{a}$ and $\boldsymbol{b}$, we can consider their optimization separately. We will consider the optimization of the parameters for predicting $\boldsymbol{a}$ since we are interested in the performance drop in predicting $A$ from interventional data.

$$\boldsymbol{\Theta}^{(A)*}, \boldsymbol{c}^{(A)*} = \underset{\boldsymbol{\Theta}^{(A)}, \boldsymbol{c}^{(A)}}{\operatorname{argmin}} \; \mathbb{E}_{P_{\text{train}}}\left[\left(A - \boldsymbol{c}^{(A)\top} \boldsymbol{\Theta}^{(A)\top} \boldsymbol{X}\right)^2\right]$$

where $P_{\text{train}}$ is the joint distribution of $(\boldsymbol{X}, A, B)$ during training. As mentioned earlier, $P_{\text{train}}$ is a mixture of observational distribution $P_{\text{obs}}$ and interventional distribution $P_{\text{int}}$, with $(1-\beta)$ and $\beta$ acting as the mixture weights. Therefore, the training objective can be rewritten as,

$$\boldsymbol{\Theta}^{(A)*}, \boldsymbol{c}^{(A)*} = \underset{\boldsymbol{\Theta}^{(A)}, \boldsymbol{c}^{(A)}}{\operatorname{argmin}} \; J(\boldsymbol{\Theta}^{(A)}, \boldsymbol{c}^{(A)})$$

$$\text{where, } J(\boldsymbol{\Theta}^{(A)}, \boldsymbol{c}^{(A)}) = \left((1-\beta)\mathbb{E}_{P_{\text{obs}}}\left[\left(A - \boldsymbol{c}^{(A)\top} \boldsymbol{\Theta}^{(A)\top} \boldsymbol{X}\right)^2\right] + \beta \mathbb{E}_{P_{\text{int}}}\left[\left(A - \boldsymbol{c}^{(A)\top} \boldsymbol{\Theta}^{(A)\top} \boldsymbol{X}\right)^2\right]\right) \quad (10)$$

Expanding the error term on observational data, we have,

$$\boldsymbol{c}^{(A)\top} \boldsymbol{\Theta}^{(A)\top} \boldsymbol{X} = \boldsymbol{c}^{(A)\top} \left(X_A \boldsymbol{\theta}_A^{(A)} + X_B \boldsymbol{\theta}_B^{(A)} + \boldsymbol{\Theta}^{(A)\top} \boldsymbol{U}\right)$$

$$= w_A A \boldsymbol{c}^{(A)\top} \boldsymbol{\theta}_A^{(A)} + w_B B \boldsymbol{c}^{(A)\top} \boldsymbol{\theta}_B^{(A)} + \boldsymbol{c}^{(A)\top} \boldsymbol{\Theta}^{(A)\top} \boldsymbol{U}$$

$$= w_A A \boldsymbol{c}^{(A)\top} \boldsymbol{\theta}_A^{(A)} + w_B w_{AB} A \boldsymbol{c}^{(A)\top} \boldsymbol{\theta}_B^{(A)} + \boldsymbol{c}^{(A)\top} \boldsymbol{\Theta}^{(A)\top} \boldsymbol{U}$$

$$\therefore \left(A - \boldsymbol{c}^{(A)\top} \boldsymbol{\Theta}^{(A)\top} \boldsymbol{X}\right)^2 = \left(A - w_A A \boldsymbol{c}^{(A)\top} \boldsymbol{\theta}_A^{(A)} - w_B w_{AB} A \boldsymbol{c}^{(A)\top} \boldsymbol{\theta}_B^{(A)} - \boldsymbol{c}^{(A)\top} \boldsymbol{\Theta}^{(A)\top} \boldsymbol{U}\right)^2$$

$$= \left(\left(1 - w_A \boldsymbol{c}^{(A)\top} \boldsymbol{\theta}_A^{(A)} - w_B w_{AB} \boldsymbol{c}^{(A)\top} \boldsymbol{\theta}_B^{(A)}\right) A - \boldsymbol{c}^{(A)\top} \boldsymbol{\Theta}^{(A)\top} \boldsymbol{U}\right)^2$$

$$= \left(1 - w_A \boldsymbol{c}^{(A)\top} \boldsymbol{\theta}_A^{(A)} - w_B w_{AB} \boldsymbol{c}^{(A)\top} \boldsymbol{\theta}_B^{(A)}\right)^2 A^2 + \tilde{U}^2$$

$$\quad - 2\left(1 - w_A \boldsymbol{c}^{(A)\top} \boldsymbol{\theta}_A^{(A)} - w_B w_{AB} \boldsymbol{c}^{(A)\top} \boldsymbol{\theta}_B^{(A)}\right) A \tilde{U}$$

where $\tilde{U} = \boldsymbol{c}^{(A)\top} \boldsymbol{\Theta}^{(A)\top} \boldsymbol{U} = U_A \boldsymbol{c}^{(A)\top} \boldsymbol{\theta}_A^{(A)} + U_B \boldsymbol{c}^{(A)\top} \boldsymbol{\theta}_B^{(A)}$ from App. B.1. Since the exogenous variable $\boldsymbol{U}$ is independent of $A$ and $B$, the expectation of their products over the observational distribution becomes zero.

Therefore,

$$\mathbb{E}_{P_{\text{obs}}}\left[\left(A - \boldsymbol{c}^{(A)\top}\boldsymbol{\Theta}^{(A)\top}\boldsymbol{X}\right)^2\right] = \left(1 - w_A\boldsymbol{c}^{(A)\top}\boldsymbol{\theta}_A^{(A)} - w_B w_{AB}\boldsymbol{c}^{(A)\top}\boldsymbol{\theta}_B^{(A)}\right)^2 \mathbb{E}_{P_{\text{obs}}}\left[A^2\right] + \mathbb{E}_{P_{\text{obs}}}\left[\tilde{U}^2\right]$$

$$= \left(1 - w_A\boldsymbol{c}^{(A)\top}\boldsymbol{\theta}_A^{(A)} - w_B w_{AB}\boldsymbol{c}^{(A)\top}\boldsymbol{\theta}_B^{(A)}\right)^2 \rho_A^2 + \left(\boldsymbol{c}^{(A)\top}\boldsymbol{\theta}_A^{(A)}\right)^2 \rho_{\boldsymbol{U}_A}^2 + \left(\boldsymbol{c}^{(A)\top}\boldsymbol{\theta}_B^{(A)}\right)^2 \rho_{\boldsymbol{U}_B}^2 \tag{11}$$

Note that, $\rho_A^2 = \mathbb{E}_{P_{\text{obs}}}\left[A^2\right]$, $\rho_{\boldsymbol{U}_A}^2 = \mathbb{E}_{P_{\text{obs}}}\left[U_A^2\right]$, and $\rho_{\boldsymbol{U}_B}^2 = \mathbb{E}_{P_{\text{obs}}}\left[U_B^2\right]$ similar to App. B.1 since these values are unaffected by interventions. The expansion of the error term on interventional data was derived in Eq. (9). Thus, the overall training objective Eq. (10) can be written as,

$$J(\boldsymbol{\Theta}^{(A)}, \boldsymbol{c}^{(A)}) = (1-\beta)\left(\left(1 - w_A\boldsymbol{c}^{(A)\top}\boldsymbol{\theta}_A^{(A)} - w_B w_{AB}\boldsymbol{c}^{(A)\top}\boldsymbol{\theta}_B^{(A)}\right)^2 \rho_A^2 + \left(\boldsymbol{c}^{(A)\top}\boldsymbol{\theta}_A^{(A)}\right)^2 \rho_{\boldsymbol{U}_A}^2 + \left(\boldsymbol{c}^{(A)\top}\boldsymbol{\theta}_B^{(A)}\right)^2 \rho_{\boldsymbol{U}_B}^2\right)$$

$$+ \beta\left(\left(1 - w_A\boldsymbol{c}^{(A)\top}\boldsymbol{\theta}_A^{(A)}\right)^2 \rho_A^2 + \left(w_B\boldsymbol{c}^{(A)\top}\boldsymbol{\theta}_B^{(A)}\right)^2 \rho_{B'}^2 + \left(\boldsymbol{c}^{(A)\top}\boldsymbol{\theta}_A^{(A)}\right)^2 \rho_{\boldsymbol{U}_A}^2 + \left(\boldsymbol{c}^{(A)\top}\boldsymbol{\theta}_B^{(A)}\right)^2 \rho_{\boldsymbol{U}_B}^2\right)$$

We set $\psi_1 = \boldsymbol{c}^{(A)\top}\boldsymbol{\theta}_A^{(A)}$ and $\psi_2 = \boldsymbol{c}^{(A)\top}\boldsymbol{\theta}_B^{(A)}$. Since ERM jointly optimizes the feature extractors and the classifiers, no unique solution minimizes the prediction loss. For example, scaling the feature extractor parameters by an arbitrary constant scalar $\gamma$ and the classifier parameters by $1/\gamma$ will give the same error. Therefore, we can optimize $J(\boldsymbol{\Theta}^{(A)}, \boldsymbol{c}^{(A)})$ over $\psi_1$ and $\psi_2$, similar to (Arjovsky et al., 2019).

$$J(\boldsymbol{\Theta}^{(A)}, \boldsymbol{c}^{(A)}) = (1-\beta)\left((1 - w_A\psi_1 - w_B w_{AB}\psi_2)^2 \rho_A^2 + \psi_1^2\rho_{\boldsymbol{U}_A}^2 + \psi_2^2\rho_{\boldsymbol{U}_B}^2\right)$$

$$+ \beta\left((1 - w_A\psi_1)^2 \rho_A^2 + w_B^2\psi_2^2\rho_{B'}^2 + \psi_1^2\rho_{\boldsymbol{U}_A}^2 + \psi_2^2\rho_{\boldsymbol{U}_B}^2\right) \tag{12}$$

The optimal values of $\psi_1$ and $\psi_2$ are the stationary points of $J(\boldsymbol{\Theta}^{(A)}, \boldsymbol{c}^{(A)})$ (denoted by $J$ for brevity). Thus the optimal parameter values can be solved for by taking the first-order derivatives of $J$ w.r.t. $\psi_1$ and $\psi_2$ and setting them to zero.

$$\frac{\partial J}{\partial \psi_1} = 2(1-\beta)\left(-(1 - w_A\psi_1 - w_B w_{AB}\psi_2)w_A\rho_A^2 + \psi_1\rho_{\boldsymbol{U}_A}^2\right) + 2\beta\left(-(1 - w_A\psi_1)w_A\rho_A^2 + \psi_1\rho_{\boldsymbol{U}_A}^2\right)$$

$$\frac{\partial J}{\partial \psi_2} = 2(1-\beta)\left(-(1 - w_A\psi_1 - w_B w_{AB}\psi_2)w_B w_{AB}\rho_A^2 + \psi_2\rho_{\boldsymbol{U}_B}^2\right) + 2\beta\left(w_B^2\psi_2\rho_{B'}^2 + \psi_2\rho_{\boldsymbol{U}_B}^2\right)$$

Setting $\frac{\partial J}{\partial \psi_1} = \frac{\partial J}{\partial \psi_2} = 0$, we have,

$$\left(w_A^2\rho_A^2 + \rho_{\boldsymbol{U}_A}^2\right)\psi_1 \qquad\qquad +(1-\beta)w_A w_B w_{AB}\rho_A^2\psi_2 \qquad\qquad -w_A\rho_A^2 \quad = 0$$

$$(1-\beta)w_A w_B w_{AB}\rho_A^2\psi_1 \quad + \left(\beta w_B^2\rho_{B'}^2 + (1-\beta)w_B^2 w_{AB}^2\rho_A^2 + \rho_{\boldsymbol{U}_B}^2\right)\psi_2 \quad -(1-\beta)w_B w_{AB}\rho_A^2 \quad = 0$$

The equations are of the form $u_1\psi_1 + v_1\psi_2 + w_1 = 0$ and $u_2\psi_1 + v_2\psi_2 + w_2 = 0$. We can solve for $\psi_2$ as $\psi_2 = \frac{w_2 u_1 - w_1 u_2}{v_1 u_2 - v_2 u_1}$. Since we are only interested in probing the robustness of ERM models, we will check if $\psi_2$ is zero instead of fully solving the system of linear equations. $E_A^{(2)}$ in Eq. (9) is zero if $\psi_2 = 0$, i.e. if $w_2 u_1 - w_1 u_2 = 0$.

$$w_2 u_1 - w_1 u_2 = -(1-\beta)w_B w_{AB}\left(w_A^2\rho_A^2 + \rho_{\boldsymbol{U}_A}^2\right)\rho_A^2 + 4(1-\beta)w_A^2 w_B w_{AB}\rho_A^4$$

$$= -(1-\beta)w_B w_{AB}\rho_A^2\rho_{\boldsymbol{U}_A}^2$$

Unless the training data is entirely composed of interventional data (i.e., $\beta = 1$), $w_2 u_1 - w_1 u_2$ is not zero. Thus, the optimal ERM model is not robust against interventional distribution shifts.

## B.3 Minimizing Linear Dependence

In Sec. 3.3, we showed that dependence between interventional features correlated positively with the drop in accuracy on interventional data. We will now verify if minimizing dependence between interventional

features can minimize the drop in accuracy. For ease of exposition, we will minimize the linear dependence between interventional features instead of enforcing statistical independence. The interventional features are given by $\boldsymbol{F}_A = \boldsymbol{\Theta}^{(A)\top}\boldsymbol{X}$ and $\boldsymbol{F}'_B = \boldsymbol{\Theta}^{(B)\top}\boldsymbol{X}$.

$$\boldsymbol{F}_A = \boldsymbol{\Theta}^{(A)\top}\boldsymbol{X} = \begin{bmatrix} \boldsymbol{\theta}_A^{(A)} & \boldsymbol{\theta}_B^{(A)} \end{bmatrix} \begin{bmatrix} X_A \\ X_B \end{bmatrix}$$

$$= X_A\boldsymbol{\theta}_A^{(A)} + X_B\boldsymbol{\theta}_B^{(A)}$$

$$\boldsymbol{F}'_B = \boldsymbol{\Theta}^{(B)\top}\boldsymbol{X} = \begin{bmatrix} \boldsymbol{\theta}_A^{(B)} & \boldsymbol{\theta}_B^{(B)} \end{bmatrix} \begin{bmatrix} X_A \\ X_B \end{bmatrix}$$

$$= X_A\boldsymbol{\theta}_A^{(B)} + X_B\boldsymbol{\theta}_B^{(B)}$$

To define linear independence between interventional features, we use the following definition of cross-covariance from (Gretton et al., 2005):

**Definition 1.** *The* cross-covariance operator *associated with the joint probability $p_{XY}$ is a linear operator $C_{XY} : \mathcal{G} \to \mathcal{F}$ defined as*

$$C_{XY} = \mathbb{E}_{XY}\left[(\phi(X) - \mu_X) \otimes (\psi(Y) - \mu_Y)\right]$$

*where $\mathcal{G}$ and $\mathcal{F}$ are reproducing kernel Hilbert spaces (RKHSs) defined by feature maps $\phi$ and $\psi$ respectively, and $\otimes$ is the tensor product defined as follows*

$$(f \otimes g)h := f\langle g, h\rangle_{\mathcal{G}} \text{ for all } h \in \mathcal{G}$$

*where $\langle \cdot, \cdot \rangle$ is the inner product defined over $\mathcal{G}$.*

In our case, instead of RKHS, we have finite-dimensional feature space $\mathbb{R}^d$. Therefore, we have the cross-covariance matrix as follows,

$$C_{XY} = \mathbb{E}_{XY}\left[\phi(X) \otimes \psi(Y)\right] = \mathbb{E}_{XY}\left[\phi(X)\psi(Y)^{\top}\right]$$

given that the feature maps have zero mean. Following the definition of HSIC (Gretton et al., 2005), linear dependence in the finite-dimensional case between $X$ and $Y$ is defined as the Frobenius norm of the cross-covariance matrix. Therefore, we define the linear dependence loss between the interventional features as,

$$\mathcal{L}_{\text{dep}} = \text{Dep}\left(\boldsymbol{F}_A, \boldsymbol{F}'_B\right) = \left\|\mathbb{E}_{P_{\text{int}}}\left[\boldsymbol{F}_A\boldsymbol{F}'^{\top}_B\right]\right\|_F^2 \tag{13}$$

Leveraging the independence relations during interventions, we can expand Eq. (13) as,

$$\mathbb{E}_{P_{\text{int}}}\left[\boldsymbol{F}_A\boldsymbol{F}'^{\top}_B\right] = \mathbb{E}_{P_{\text{int}}}\left[\left(X_A\boldsymbol{\theta}_A^{(A)} + X_B\boldsymbol{\theta}_B^{(A)}\right)\left(X_A\boldsymbol{\theta}_A^{(B)} + X_B\boldsymbol{\theta}_B^{(B)}\right)^{\top}\right]$$

$$= \mathbb{E}_{P_{\text{int}}}\left[X_A^2\boldsymbol{\theta}_A^{(A)}\boldsymbol{\theta}_A^{(B)\top} + X_AX_B\boldsymbol{\theta}_A^{(A)}\boldsymbol{\theta}_B^{(B)\top} + X_AX_B\boldsymbol{\theta}_B^{(A)}\boldsymbol{\theta}_A^{(B)\top} + X_B^2\boldsymbol{\theta}_B^{(A)}\boldsymbol{\theta}_B^{(B)\top}\right]$$

$$= (w_A^2\rho_A^2 + \rho_{\boldsymbol{U}_A}^2)\boldsymbol{\theta}_A^{(A)}\boldsymbol{\theta}_A^{(B)\top} + (w_B^2\rho_{B'}^2 + \rho_{\boldsymbol{U}_B}^2)\boldsymbol{\theta}_B^{(A)}\boldsymbol{\theta}_B^{(B)\top}$$

$$\therefore \mathcal{L}_{\text{dep}} = \left\|(w_A^2\rho_A^2 + \rho_{\boldsymbol{U}_A}^2)\boldsymbol{\theta}_A^{(A)}\boldsymbol{\theta}_A^{(B)\top} + (w_B^2\rho_{B'}^2 + \rho_{\boldsymbol{U}_B}^2)\boldsymbol{\theta}_B^{(A)}\boldsymbol{\theta}_B^{(B)\top}\right\|_F^2$$

In the last step, all cross-covariance terms are zero due to the independence of the corresponding random variables in the causal graph. The dependence loss is the Frobenius norm of a sum of rank-one matrices $\boldsymbol{\theta}_A^{(A)}\boldsymbol{\theta}_A^{(B)\top}$ and $\boldsymbol{\theta}_B^{(A)}\boldsymbol{\theta}_B^{(B)\top}$. Consider the following general form: $\boldsymbol{Z} = \boldsymbol{a}\boldsymbol{b}^{\top} + \boldsymbol{c}\boldsymbol{d}^{\top}$. Then $Z_{ij} = a_ib_j + c_id_j$.

$$\|\boldsymbol{Z}\|_F^2 = \sum_{ij}(a_ib_j + c_id_j)^2$$

$\|\boldsymbol{Z}\|_F^2$ is a sum of squares and thus is zero iff $a_ib_j + c_id_j = 0$, $\forall i, j$. Therefore, $\mathcal{L}_{\text{dep}}$ is minimized when $\theta_{Ai}^{(A)}\theta_{Aj}^{(B)} + \theta_{Bi}^{(A)}\theta_{Bj}^{(B)} = 0$, $\forall i, j$. The potential solutions that minimize $\mathcal{L}_{\text{dep}}$ are (1) $\boldsymbol{\theta}_A^{(A)} = \boldsymbol{\theta}_B^{(A)} = \boldsymbol{\theta}_A^{(B)} =$

$\boldsymbol{\theta}_B^{(B)} = \mathbf{0}$, (2) $\boldsymbol{\theta}_A^{(A)} = \pm\gamma\boldsymbol{\theta}_B^{(A)}$ and $\gamma\boldsymbol{\theta}_A^{(B)} = \mp\boldsymbol{\theta}_B^{(B)}$ for some $\gamma \neq 0$, and (3) $\boldsymbol{\theta}_A^{(A)} = \mathbf{0}$ or $\boldsymbol{\theta}_B^{(A)} = \mathbf{0}$, and $\boldsymbol{\theta}_B^{(B)} = \mathbf{0}$ or $\boldsymbol{\theta}_B^{(B)} = \mathbf{0}$. The former two solutions result in trivial features and will increase the classification error. The latter solution contains four possible solutions, out of which two solutions result in trivial features. Solutions resulting in trivial features are unlikely to occur during optimization due to a large classification error. Therefore, we need to consider only the remaining two solutions.

The possible solutions are: (1) $\boldsymbol{\theta}_A^{(A)} = \mathbf{0}, \boldsymbol{\theta}_B^{(B)} = \mathbf{0}$, and (2) $\boldsymbol{\theta}_B^{(A)} = \mathbf{0}, \boldsymbol{\theta}_A^{(B)} = \mathbf{0}$. Intuitively, in the former solution, $A$ and $B$ will be predicted using $X_B$ and $X_A$ respectively, and the latter solution corresponds to a robust feature extractor that minimizes the reducible error in Eq. (9). We will compare the predictive error achieved by these solutions to compare their likelihood during training.

Recall the expression for training error in predicting $A$ from Eq. (12).

$$\begin{aligned}
J_A(\boldsymbol{\Theta}^{(A)}, \boldsymbol{c}^{(A)}) &= (1-\beta)\left((1 - w_A\psi_{A1} - w_B w_{AB}\psi_{A2})^2 \rho_A^2 + \psi_{A1}^2\rho_{\boldsymbol{U}_A}^2 + \psi_{A2}^2\rho_{\boldsymbol{U}_B}^2\right) \\
&\quad + \beta\left((1 - w_A\psi_{A1})^2\rho_A^2 + w_B^2\psi_{A2}^2\rho_{B'}^2 + \psi_{A1}^2\rho_{\boldsymbol{U}_A}^2 + \psi_{A2}^2\rho_{\boldsymbol{U}_B}^2\right) \\
&= (1-\beta)\left((1 - w_A\psi_{A1} - w_B w_{AB}\psi_{A2})^2\rho_A^2\right) \\
&\quad + \beta\left((1 - w_A\psi_{A1})^2\rho_A^2 + w_B^2\psi_{A2}^2\rho_{B'}^2\right) + \psi_{A1}^2\rho_{\boldsymbol{U}_A}^2 + \psi_{A2}^2\rho_{\boldsymbol{U}_B}^2
\end{aligned}$$

We use $\psi_{A1}$ and $\psi_{A2}$ instead of $\psi_1$ and $\psi_2$ respectively to denote the parameters for predicting $A$. A similar expression can be written for the error in predicting $B$ with $\psi_{B1}$ and $\psi_{B2}$ denoting the parameters for predicting $B$.

$$\begin{aligned}
J_B(\boldsymbol{\Theta}^{(B)}, \boldsymbol{c}^{(B)}) &= (1-\beta)\left((1 - w_A\psi_{B1} - w_B w_{AB}\psi_{B2})^2\rho_A^2 + \psi_{B1}^2\rho_{\boldsymbol{U}_A}^2 + \psi_{B2}^2\rho_{\boldsymbol{U}_B}^2\right) \\
&\quad + \beta\left(w_A^2\psi_{B1}^2\rho_A^2 + (1 - w_B\psi_{B2})^2\rho_{B'}^2 + \psi_{B1}^2\rho_{\boldsymbol{U}_A}^2 + \psi_{B2}^2\rho_{\boldsymbol{U}_B}^2\right) \\
&= (1-\beta)\left((1 - w_A\psi_{B1} - w_B w_{AB}\psi_{B2})^2\rho_A^2\right) \\
&\quad + \beta\left(w_A^2\psi_{B1}^2\rho_A^2 + (1 - w_B\psi_{B2})^2\rho_{B'}^2\right) + \psi_{B1}^2\rho_{\boldsymbol{U}_A}^2 + \psi_{B2}^2\rho_{\boldsymbol{U}_B}^2
\end{aligned}$$

**Case 1: When $\boldsymbol{\theta}_A^{(A)} = \mathbf{0}, \boldsymbol{\theta}_B^{(B)} = \mathbf{0}$:** In this case, $\psi_{A1} = 0$ and $\psi_{B2} = 0$. Therefore, the predictive error during training for each latent variable can be written as,

$$J_A = (1-\beta)(w_B w_{AB}\psi_{A2} - 1)^2\rho_A^2 + \beta\rho_A^2 + \beta w_B^2\psi_{A2}^2\rho_{B'}^2 + \psi_{A2}^2\rho_{\boldsymbol{U}_B}^2$$
$$J_B = (1-\beta)(w_A\psi_{B1} - w_{AB})^2\rho_A^2 + \beta w_A^2\psi_{B1}^2\rho_A^2 + \beta\rho_{B'}^2 + \psi_{B1}^2\rho_{\boldsymbol{U}_A}^2$$

The optimal values of $\psi_{A2}$ and $\psi_{B1}$ can be obtained by equating the gradients of $R_A$ and $R_B$ to zero.

$$\frac{\partial J_A}{\partial \psi_{A2}} = 2(1-\beta)w_B w_{AB}(w_B w_{AB}\psi_{A2} - 1)\rho_A^2 + 2\beta w_B^2\psi_{A2}\rho_{B'}^2 + 2\psi_{A2}\rho_{\boldsymbol{U}_B}^2 = 0$$

$$\therefore \psi_{A2}^* = \frac{(1-\beta)w_B w_{AB}\rho_A^2}{(1-\beta)w_B^2 w_{AB}^2\rho_A^2 + \beta w_B^2\rho_{B'}^2 + \rho_{\boldsymbol{U}_B}^2}$$

$$J_A^* = \frac{(1-\beta)\rho_A^2\left(\beta w_B^2\rho_{B'}^2 + \rho_{\boldsymbol{U}_B}^2\right)}{(1-\beta)w_B^2 w_{AB}^2\rho_A^2 + \beta w_B^2\rho_{B'}^2 + \rho_{\boldsymbol{U}_B}^2} + \beta\rho_A^2$$

$$\frac{\partial J_B}{\partial \psi_{B1}} = 2(1-\beta)w_A(w_A\psi_{B1} - w_{AB})\rho_A^2 + 2\beta w_A^2\psi_{B1}\rho_A^2 + 2\psi_{B1}\rho_{\boldsymbol{U}_A}^2 = 0$$

$$\therefore \psi_{B1}^* = \frac{(1-\beta)w_A w_{AB}\rho_A^2}{w_A^2\rho_A^2 + \rho_{\boldsymbol{U}_A}^2}$$

$$J_B^* = \frac{(1-\beta)w_{AB}^2\rho_A^2(\beta w_A^2\rho_A^2 + \rho_{\boldsymbol{U}_A}^2)}{w_A^2\rho_A^2 + \rho_{\boldsymbol{U}_A}^2} + \beta\rho_{B'}^2$$

The combined training error for this solution is,

$$
\begin{aligned}
J_1^* &= J_A^* + J_B^* \\
&= \frac{(1-\beta)\rho_A^2 \left(\beta w_B^2 \rho_{B'}^2 + \rho_{U_B}^2\right)}{(1-\beta)w_B^2 w_{AB}^2 \rho_A^2 + \beta w_B^2 \rho_{B'}^2 + \rho_{U_B}^2} + \beta\rho_A^2 \\
&\quad + \frac{(1-\beta)w_{AB}^2 \rho_A^2 (\beta w_A^2 \rho_A^2 + \rho_{U_A}^2)}{w_A^2 \rho_A^2 + \rho_{U_A}^2} + \beta\rho_{B'}^2
\end{aligned}
\tag{14}
$$

**Case 2: When $\boldsymbol{\theta}_B^{(A)} = \mathbf{0}, \boldsymbol{\theta}_A^{(B)} = \mathbf{0}$:** Here, $\psi_{A2} = 0$ and $\psi_{B1} = 0$. The predictive error during training for each latent variable can be written as,

$$
\begin{aligned}
J_A &= (w_A \psi_{A1} - 1)^2 \rho_A^2 + \psi_{A1}^2 \rho_{U_A}^2 \\
J_B &= \left((1-\beta)w_{AB}^2 \rho_A^2 + \beta\rho_{B'}^2\right)(w_B \psi_{B2} - 1)^2 + \psi_{B2}^2 \rho_{U_B}^2
\end{aligned}
$$

We follow the former procedure to estimate the optimal values of $\psi_{A1}$ and $\psi_{B2}$.

$$
\frac{\partial J_A}{\partial \psi_{A1}} = 2w_A (w_A \psi_{A1} - 1)\rho_A^2 + 2\psi_{A1}\rho_{U_A}^2 = 0
$$

$$
\therefore \psi_{A1}^* = \frac{w_A \rho_A^2}{w_A^2 \rho_A^2 + \rho_{U_A}^2}
$$

$$
J_A^* = \frac{\rho_A^2 \rho_{U_A}^2}{w_A^2 \rho_A^2 + \rho_{U_A}^2}
$$

$$
\frac{\partial J_B}{\partial \psi_{B2}} = 2w_B \left((1-\beta)w_{AB}^2 \rho_A^2 + \beta\rho_{B'}^2\right)(w_B \psi_{B2} - 1) + 2\psi_{B2}\rho_{U_B}^2
$$

$$
\therefore \psi_{B2}^* = \frac{(1-\beta)w_B w_{AB}^2 \rho_A^2 + \beta w_B \rho_{B'}^2}{(1-\beta)w_B^2 w_{AB}^2 \rho_A^2 + \beta w_B^2 \rho_{B'}^2 + \rho_{U_B}^2}
$$

$$
J_B^* = \frac{\left((1-\beta)w_{AB}^2 \rho_A^2 + \beta\rho_{B'}^2\right)\rho_{U_B}^2}{(1-\beta)w_B^2 w_{AB}^2 \rho_A^2 + \beta w_B^2 \rho_{B'}^2 + \rho_{U_B}^2}
$$

The combined training error for this solution is,

$$
\begin{aligned}
J_2^* &= J_A^* + J_B^* \\
&= \frac{\rho_A^2 \rho_{U_A}^2}{w_A^2 \rho_A^2 + \rho_{U_A}^2} + \frac{\left((1-\beta)w_{AB}^2 \rho_A^2 + \beta\rho_{B'}^2\right)\rho_{U_B}^2}{(1-\beta)w_B^2 w_{AB}^2 \rho_A^2 + \beta w_B^2 \rho_{B'}^2 + \rho_{U_B}^2}
\end{aligned}
\tag{15}
$$

Comparing $J_1^*$ and $J_2^*$,

$$
\begin{aligned}
J_1^* - J_2^* &= \frac{(1-\beta)\beta w_B^2 \rho_A^2 \rho_{B'}^2 + (1-\beta)\rho_A^2 \rho_{U_B}^2 - (1-\beta)w_{AB}^2 \rho_A^2 \rho_{U_B}^2 - \beta\rho_{B'}^2 \rho_{U_B}^2}{(1-\beta)w_B^2 w_{AB}^2 \rho_A^2 + \beta w_B^2 \rho_{B'}^2 + \rho_{U_B}^2} \\
&\quad + \frac{(1-\beta)\beta w_A^2 w_{AB}^2 \rho_A^4 + (1-\beta)w_{AB}^2 \rho_A^2 \rho_{U_A}^2 - \rho_A^2 \rho_{U_A}^2}{w_A^2 \rho_A^2 + \rho_{U_A}^2} + \beta(\rho_A^2 + \rho_{B'}^2)
\end{aligned}
$$

Simplifying the above expression, we get the condition that $J_1^* - J_2^* > 0$ if $\beta$ satisfies the following conditions: (1) $\beta \geq 1 - \frac{1}{|w_{AB}|}$, (2) $\beta \geq \min\left(\frac{\rho_A^2}{2\rho_{B'}^2 + \rho_A^2}, \frac{\rho_{U_A}^2}{w_A^2 w_{AB}^2 \rho_A^2}\right)$. The conditions imply that enforcing linear independence results in robust feature extractors when *enough* interventional data is available during training.

However, this is only a sufficient condition that strictly ensures $J_1^* - J_2^* > 0$. In practice, $\beta$ could be much lower, especially when the total loss is of the form $\mathcal{L}_{\text{total}} = \lambda_{\text{MSE}}\mathcal{L}_{\text{MSE}} + \lambda_{\text{dep}}\mathcal{L}_{\text{dep}}$, where $\lambda_{\text{MSE}}$ and $\lambda_{\text{dep}}$

are positive hyperparameters. We verify this empirically by randomly setting the parameters of the data generation process and plotting the predictive errors $J_1^*$ and $J_2^*$ for different values of $\beta$. We calculate $J_1^*$ and $J_2^*$ for 5000 runs (shown using thin curves) and plot the average error (shown using thick curves) in Fig. 13. We observe that the average value of $J_1^*$ is always higher than that of $J_2^*$ for all values of $\beta$. But, when $\beta \to 0$, their average values get closer to each other.

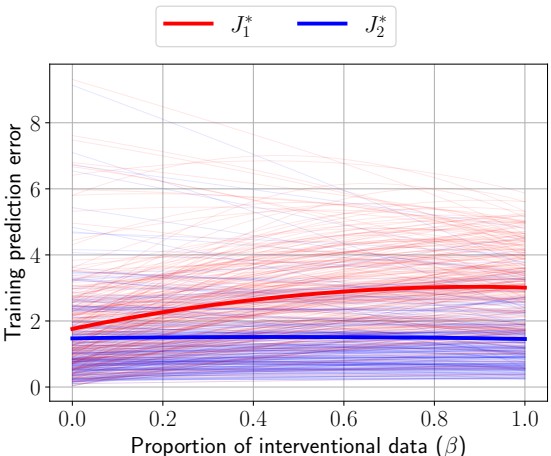

Figure 13: Comparing $J_1^*$ (Eq. (14)) and $J_2^*$ (Eq. (15)) as functions of $\beta$ for 5000 runs with randomly sampled data generation parameters. We show individual runs using thin curves and the average error values using thick curves. We only show the errors from a few randomly sampled runs for visual clarity. We observe that the average value of $J_1^*$ (shown using thick red curve) is always higher than that of $J_2^*$ (shown using thick blue curve), indicating that enforcing linear independence between interventional features is more likely to obtain robust feature extractors than degenerate solutions.

### B.4 Additional Analysis on How RepLIn Improves Interventional Robustness

In the previous section, we demonstrated, using a linear 2-variable causal model, that enforcing independence can provably improve statistical test-time risk over the interventional distribution for a sufficient proportion of interventional data in the training distribution. In this section, we will further show *how* enforcing dependence improves test-time risk over interventional distribution. We will limit this analysis to linear models, but extend it to include multiple latent variables (including exogenous noise) with possibly nonlinear causal relations between them and imperfect interventions.

Similar to our former setups, the observable data $\boldsymbol{X}$ is a function of the latent variables of interest, $\boldsymbol{Z}$, and exogenous noise variables $\boldsymbol{U}$, $\boldsymbol{X} = g_{\boldsymbol{X}}(\boldsymbol{Z}, \boldsymbol{U})$. Here, $\boldsymbol{Z}$ can be considered a concatenation of individual latent variables, such as $A$ and $B$ that we used in our previous analysis. $g_{\boldsymbol{X}}$ could be a nonlinear function. Although a linear model to predict $\boldsymbol{Z}$ from $\boldsymbol{X}$ may be insufficient for a nonlinear $g_{\boldsymbol{X}}$, the following analysis is still valid. Consider the task of predicting one of the latent variable elements in $\boldsymbol{Z}$, $Z_1$, using linear weights $\boldsymbol{w}_1$ as $\hat{Z}_1 = \boldsymbol{w}_1^\top \boldsymbol{X}$.

To learn this predictor, we have access to a training distribution $P_{\text{train}}$, which is a mixture of the observational distribution $P_{\text{obs}}$ and the interventional distribution $P_{\text{int}}$ as follows:

$$P_{\text{train}}(\boldsymbol{X}, \boldsymbol{Z}) = (1 - \beta)P_{\text{obs}}(\boldsymbol{X}, \boldsymbol{Z}) + \beta P_{\text{int}}(\boldsymbol{X}, \boldsymbol{Z})$$

During observations, some latent variables are causally related to each other. For this analysis, let $Z_1$ be a parent of multiple child variables during observation. Here also, the causal relation from $Z_1$ to its child nodes may be nonlinear. These child variables are also affected by external noise variables. Similar to our previous setups, during interventions, we intervene on one or more of these child nodes, rendering them independent of $Z_1$. To understand how enforcing independence between the predictors for $Z_1$ and its child nodes over interventional distribution improves robustness, we will compare the weights $\boldsymbol{w}_1$ obtained through vanilla risk

minimization (minimizing only the prediction error), denoted by $\boldsymbol{w}_{\mathrm{vrm},1}$, and the weights obtained through the proposed approach, denoted by $\boldsymbol{w}_{\mathrm{dep},1}$ against the weights of a robust linear predictor, denoted by $\boldsymbol{w}_{\mathrm{rob},1}$.

**Robust linear predictor**: The weights for a linear predictor that is robust against interventional distribution shifts can be obtained by minimizing the prediction error over a hypothetical training distribution that consists of only interventional data. Here, $P_{\mathrm{int}}$ is the interventional distribution where all child nodes of $Z_1$ are intervened on.

$$\boldsymbol{w}_{\mathrm{rob},1}^* = \underset{\boldsymbol{w}_{\mathrm{rob},1}}{\operatorname{argmin}} \, \mathbb{E}_{P_{\mathrm{int}}} \left[ \left( Z_1 - \boldsymbol{w}_{\mathrm{rob},1}^\top \boldsymbol{X} \right)^2 \right]$$

The closed-form solution to the above equation, under assumptions of zero-mean latent variables and a mean-preserving mixing function $g_{\boldsymbol{X}}$, is

$$\boldsymbol{w}_{\mathrm{rob},1}^* = \boldsymbol{C}_{\boldsymbol{X}_{\mathrm{int}}}^{-1} \boldsymbol{c}_{\boldsymbol{Z}_1 \boldsymbol{X}_{\mathrm{int}}}, \tag{16}$$

where $\boldsymbol{C}_{\boldsymbol{X}_{\mathrm{int}}} = \mathbb{E}_{P_{\mathrm{int}}} \left[ \boldsymbol{X} \boldsymbol{X}^\top \right]$ is the covariance matrix of the input data $\boldsymbol{X}$ over the interventional distribution, and $\boldsymbol{c}_{\boldsymbol{Z}_1 \boldsymbol{X}_{\mathrm{int}}} = \mathbb{E}_{P_{\mathrm{int}}} \left[ \boldsymbol{X} Z_1 \right]$ is the cross-covariance vector between $\boldsymbol{X}$ and $Z_1$ during intervention. Note that $\boldsymbol{w}_{\mathrm{rob},1}^*$ are the weights of a robust linear predictor, irrespective of whether the mixing function $g_{\boldsymbol{X}}$ is linear or not.

**Optimal linear predictor under vanilla risk minimization**: The optimal linear predictor for $Z_1$ under vanilla risk minimization can be obtained by minimizing the prediction error for $Z_1$ under the training distribution. The weights of this optimal predictor will appear similar to Eq. (16), except the involving terms will be computed over the training distribution. The optimal weights are

$$\boldsymbol{w}_{\mathrm{vrm},1}^* = \boldsymbol{C}_{\boldsymbol{X}_{\mathrm{train}}}^{-1} \boldsymbol{c}_{\boldsymbol{Z}_1 \boldsymbol{X}_{\mathrm{train}}}. \tag{17}$$

$\boldsymbol{C}_{\boldsymbol{X}_{\mathrm{train}}} = \mathbb{E}_{P_{\mathrm{train}}} \left[ \boldsymbol{X} \boldsymbol{X}^\top \right]$ and $\boldsymbol{c}_{\boldsymbol{Z}_1 \boldsymbol{X}_{\mathrm{train}}} = \mathbb{E}_{P_{\mathrm{train}}} [\boldsymbol{X} Z_1]$, similar to the corresponding quantities defined earlier for interventional distribution. Due to the discrepancy between training and intervention distributions, the optimal linear predictor under the training distribution will have excess risk compared to the robust linear predictor in Eq. (16). This excess risk can be quantified as

$$\begin{aligned} e_{\mathrm{excess}}(\boldsymbol{w}_{\mathrm{vrm},1}^*) &= \mathbb{E}_{P_{\mathrm{int}}} \left[ \left( \boldsymbol{w}_{\mathrm{vrm},1}^{*\top} \boldsymbol{X} - \boldsymbol{w}_{\mathrm{rob},1}^{*\top} \boldsymbol{X} \right)^2 \right] \\ &= \mathbb{E}_{P_{\mathrm{int}}} \left[ \left( \boldsymbol{w}_{\mathrm{vrm},1}^* - \boldsymbol{w}_{\mathrm{rob},1}^* \right)^\top \boldsymbol{X} \boldsymbol{X}^\top \left( \boldsymbol{w}_{\mathrm{vrm},1}^* - \boldsymbol{w}_{\mathrm{rob},1}^* \right) \right] \\ &= \left( \boldsymbol{w}_{\mathrm{vrm},1}^* - \boldsymbol{w}_{\mathrm{rob},1}^* \right)^\top \boldsymbol{C}_{\boldsymbol{X}_{\mathrm{int}}} \left( \boldsymbol{w}_{\mathrm{vrm},1}^* - \boldsymbol{w}_{\mathrm{rob},1}^* \right) = \left\| \boldsymbol{w}_{\mathrm{vrm},1}^* - \boldsymbol{w}_{\mathrm{rob},1}^* \right\|_{\boldsymbol{C}_{\boldsymbol{X}_{\mathrm{int}}}}^2 \end{aligned} \tag{18}$$

**How does enforcing independence over interventional distribution help?** For this analysis, we will use the simplified version of RepLIn that we used for our analysis in the previous section, which consists of only the dependence loss. Specifically, we will minimize the squared covariance between the predictors for $Z_1$ and its parent nodes over the interventional distribution when we intervene on $Z_1$. Recollect that the latent variables had zero mean, and the mixing function was mean-preserving. Combining this dependence loss with the prediction loss from vanilla risk minimization, our training objective becomes the following.

$$\begin{aligned} J\left( \boldsymbol{w}_1, \ldots, \boldsymbol{w}_{d_Z} \right) &= \mathbb{E}_{P_{\mathrm{train}}} \left[ \sum_i \left( Z_i - \boldsymbol{w}_i^\top \boldsymbol{X} \right)^2 \right] + \sum_{j \in \mathcal{C}\mathrm{h}(1)} \mathbb{E}_{P_{\mathrm{int}}}^2 \left[ \boldsymbol{w}_1^\top \boldsymbol{X} \cdot \boldsymbol{w}_j^\top \boldsymbol{X} \right] \\ &= \underbrace{\sum_i c_{z_i}^2 + \boldsymbol{w}_i^\top \boldsymbol{C}_{\boldsymbol{X}_{\mathrm{train}}} \boldsymbol{w}_i - 2 \boldsymbol{w}_i^\top \boldsymbol{c}_{\boldsymbol{Z}_i \boldsymbol{X}_{\mathrm{train}}}}_{\text{prediction loss}} + \underbrace{\sum_{j \in \mathcal{P}\mathrm{a}(1)} \left( \boldsymbol{w}_1^\top \boldsymbol{C}_{\boldsymbol{X}_{\mathrm{int}}} \boldsymbol{w}_j \right)^2}_{\text{dependence loss}} \end{aligned}$$

where $\mathcal{P}\mathrm{a}(1)$ is the set of $Z_1$'s parent nodes' indices. The dependence loss in the above equation is equivalent to HSIC without any additional nonlinear feature extractors over the predictors. Computing the gradient of

$J\left(\boldsymbol{w}_1, \ldots, \boldsymbol{w}_{d_Z}\right)$ w.r.t. $\boldsymbol{w}_1$ and equating it to the zero vector gives the following:

$$\frac{\partial J}{\partial \boldsymbol{w}_1} = 2\boldsymbol{C}_{\boldsymbol{X}_{\text{train}}}\boldsymbol{w}_1 + 2\sum_{j \in \mathcal{C}\text{h}(1)} \boldsymbol{w}_1^\top \boldsymbol{C}_{\boldsymbol{X}_{\text{int}}}\boldsymbol{w}_j \cdot \boldsymbol{C}_{\boldsymbol{X}_{\text{int}}}\boldsymbol{w}_j - 2\boldsymbol{c}_{\boldsymbol{Z}_1\boldsymbol{X}_{\text{train}}} = 0$$

$$\implies \boldsymbol{C}_{\boldsymbol{X}_{\text{train}}}\boldsymbol{w}_1 + \sum_{j \in \mathcal{C}\text{h}(1)} \boldsymbol{w}_1^\top \boldsymbol{C}_{\boldsymbol{X}_{\text{int}}}\boldsymbol{w}_j \cdot \boldsymbol{C}_{\boldsymbol{X}_{\text{int}}}\boldsymbol{w}_j = \boldsymbol{c}_{\boldsymbol{Z}_1\boldsymbol{X}_{\text{train}}}$$

$$\implies \boldsymbol{w}_1 + \underbrace{\sum_{j \in \mathcal{C}\text{h}(1)} \boldsymbol{w}_1^\top \boldsymbol{C}_{\boldsymbol{X}_{\text{int}}}\boldsymbol{w}_j \cdot \boldsymbol{C}_{\boldsymbol{X}_{\text{train}}}^{-1}\boldsymbol{C}_{\boldsymbol{X}_{\text{int}}}\boldsymbol{w}_j}_{\boldsymbol{w}_{\text{corr},1} := \text{ correction vector for } \boldsymbol{w}_1} = \underbrace{\boldsymbol{C}_{\boldsymbol{X}_{\text{train}}}^{-1}\boldsymbol{c}_{\boldsymbol{Z}_1\boldsymbol{X}_{\text{train}}}}_{=\boldsymbol{w}_{\text{vrm},1}^*} \tag{19}$$

Note that the RHS of Eq. (19) is $\boldsymbol{w}_{\text{vrm},1}^*$ from Eq. (17). Thus, we obtain $\boldsymbol{w}_1 = \boldsymbol{w}_{\text{vrm},1}^* - \boldsymbol{w}_{\text{corr},1}$. This means that enforcing independence between $Z_1$ and its child nodes over the interventional distribution essentially adds a *correction vector* to the optimal predictor under vanilla risk minimization.

**Can we obtain an analytical solution for $\boldsymbol{w}_{\text{dep},.}$?**: We may write similar equations as Eq. (19) for the predictors of other latent variables in $\boldsymbol{Z}$. In a 2-variable case, these equations would be those of a hyperboloid, implying infinitely many solutions. A unique solution may be arrived at, although not analytically, through additional commonplace regularization such as $L_2$ regularization or SGD's implicit regularization, resulting in a minimum norm solution that satisfies Eq. (19). Even with heuristics such as selecting a minimum-norm solution, it is not easy to obtain an analytical solution for $\boldsymbol{w}_{\text{dep},.}$ since the correction vectors are interdependent on the weights of the predictors for other latent variables.

Let $\boldsymbol{w}_{\text{dep},1}^*$ be a solution to Eq. (19). The excess risk for $\boldsymbol{w}_{\text{dep},1}^*$ can be written similar to Eq. (18) as

$$e_{\text{excess}}(\boldsymbol{w}_{\text{dep},1}^*) = \left\|\boldsymbol{w}_{\text{dep},1}^* - \boldsymbol{w}_{\text{rob},1}^*\right\|_{\boldsymbol{C}_{\boldsymbol{X}_{\text{int}}}}^2 \tag{20}$$

We can conclude that enforcing independence between predictors over interventional distribution improves robustness if the excess risk for $\boldsymbol{w}_{\text{dep},1}^*$ is lower than that for $\boldsymbol{w}_{\text{vrm},1}^*$, i.e., if $e_{\text{excess}}(\boldsymbol{w}_{\text{dep},1}^*) < e_{\text{excess}}(\boldsymbol{w}_{\text{vrm},1}^*)$. In the next part, we will empirically observe the effects of imperfect intervention and the amount of interventional data in the training distribution on the utility of RepLIn.

**Empirical Analysis of $\boldsymbol{w}_{\text{corr},1}$**: The empirical analysis in this section will focus on two aspects: (1) the effect of imperfect interventions, and (2) how the correcting vector acts on $\boldsymbol{w}_{\text{dep},1}$. To answer both questions, we construct a 2-latent variable toy dataset. The latent variables are $Z_1 \sim \mathcal{N}(0, \sigma_1^2)$ and $Z_2 \sim \frac{\sigma_2}{\sigma_1}Z_1 + \epsilon$, where $\sigma_1, \sigma_2 \sim \mathcal{U}(0, 5)$ and $\epsilon \sim \mathcal{N}(0, 10^{-4})$. The observable data $\boldsymbol{X}$ is then constructed as $\boldsymbol{X} = \boldsymbol{W}\begin{bmatrix} Z_1 \\ Z_2 \\ U \end{bmatrix}$, where $\boldsymbol{U} \sim \mathcal{N}(0, 10^{-2})$ is the exogenous noise variable and $\boldsymbol{W} \in \mathbb{R}^{d_X \times 3}$ is a randomly generated orthogonal matrix that acts as the linear mixing function. $d_X$ is also chosen randomly from $\{3, \ldots, 20\}$.

We model the imperfect intervention using an imperfectness hyperparameter $\eta$ by essentially replacing each interventional sample in the training distribution with an observational sample with $\eta$ probability. During training, we minimize dependence between predictors over this imperfect intervention. As $\eta$ increases, the proportion of causally related latent variables masquerading as independent variables increases, and enforcing independence between predictors over this imperfect intervention can then hurt the predictive performance. Our intuition stems from the boundary case of $\eta \to 1$ (all interventional samples replaced with observational samples), where only random predictors can satisfy the independence condition that we aim to achieve.

Fig. 14 shows the results of RepLIn on imperfect interventions. In Fig. 14a, we compare the test error for RepLIn for various values of $\eta$ against the test error of a vanilla model (shown in black) trained on a dataset with observational and perfect-interventional data ($\eta = 0$). We can see that the test errors of RepLIn are either lower or equal (near the boundary values of $\beta$) compared to the vanilla predictor when $\eta \to 0$. The errors match as $\beta \to 0$ and $\beta \to 1$ due to the unavailability and abundance of interventional data, respectively. As the interventions become more imperfect, the test errors for RepLIn increase. In Fig. 14b, we view the difference between the excess risks of vanilla predictors and RepLIn predictors for various values

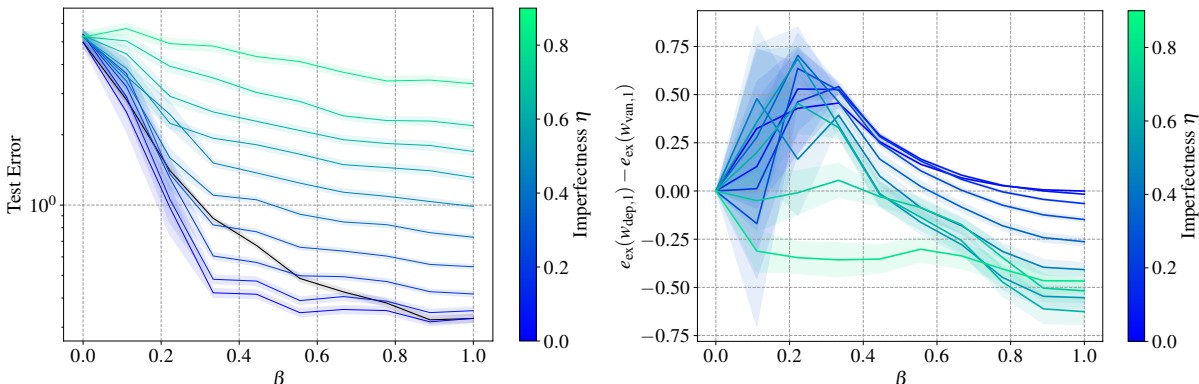

(a) Test error for RepLIn for various values of $\eta$ compared to a vanilla predictor trained on a training distribution with $\eta = 0$.

(b) Difference between excess risks for a vanilla predictor and a RepLIn predictor for various values of $\eta$.

Figure 14: We examine the effect of imperfect intervention on RepLIn by (a) comparing the test error of RepLIn predictors trained on distributions with various imperfectness probability $\eta$ against a vanilla predictor trained on a distribution with observational and perfect interventional data ($\eta = 0$), and (b) comparing the errors for RepLIn and vanilla predictors trained on distributions with various values of $\eta$. The plots indicate that imperfect intervention can hurt RepLIn's performance, especially for higher values of $\beta$.

of $\eta$. Here, we note that RepLIn consistently outperforms the corresponding vanilla predictor for most values of $\eta$ at lower values of $\beta$. As $\eta$ increases, RepLIn begins to perform worse than vanilla predictors for higher values of $\beta$, and RepLIn eventually consistently underperforms vanilla predictors for $\eta \to 1$.

**Conclusion**: Since RepLIn relies on enforcing independence between samples where the underlying variables are truly independent, it is naturally prone to imperfect interventions, particularly for large values of $\beta$. However, we envisioned RepLIn for scenarios where interventional data is scarce ($\beta \ll 1$), and where targeted approaches to improve robustness are desirable. In this regime of $\beta \ll 1$, our results indicate RepLIn performs better than vanilla predictors even when intervention noise $\eta$ is considerable.

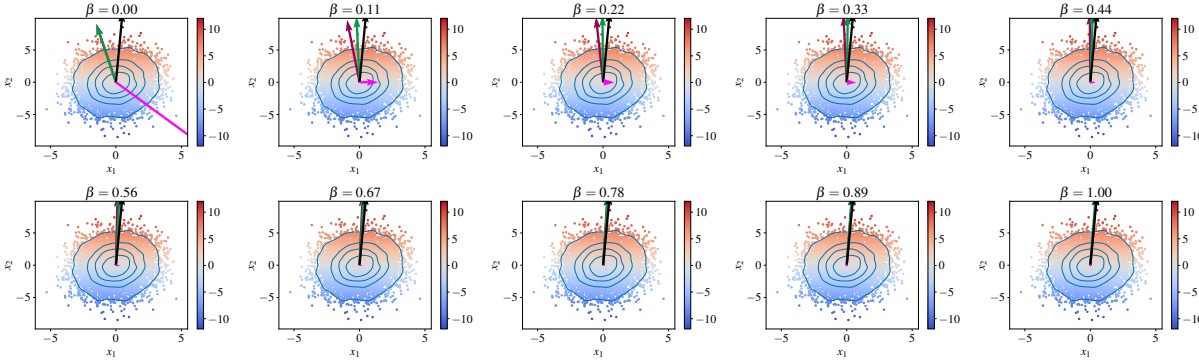

Figure 15: The weights of a linear RepLIn predictor are the sum of the weights of a vanilla predictor and a correction vector (Eq. (19)). In this plot, black, red, and green arrows show the weights of robust, vanilla, and RepLIn predictors. The negative correction vector is shown in magenta. We see that the correction vector points nearly orthogonal to the robust predictor (black), and its magnitude decreases as $\beta$ increases, when the vanilla predictor (red) approaches the robust predictor.

We repeat this experiment with $d_X = 2$ so that we can plot the resulting predictor weights, as well as the correction weights. In Fig. 15, we plot the weights of robust (black), vanilla (red), and RepLIn (green) predictors for various values of $\beta$ over the density plot of $\boldsymbol{X}$ samples. The observed samples from $\boldsymbol{X}$ are

also shown underneath the contours of density. The samples are colored based on their corresponding value of $Z_1$. Among the subplots, $\beta = 0$ corresponds to the case without any interventional data, where red and green arrows overlap. In the remaining scenarios, the RepLIn predictor (green) lies between robust (black) and vanilla (red) predictors. From Eq. (19), we know that $\boldsymbol{w}_{\text{dep},1}^*$ is the sum of the vanilla predictor weights $\boldsymbol{w}_{\text{vrm},1}^*$ and the negative of the correction vector $\boldsymbol{w}_{\text{corr},1}$. Therefore, we also plot the negative of the correction vector using magenta in Fig. 15. We can see that $\boldsymbol{w}_{\text{corr},1}$ is always nearly orthogonal to the robust predictor weights, and indeed acts as a correction vector that "pushes" $\boldsymbol{w}_{\text{vrm},1}^*$ towards $\boldsymbol{w}_{\text{rob},1}^*$. As $\beta$ increases, the vanilla predictor improves its performance, as we saw in Fig. 14, eventually matching the robust predictor. The magnitude of the correction vector also subsequently reduces as $\beta$ increases.

## C Review of identifiable causal representation learning

The primary objective of identifiable causal representation learning (ICRL) is to learn a representation such that it is possible to identify the latent factors (up to permutation and elementwise transformation) from the representation. These methods are commonly built upon autoencoder-based approaches and learn generative representations. The advantage of learning a causal representation is that the decoder then implicitly acts as the true underlying causal model, facilitating counterfactual evaluation and interpretable representations.

Locatello et al. (2019); Khemakhem et al. (2020) showed that disentangled representation learning was impossible without additional assumptions on both the model and the data. Some of the inductive biases that have been proposed since to learn disentangled representations include auxiliary labels (Hyvarinen & Morioka, 2016; Hyvarinen et al., 2019; Sorrenson et al., 2020; Khemakhem et al., 2020; Lu et al., 2022; Ahuja et al., 2022b; Kong et al., 2022), temporal data (Klindt et al., 2021; Yao et al., 2022; Song et al., 2023), and assumptions on the mixing function (Sorrenson et al., 2020; Yang et al., 2022; Lachapelle et al., 2022; Zheng et al., 2022; Moran et al., 2022).

**Use of interventional data:** Some works also use interventional data as weak supervision for identifiable representation learning (Lippe et al., 2022b; Brehmer et al., 2022; Ahuja et al., 2022a; 2023; Varıcı et al., 2023; von Kügelgen et al., 2023). Lippe et al. (2022b) learns identifiable representations from temporal sequences with possible interventions at any time step. Similar to our setting, they assume the knowledge of the intervention target. They also assume that the intervention on a latent variable at a time step does not affect other latent variables in the same time step. Lippe et al. (2023) relaxes the latter assumption as long as perfect interventions with known targets are available. Von Kügelgen et al. (2021); Zimmermann et al. (2021) showed that self-supervised learning with data augmentations allowed for identifiable representation learning. Brehmer et al. (2022) use pairs of data samples before and after some unknown intervention to learn latent causal models. Ahuja et al. (2022a) learns identifiable representations from sparse perturbations, with identifiability guarantees depending on the sparsity of these perturbations. Sparse perturbations can be treated as a parent class of interventions where the latent is intervened through an external action such as in reinforcement learning. Ahuja et al. (2022b) use interventional data for causal learning for polynomial mixing functions, under some assumptions on the nature of support for non-intervened variables. Varıcı et al. (2024a) relaxes the polynomial assumption on the mixing function and proves identifiability when two uncoupled hard interventions per node are available along with observational data. Varıcı et al. (2023) learn identifiable representations from data observed under different interventional distributions with the help of the score function during interventions. von Kügelgen et al. (2023) uses interventional data to learn identifiable representations up to nonlinear scaling. In addition to the above uses of interventional data, a few works (Saengkyongam & Silva, 2020; Saengkyongam et al., 2024; Zhang et al., 2023) have also attempted to predict the effect of unseen joint interventions with the help of observational and atomic interventions under various assumptions on the underlying causal model.

**Difference from our setting:** The general objective in ICRL is to "learn both the true joint distribution over both observed and latent variables" (Khemakhem et al., 2020). In contrast, the objective of our work is to learn representations corresponding to latent variables that are robust against interventional distributional shifts by leveraging *known* interventional independence relations. We pursue this objective in the hope that, as large models (Radford et al., 2021; Brown et al., 2020; Touvron et al., 2023; Dehghani et al., 2023) become more ubiquitous, efficient methods to improve these models with minimal amounts of

experimentally collected data will be of interest. Stated more formally, full identifiability of the underlying causal model is not in our interest, as robustness to interventional distribution shift can be achieved without full identifiability. For instance, consider the following setup: Let $A = [A_1, A_2]$ cause $B$ during observation. Here, $A_1$ is a binary variable (also, the class we are interested in predicting) and $A_2$ is a continuous variable from a Gaussian mixture with 2 modes. The mode is decided by $A_1$, and therefore informs the class. The observed data is $X = [X_{A_1} X_{A_2} X_B]$, where $X_{A_1}$ depends only on $A_1$, $X_{A_2}$ only on $A_2$, and $X_B$ only on $B$. Suppose the relations from the latent variables to the corresponding observed variables are such that it is possible to learn $A_1$ and $B$ from $X$, but not $A_2$ (say, due to noise or information loss in the mixing function). The discriminative task here is to predict which class $A$ belongs to. Here, RepLIn can learn robust representations that can fully predict the class of $A$ (through $A_1$), but is not fully identifiable since it does not have information about $A_2$.

Moreover, the representations learned by ICRL methods usually have permutation ambiguity. That is, the representations are disentangled but not mapped to the semantic attribute that we wish to predict in a downstream task. Our approach overcomes this ambiguity by explicitly learning attribute-specific representations.

## D  Differences w.r.t Domain Generalization/Out-of-Distribution Setting

The setting of RepLIn differs from domain generalization (DG) or out-of-distribution (OOD) tasks due to the assumptions in our setting, which RepLIn exploits to get more robustness benefits. Expressed in terms of the random variables $A$, $B$, and $X$ in our problem setting from Sec. 3, the task in DG is to predict some variable of interest $A$ from the observed data $X$ such that the learned model can generalize to unseen domains (Wang et al., 2022a; Ding et al., 2022b). To ensure robust prediction, multiple sets of training data that vary in their distributions of some variable $B$ are provided. DG does not typically have any requirements on the learned predictor, apart from its transferability to different domains. As a result, the learned predictor may store information about the environment to improve their predictions (Rosenfeld et al., 2022). In contrast, RepLIn removes information from intervened variables to improve their robustness of predictors that use RepLIn representations. In summary, RepLIn operates on a setting similar to DG/OOD, but under more assumptions, with the goal of obtaining more trustworthy representations. Tab. 7 shows the differences between the DG framework and ours. The first two rows show the differences in settings, while the last two rows show the differences in learned representations.

| Differences | DG/OOD | RepLIn |
|---|---|---|
| Relation from $A$ to $X$ between domains | May or may not change | Does not change |
| Is $A$ independent of $B$ in one or more domains? | May or may not be. It is also possible that $A$ is always independent of $B$. | $A \rightarrow B$ in observational data and $A$ independent of $B$ in interventional data. |
| Can accommodate more than one $B$? | No. $B$ is not of interest. | Yes. Learns representations for $B$ as well. |
| Is the representation learned for $A$ free of information from $B$? | Not necessarily. Some DG methods are designed to remove information from $B$, while others are not (e.g, DARE (Rosenfeld et al., 2022)) | Yes, the dependence loss ensures that the features for $A$ are free of information from $B$ in the interventional data. |

Table 7: Differences between the problem settings of domain generalization and RepLIn.

In addition to domain-level differences, training with group-imbalanced data also leads to models that suffer from group-bias during inference. In such cases, resampling the data according to the inverse sample frequency can improve generalization and robustness. Studies such as (Gulrajani & Lopez-Paz, 2021; Idrissi et al., 2022) have shown that ERM with resampling is effective against spurious correlations and is a strong baseline for domain generalization. Recent works such as dynamic importance reweighting (Fang et al., 2020),

SRDO (Shen et al., 2020), and MAPLE (Zhou et al., 2022) *learn to resample* using a separate validation set that acts as a proxy for the test set. However, learning such a resampling requires a large dataset of both *observational* and *interventional* data, which is often not practically feasible. In contrast, we will exploit known independence relations during interventions to improve robustness to interventional distributional shifts.

## E  Additional Results from Experiments

As mentioned in the main paper, our objective is to improve the robustness of representations against interventional distribution shifts. However, this robustness might come at the cost of observational accuracy since it removes spurious information that gives better performance on observational data. In this section, we report the results of the baselines and our methods on WINDMILL, CelebA, and CivilComments datasets. The accuracies of various methods in WINDMILL, CelebA, and CivilComments datasets are given in Tabs. 8 to 10, respectively.

| Method | $\beta = 0.5$ | $\beta = 0.3$ | $\beta = 0.1$ | $\beta = 0.05$ | $\beta = 0.01$ |
|---|---|---|---|---|---|
| ERM | $93.85 \pm 1.84$ | $98.06 \pm 1.20$ | $99.70 \pm 0.08$ | $99.92 \pm 0.02$ | $99.98 \pm 0.01$ |
| ERM-Res. | $94.53 \pm 0.89$ | $94.13 \pm 1.19$ | $94.84 \pm 0.92$ | $94.56 \pm 0.71$ | $94.53 \pm 1.14$ |
| IRMv1 | $93.37 \pm 0.85$ | $93.59 \pm 0.32$ | $93.72 \pm 0.73$ | $92.52 \pm 0.35$ | $94.04 \pm 0.63$ |
| Fish | $95.54 \pm 0.42$ | $95.37 \pm 0.36$ | $95.42 \pm 0.59$ | $95.83 \pm 0.51$ | $96.28 \pm 1.12$ |
| GroupDRO | $82.02 \pm 2.00$ | $84.40 \pm 2.72$ | $85.35 \pm 2.35$ | $84.25 \pm 0.91$ | $92.28 \pm 1.11$ |
| SAGM | $94.77 \pm 0.62$ | $95.17 \pm 0.71$ | $94.13 \pm 1.68$ | $95.61 \pm 0.69$ | $94.04 \pm 1.98$ |
| DiWA | $94.64 \pm 0.96$ | $94.30 \pm 0.36$ | $94.57 \pm 0.64$ | $94.39 \pm 0.99$ | $94.24 \pm 0.59$ |
| TEP | $65.20 \pm 14.22$ | $66.94 \pm 3.78$ | $61.34 \pm 19.35$ | $63.02 \pm 15.59$ | $73.77 \pm 9.01$ |
| RepLIn | $95.16 \pm 0.53$ | $97.83 \pm 0.40$ | $99.24 \pm 0.37$ | $98.75 \pm 0.43$ | $99.10 \pm 0.47$ |
| RepLIn-Res. | $95.57 \pm 0.62$ | $95.77 \pm 0.68$ | $95.59 \pm 1.08$ | $95.90 \pm 0.35$ | $95.51 \pm 1.71$ |

(a) Observational

| Method | $\beta = 0.5$ | $\beta = 0.3$ | $\beta = 0.1$ | $\beta = 0.05$ | $\beta = 0.01$ |
|---|---|---|---|---|---|
| ERM | $76.87 \pm 1.08$ | $69.86 \pm 3.19$ | $62.78 \pm 1.77$ | $59.52 \pm 1.30$ | $60.15 \pm 3.12$ |
| ERM-Res. | $73.70 \pm 3.19$ | $71.19 \pm 3.23$ | $73.62 \pm 1.54$ | $71.03 \pm 2.83$ | $70.20 \pm 3.73$ |
| IRMv1 | $78.24 \pm 0.79$ | $74.83 \pm 1.74$ | $78.61 \pm 2.24$ | $76.28 \pm 1.87$ | $71.75 \pm 2.03$ |
| Fish | $77.23 \pm 2.24$ | $77.23 \pm 1.32$ | $78.24 \pm 2.09$ | $76.42 \pm 1.95$ | $73.92 \pm 2.53$ |
| GroupDRO | $80.10 \pm 1.66$ | $80.96 \pm 1.33$ | $80.35 \pm 1.01$ | $77.40 \pm 1.16$ | $71.86 \pm 1.60$ |
| SAGM | $76.43 \pm 2.37$ | $79.05 \pm 2.23$ | $76.96 \pm 4.36$ | $79.86 \pm 1.81$ | $72.81 \pm 3.10$ |
| DiWA | $76.61 \pm 2.15$ | $76.71 \pm 0.59$ | $76.09 \pm 0.69$ | $75.83 \pm 1.83$ | $73.39 \pm 1.31$ |
| TEP | $58.68 \pm 4.72$ | $60.42 \pm 1.30$ | $56.07 \pm 3.35$ | $58.52 \pm 4.36$ | $59.23 \pm 1.13$ |
| RepLIn | $87.94 \pm 1.46$ | $87.76 \pm 2.30$ | $83.23 \pm 2.67$ | $73.63 \pm 2.43$ | $67.52 \pm 2.30$ |
| RepLIn-Res. | $88.46 \pm 0.96$ | $88.05 \pm 1.04$ | $87.91 \pm 1.36$ | $86.38 \pm 0.85$ | $78.41 \pm 1.27$ |

(b) Interventional

Table 8: Accuracy of various methods used in Sec. 5.1.

| Method | $\beta = 0.5$ | $\beta = 0.4$ | $\beta = 0.3$ | $\beta = 0.2$ | $\beta = 0.1$ | $\beta = 0.05$ |
|---|---|---|---|---|---|---|
| ERM-Res. | $91.38 \pm 0.09$ | $91.52 \pm 0.06$ | $91.39 \pm 0.07$ | $90.89 \pm 0.10$ | $90.57 \pm 0.09$ | $91.82 \pm 0.14$ |
| RepLIn-Res. | $86.02 \pm 0.18$ | $86.35 \pm 0.24$ | $86.58 \pm 0.11$ | $86.94 \pm 0.36$ | $87.67 \pm 0.21$ | $89.83 \pm 0.11$ |

(a) Observational

| Method | $\beta = 0.5$ | $\beta = 0.4$ | $\beta = 0.3$ | $\beta = 0.2$ | $\beta = 0.1$ | $\beta = 0.05$ |
|---|---|---|---|---|---|---|
| ERM-Res. | $81.09 \pm 0.17$ | $80.56 \pm 0.23$ | $80.06 \pm 0.17$ | $79.08 \pm 0.16$ | $76.63 \pm 0.24$ | $73.42 \pm 0.27$ |
| RepLIn-Res. | $81.97 \pm 0.14$ | $81.94 \pm 0.17$ | $81.84 \pm 0.18$ | $80.65 \pm 0.22$ | $78.56 \pm 0.20$ | $75.77 \pm 0.05$ |

(b) Interventional

Table 9: Accuracy of various methods used in Sec. 5.2.

| Method | $\beta = 0.5$ | $\beta = 0.3$ | $\beta = 0.1$ | $\beta = 0.05$ | $\beta = 0.01$ |
|---|---|---|---|---|---|
| ERM-Res. | $81.26 \pm 0.12$ | $81.77 \pm 0.14$ | $79.78 \pm 0.08$ | $79.97 \pm 0.12$ | $79.13 \pm 0.09$ |
| RepLIn-Res. | $79.27 \pm 0.09$ | $80.16 \pm 0.12$ | $77.65 \pm 0.06$ | $77.84 \pm 0.12$ | $78.51 \pm 0.16$ |

(a) Observational

| Method | $\beta = 0.5$ | $\beta = 0.3$ | $\beta = 0.1$ | $\beta = 0.05$ | $\beta = 0.01$ |
|---|---|---|---|---|---|
| ERM-Res. | $74.51 \pm 0.07$ | $75.29 \pm 0.22$ | $72.03 \pm 0.18$ | $71.78 \pm 0.12$ | $69.80 \pm 0.45$ |
| RepLIn-Res. | $75.30 \pm 0.37$ | $75.81 \pm 0.31$ | $72.00 \pm 0.23$ | $71.70 \pm 0.14$ | $69.99 \pm 0.80$ |

(b) Interventional

Table 10: Accuracy of various methods used in Sec. 5.3.

## F  More GradCAM Visualization

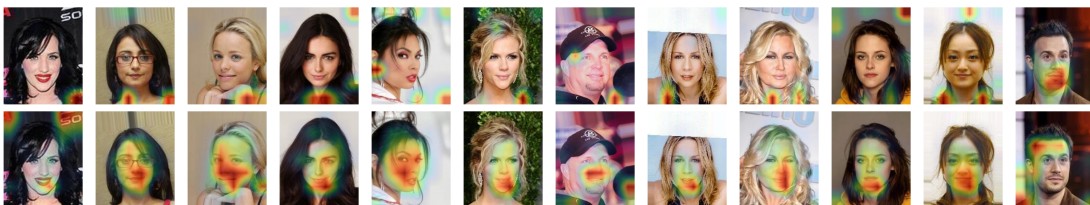

Figure 16: GradCAM visualizations for the samples that were correctly classified by ERM, but not by RepLIn. The top row shows the visualizations for ERM, while the bottom row shows that for RepLIn.

We show more GradCAM visualizations to illustrate the differences between the representations between ERM and RepLIn. In Sec. 6.1, we compared the GradCAM visualizations for those samples that were correctly classified by RepLIn, but incorrectly by ERM. Here, we visualize the GradCAM from samples that were correctly classified by ERM, but incorrectly by RepLIn. In Fig. 16, the top row shows the GradCAM visualizations of ERM, while the bottom row shows the visualizations for RepLIn. These samples are **chosen randomly**. Although these samples were incorrectly classified by RepLIn, the attention maps of RepLIn for most of the shown samples focus on the mouth region. On the other hand, attention maps of ERM do not focus on the mouth region as often.

# G   Visualization of Feature Distribution Learned on Windmill dataset

In this section, we compare the feature distributions learned by RepLIn on WINDMILL dataset against all the baselines from Sec. 5.1. The feature distributions are shown in Fig. 17.

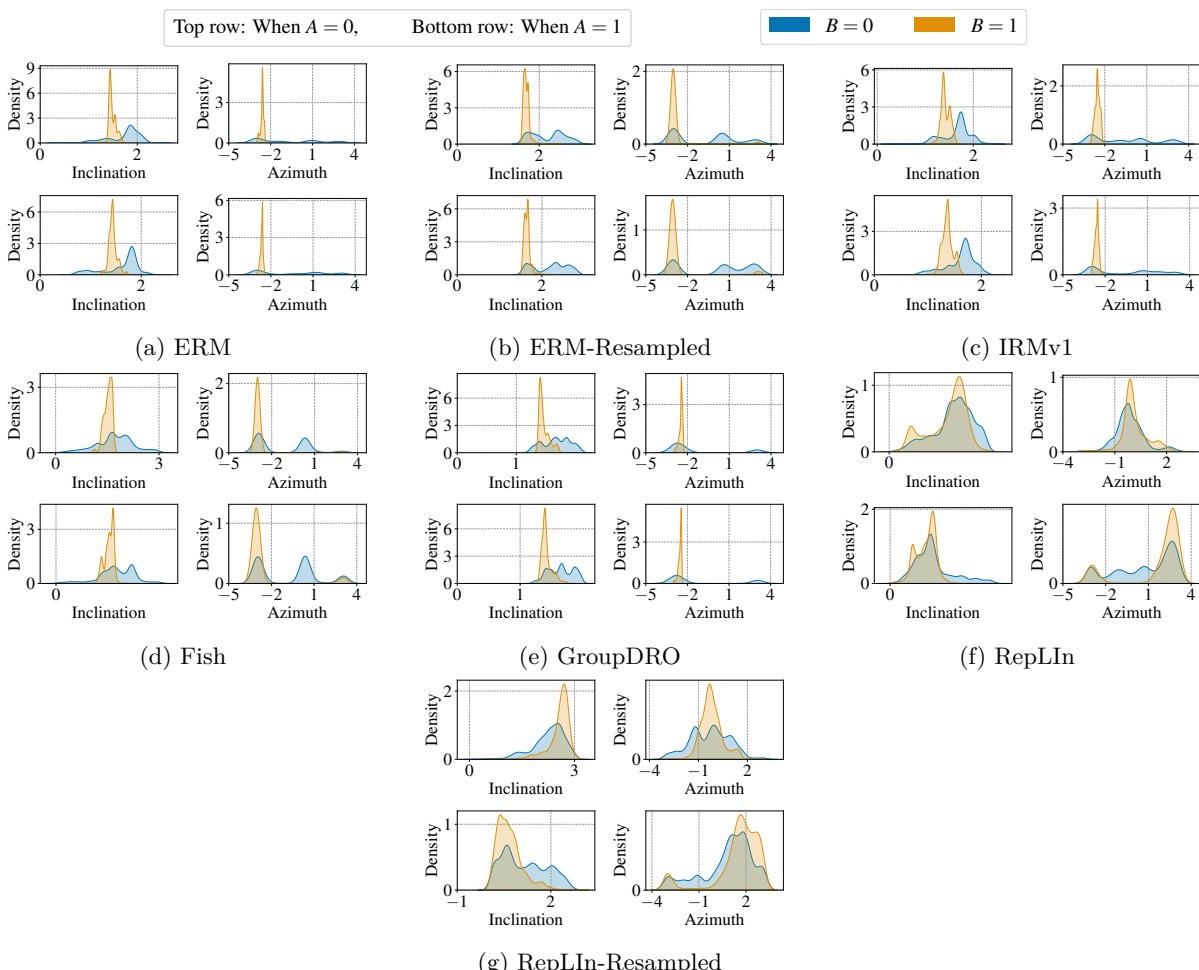

Figure 17: Visualization of interventional features learned by various methods on WINDMILL dataset.

# H   Balancing $\mathcal{L}_{\text{dep}}$ and $\mathcal{L}_{\text{self}}$ during training

The goal of RepLIn is to learn robust discriminative representations corresponding to variables of predictive interest such that each representation contains only the information from the latent variable it models, especially when these latent variables are causally related. This goal must be evaluated on two fronts – the

absolute utility of the representations for downstream tasks and performance equity between observational and interventional distributions. We quantified these evaluations in our experiments through the performance on interventional data and the relative accuracy drop between observational and interventional distributions. Our proposed loss functions also reflected these objectives: (1) self-dependence loss ($\mathcal{L}_{\text{self}}$) maximizes the information that a representation learns about its corresponding latent variable, and (2) dependence loss ($\mathcal{L}_{\text{dep}}$) minimizes the information shared by the representations of causally related latent variables during interventions, to obtain distributionally robust representations.

However, $\mathcal{L}_{\text{self}}$ and $\mathcal{L}_{\text{dep}}$ have somewhat conflicting objectives. Minimizing $\mathcal{L}_{\text{self}}$ maximizes the statistical information shared between latent variables and their corresponding representations. It does not discriminate the nature of this information and, thus, could include information about the child variables in the representation when minimized on observational data. Minimizing $\mathcal{L}_{\text{dep}}$ ensures that the interventional representations corresponding to independent variables do not share any information, regardless of whether these representations contain any discriminative information useful for predicting their corresponding latent variable. Thus, fundamentally, $\mathcal{L}_{\text{self}}$ enriches the information in the representations, while $\mathcal{L}_{\text{dep}}$ removes the information from the representations. If these loss functions are not balanced during training using their respective hyperparameters $\lambda_{\text{self}}$ and $\lambda_{\text{dep}}$, the learned representations may not be robust and discriminative.

We experimentally demonstrate the above statements with the help of a synthetic dataset with linear relations between variables, similar to the one used for theoretical analysis in Sec. 3.4.

**Experiment setup**: Our dataset consists of the high-dimensional observed signal $X \in \mathbb{R}^{100}$ from which we must predict two latent variables of interest, $A, B \in \mathbb{R}^{10}$. During observation, $A \to B$ in the underlying causal graph with the following linear causal relation between them.

$$
\begin{aligned}
A &\sim \mathcal{N}(0, I_{10}) && (I_p \text{ is } p \times p \text{ identity matrix}) \\
\epsilon &\sim \mathcal{N}(0, I_{10}) && (\text{Noise in observational relation}) \\
B &:= \sqrt{0.9}A + \sqrt{0.1}\epsilon
\end{aligned}
$$

To collect interventional data, we intervene on $B$ and set it to independently sampled $\tilde{B} \sim \mathcal{N}(0, I_{10})$. During intervention, $A \perp\!\!\!\perp \tilde{B}$. $A$ and $B$, along with exogenous random variable $U \sim \mathcal{N}(0, I_{80})$, create the observed signal $X$ from which we are tasked with learning representations corresponding to $A$ and $B$. Formally,

$$
\begin{aligned}
n &\sim \mathcal{N}(0, 0.25 I_{10}) && (\text{Noise in the mixing function}) \\
\hat{A} &= A + n \\
\hat{X} &= \begin{bmatrix} \hat{A} & B & U \end{bmatrix} && (21) \\
X &= W\tilde{X} + Z,
\end{aligned}
$$

where $W \in \mathbb{R}^{100 \times 100}$ and $Z \in \mathbb{R}^{100}$ are the linear coefficients of the mixing function whose entries are independently sampled from $\mathcal{N}(0, 1)$. During its sampling, we verify that $W$ is a full-rank matrix to ensure that a linear model can predict $A$ and $B$ from $X$. Note that the noise $n$ added to $A$ has a higher variance than the noise $\epsilon$ in the observational causal relation. This would prompt the model to learn *shortcut* (Geirhos et al., 2020) and rely on the information from $B$ to predict $A$. Since we know the variance of the noise added to $A$, we can also compute the statistical error of a robust linear model.

Our model consists of a linear layer each to learn the representations corresponding to $A$ and $B$, and a linear layer each to make the final predictions $\hat{A}$ and $\hat{B}$ from their respective representations. The model does not have any non-linear activation function. The models are trained by minimizing the mean squared error between their predictions and the ground truth, in addition to $\mathcal{L}_{\text{dep}}$ and $\mathcal{L}_{\text{self}}$ weighted by their corresponding hyperparameters $\lambda_{\text{dep}}$ and $\lambda_{\text{self}}$, respectively. Each batch comprises the entire training dataset. For each run, we first generate a different random seed $s_{\text{run}}$ that affects the sampled values for $W, Z, A,$ and $B$. Random values for $s_{\text{run}}$ are generated using a meta random seed $s_{\text{meta}}$ obtained from the system timestamp during the experiment run. We also use $s_{\text{meta}}$ to randomly sample $\lambda_{\text{dep}}$ and $\lambda_{\text{self}}$ from their uniform distributions in their log space. In total, 27,748 random hyperparameter settings were sampled.

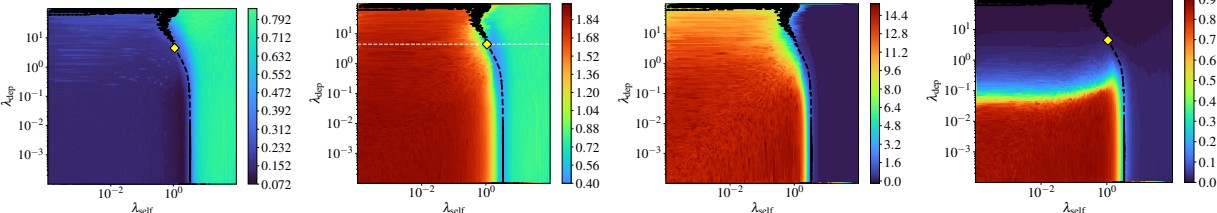

(a) Observational error in predicting $A$
(b) Interventional error in predicting $A$
(c) Relative increase in error
(d) Dependence between representations

Figure 18: Results of RepLIn models trained with different values for the hyperparameters $\lambda_{\text{dep}}$ and $\lambda_{\text{self}}$. The heatmaps show the variations of interventional accuracy (left) and relative drop in accuracy between observational and interventional distributions (right) with the hyperparameters.

The results of our experiments are shown in Fig. 18. In the results, we plot and analyze the prediction accuracy on $A$ since we intervened on $B$. To obtain continuous-valued plots, we interpolate between the sampled pairs of $\lambda_{\text{dep}}$ and $\lambda_{\text{self}}$ through triangulation. We make the following observations from the results:

**(1) Small values for $\lambda_{\text{dep}}$ and $\lambda_{\text{self}}$**: RepLIn behaves similarly to vanilla ERM method as $\lambda_{\text{dep}}, \lambda_{\text{self}} \to 0$. In Fig. 18, this setting corresponds to the lower-left quadrant of each plot. Due to the designed difficulty in predicting $A$ from $X$, the model uses information from $B$ to predict $A$, resulting in a low error in observational data (Fig. 18a) and a high error in interventional data (Fig. 18b). Statistical dependence between representations during interventions measured using NHSIC is also high (Fig. 18d), as expected.

**(2) Increasing $\lambda_{\text{dep}}$ alone**: When $\lambda_{\text{dep}}$ is increased without changing $\lambda_{\text{self}}$, dependence between representations of interventional data decreases, as expected. However, increasing $\lambda_{\text{dep}}$ sometimes provides only limited reductions in interventional error, as seen in Fig. 18c. For instance, increasing $\lambda_{\text{dep}}$ from $10^{-3}$ to 1, while keeping a constant $\lambda_{\text{self}} = 10^{-3}$ slightly decreased the error on interventional data from 1.89 to 1.76, while nominally increasing the error on observational data from 0.127 to 0.129. This shows that while minimizing interventional dependence helps learn robust representations against interventions, the benefits in performance may be marginal.

**(3) Increasing $\lambda_{\text{self}}$ alone**: Interestingly, increasing only $\lambda_{\text{self}}$ leads to a drop in interventional dependence and reduces the error disparity between observational and interventional data (Fig. 18c), even when $\lambda_{\text{dep}}$ is nearly zero. However, this decrease in performance disparity comes at the cost of higher observational error (left to right in Fig. 18a).

**(4) Lowest interventional error**: In Fig. 18b, we can observe a valley of relatively lower interventional error. The hyperparameter combination corresponding to the lowest interventional error occurs within this valley, marked with a yellow diamond. The same position is marked on other plots for ease of viewing. The lowest interventional error obtained experimentally was 0.4, considerably higher than the theoretical interventional error of 0.25 that a robust model would have attained. This indicates that the best hyperparameter combination did not result in a fully robust model. However, this is not surprising since our theoretical results in Sec. 3.4 suggested that a linear model cannot learn a fully robust model if the training dataset contains any observational data. Additionally, note that this hyperparameter combination did not result in the lowest performance disparity between the distributions and, instead, it appeared near a *phase change* in the loss values. To observe this phase change more clearly, we plot the loss values along the white dashed line in Fig. 18b, where we vary $\lambda_{\text{self}}$ and fix $\lambda_{\text{dep}}$ to the value it takes in the best hyperparameter combination (yellow diamond).

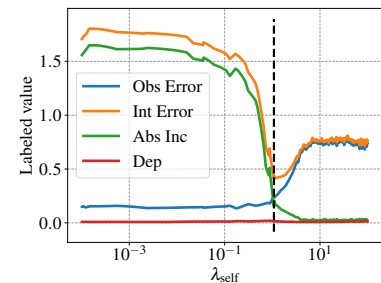

Figure 19: Change in observational and interventional error values for a fixed $\lambda_{\text{dep}}$ corresponding to the yellow diamond in Fig. 18b and varying $\lambda_{\text{self}}$.

In Fig. 19, we observe that as $\lambda_{\text{self}}$ increases, interventional error drops rapidly, achieving its minimum at $\lambda_{\text{dep}}$ corresponding to the yellow diamond (denoted by the dashed black line in Fig. 19), and then increases steadily to eventually saturate. Similarly, observational error gradually increases with increasing $\lambda_{\text{self}}$ initially and then displays a more rapid increase, eventually matching the interventional error at higher values of $\lambda_{\text{self}}$. Throughout these changes, the statistical dependence between the representations of interventional data remains nearly zero.

Our results indicate that, while both $\mathcal{L}_{\text{dep}}$ and $\mathcal{L}_{\text{self}}$ are needed to learn discriminative representations that are robust to interventional distribution shifts without losing their utility in downstream applications, hyperparameter tuning is still necessary to balance the effects of these loss functions.

### H.1 Why do the hyperparameters change between experiments?

In our main experiments, we chose different hyperparameters for different experiments. In this section, we explore the factors that affect the choice of hyperparameters between experiments. In particular, we focus on $\lambda_{\text{dep}}$, as $\mathcal{L}_{\text{dep}}$ is the primary loss function responsible for enforcing statistical independence between the interventional representations. We start by noting that robust representations are obtained by (at least partially) inverting the data-generating function from the latent variable to the observed signal. Therefore, we hypothesize that as the complexity of this data-generating function increases, $\lambda_{\text{dep}}$ and $\lambda_{\text{self}}$ generally increase. Here, we use the term "complexity" to roughly mean the minimum degree of a polynomial required to model the data-generating function. Informally, the more complex the data-generating function, the more hesitant the model is to learn robust representations (Geirhos et al., 2020).

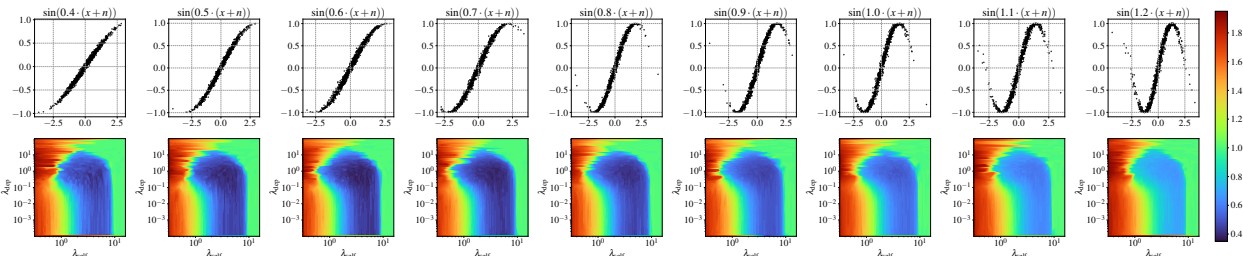

Figure 20: (*Top*) Sinusoidal transformation of a 1D Gaussian random variable $x$ with added noise $n \sim \mathcal{N}(0, 0.01)$, and (*bottom*) variation of interventional error for various values of $\lambda_{\text{dep}}$ and $\lambda_{\text{self}}$.

We now formally verify our hypothesis by adding a non-linear function in Eq. (21) of the simple dataset that we used to investigate the effect of $\lambda_{\text{dep}}$ and $\lambda_{\text{self}}$. We modify Eq. (21) as follows:

$$\hat{X} = \left[\sin\left(s \cdot \hat{A}\right) \quad B \quad U\right], \tag{22}$$

where $s$ controls the amount of non-linearity. A very low value for $s$ will result in a nearly linear function, as sin function is approximately linear near the origin. As $s$ increases, the non-linearity also increases. For higher values of $s$, multiple values of $\hat{A}$ will be mapped to the same value. For the remainder of this section, we will refer to $s$ as the "non-linearity factor." See Fig. 20 (top) on how the value of $s$ affects the sinusoidal transformation of a Gaussian random variable with added noise.

In addition to modifying the data generation process by using a non-linear relation from the latent variable to the observed signal, we also use non-linear models to learn the representations for each variable. Specifically, we use MLPs with 2 hidden layers and the ReLU activation function. We are interested in the variation in $\lambda_{\text{dep}}$ that gives us the minimum value of interventional error as the non-linearity factor $s$ changes. If our hypothesis is correct, then $\lambda_{\text{dep}}$ must increase as the non-linearity factor $s$ in Eq. (22) increases. Following the previous setup, we will sample several values for $\lambda_{\text{dep}}$ and $\lambda_{\text{self}}$, and train models for each combination of $\lambda_{\text{dep}}$ and $\lambda_{\text{self}}$. To save compute, we restrict the range of $\lambda_{\text{self}}$ to $[10^{-0.5}, 10^{1.2}]$ while sampling $\lambda_{\text{dep}}$ and $\lambda_{\text{self}}$, as the minimum interventional error usually lies in that range. Additionally, we utilize Bayesian optimization, employing the probability of improvement, to guide hyperparameter selection, thereby sampling more hyperparameters

from promising regions where the interventional error is typically low. The interpolated heatmap showing the interventional errors for various chosen hyperparameters is shown in Fig. 20 (bottom).

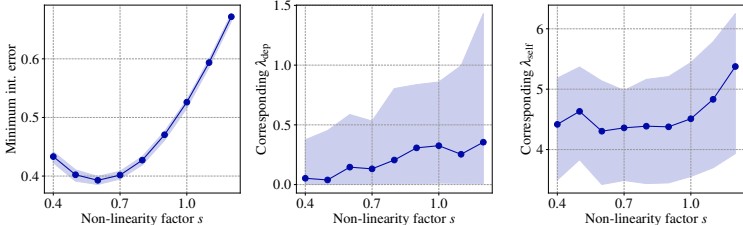

Figure 21: Variation in the minimum interventional error, and the corresponding $\lambda_{\text{dep}}$ and $\lambda_{\text{self}}$ when the non-linearity factor $s$ is increased. The results are for the bottom-5% percentile lowest interventional error values obtained over 4500 runs for each value of $s$. The median values are shown using dark lines, and the region between the first and third quartiles is shaded.

Fig. 21 shows three plots: bottom-5% percentile of interventional error, and corresponding $\lambda_{\text{dep}}$ and $\lambda_{\text{self}}$ values. For each of these values, we plot the median value using dark blue curves, and the region between the first and third quartiles is shaded in light blue. As expected, the minimum interventional error increases with the non-linearity factor $s$. A similar trend can be observed for $\lambda_{\text{dep}}$ and $\lambda_{\text{self}}$. Particularly, for $\lambda_{\text{dep}}$, the shaded region expands as $s$ increases, indicating that higher values of $\lambda_{\text{dep}}$ can now obtain very low values of interventional error.

## I  Generating Windmill Dataset

We provide the exact mathematical formulation of WINDMILL dataset described in Sec. 3.1. We define the following constants:

| Constants | Description | Default value |
|---|---|---|
| $n_{\text{arms}}$ | Number of "arms" in WINDMILL dataset | 4 |
| $r_{\text{max}}$ | Radius of the circular region spanned by the observed data | 2 |
| $\theta_{\text{wid}}$ | Angular width of each arm | $\frac{0.9\pi}{n_{\text{arms}}} = 0.7068$ |
| $\lambda_{\text{off}}$ | Offset wavelength. Determines the complexity of the dataset | 6 |
| $\theta_{\text{max-off}}$ | Maximum offset for the angle | $\pi/6$ |

Table 11: Constants used for generating WINDMILL dataset, their meaning, and their values.

$$R_B \sim \mathcal{B}(1, 2.5) \qquad \text{(Sample radius)}$$

$$R = \frac{r_{\text{max}}}{2}\left(BR_B + (1-B)(2 - R_B)\right) \qquad \text{(Modify sampled radius based on } B)$$

$$\Theta_A \sim \mathcal{C}\left(\left\{2\pi\frac{i}{n_{\text{arms}}+1} : i = 0, \ldots, n_{\text{arms}} - 1\right\}\right) \qquad \text{(Choose an arm)}$$

$$\Theta_{\text{off}} = \theta_{\text{max-off}}\sin\left(\pi\lambda_{\text{off}}\frac{R}{r_{\text{max}}}\right) \qquad \text{(Calculate radial offset for the angle)}$$

$$U \sim \mathcal{U}(0, 1) \qquad \text{(To choose a random angle)}$$

$$\Theta = \theta_{\text{wid}}(U - 0.5) + A\left(\Theta_A + \frac{\pi}{n_{\text{arms}}}\right) + (1 - A)\Theta_A + \Theta_{\text{off}}$$

$$\text{(Angle is decided by } A \text{ and the radial offset)}$$

$$X_1 = R\cos\Theta, X_2 = R\sin\Theta, X = \begin{bmatrix} X_1 \\ X_2 \end{bmatrix} \qquad \text{(Convert to Cartesian coordinates)}$$

PyTorch code to generate WINDMILL dataset is provided in Listing 1.

Listing 1: Code for WINDMILL dataset

```python
import math
import torch

# Constants
num_arms = 4 # number of blades in the windmill
max_th_offset = 0.5236 # max offset that can be added to the angle for shearing (= pi/6)
r_max = 2 # length of the blade
num_p = 20000 # number of points to be generated
offset_wavelength = 6 # adjusts the complexity of the blade

# Sample latent variables according to the causal graph.
A = torch.bernoulli(torch.ones(num_points) * 0.6)
if observational_data:
    B = A
else:
    B = torch.bernoulli(torch.ones(num_points) * 0.5)

# Convert A, B to X.
th_A0 = torch.linspace(0, 2*math.pi, num_arms+1)[:-1]
th_A1 = torch.linspace(0, 2*math.pi, num_arms+1)[:-1] + math.pi/num_arms
# Choose a random arm for A=0 from possible arms. Likewise for A=1.
th_A0 = th_A0[torch.randint(num_arms, (num_p,))]
th_A1 = th_A1[torch.randint(num_arms, (num_p,))]

# beta distribution with alpha=1, beta=3
beta_dist = torch.distributions.beta.Beta(1, 2.5)

# Sample r according to B. If B=0, sample a small r, else sample a large r.
# r ranges from 0 to r_max
B0_r = beta_dist.sample(torch.Size([num_p])) * r_max/2.
B1_r = r_max - beta_dist.sample(torch.Size([num_p])) * r_max/2.
r = B * B0_r + (1-B) * B1_r

# Sample theta according to A.
# Choose the theta arm according to A and then sample from this arm using a uniform distribution.

# First we will have a cartwheel style.
theta = torch.rand(num_p)*th_wid + th_A0*(1-A) + th_A1*A - th_wid/2.

# Add an offset to theta according to r.
th_offset_mod = torch.sin((r/r_max)*offset_wavelength*math.pi)
th_offset = max_th_offset*th_offset_mod
theta += th_offset

x1 = r*torch.cos(theta)
x2 = r*torch.sin(theta)

data = torch.stack([x1, x2], dim=1)
labels = torch.stack([A, B], dim=1).type(torch.long)
```

