# OpenReview forum: "Incorporating Interventional Independence Improves Robustness against Interventional Distribution Shift"
_TMLR — Accepted by TMLR_

### Review · Reviewer_MhWe · 2025-08-29

**Summary Of Contributions:**

This paper deals with the problem of learning representations that are robust to distribution shift arising from causal interventions. Throughout the paper, it assumes that the underlying causal structure is known, i.e. it is known which variables are intervened upon and on which other variables they depend. The paper contributes new insights into the interplay between adherence to the underlying causal structure and the predictive performance on both observational and interventional data. That is, representations that violate the given causal structure by, for instance, not enforcing independence between interventional and observational variables correlate with representations that exhibit a large gap in performance on observational and interventional data. The authors use this initial empirical insight to then provide certain sufficient conditions for a linear causal model to provably adhere to the causal structure and later to propose a novel method called RepLIn that aims to improve causal adherence through the addition of two connected loss functions. RepLIn is evaluates on a synthetic and two more realistic datasets with an induced causal structure and compared to a range of state-of-the-art baselines.

**Additional Comments:**

#### Strengths
1. The paper is generally very well written. It is easy to follow from the very start as the authors make it very clear what the aim of their research is by using a clear example. The remainder of the paper is equally easy to read and follow. Some more technical parts do need to be read twice to fully understand, but I fully attribute that to being more technical.
2. It seems the general idea of the paper to assume that the causal structure is known and that you want to enforce that knowledge as much as possible is quite novel. Most other approaches do not assume so and need to jointly learn the causal structure with their representations. While the latter is more flexible, I believe it might also lead to learning causal structures that help the representations to solve their task instead of being truly correct.
3. The experimental section provides sufficient evidence that RepLIn does both increase predictive performance as well as lower the predictive gap between observational and interventional data. Apart from that, I also appreciate how the synthetic experiment is used to clarify the initial intuitions.

**Audience:**

Yes

**Audience Explanation:**

This paper proposes a method to improve adherence of learned representations to a given causal model. Given the ubiquity of machine learning and representation learning, enforcing causality, when known, is an interesting research direction.

**Broader Impact Concerns:**

No concerns.

**Claims And Evidence:**

Yes

**Claims Explanation:**

The paper provides ample empirical evidence that supports its claims of lowering the gap in predictive performance between observational and interventional data.

**Requested Changes:**

While I am generally positive about this submission, I do have a couple of questions and remarks.
1. Table 1: Why are these results not also provided with standard deviation/error like all other results later on? The results also seem to be different from those reported in Table 2 (ERM-resampled reaches a observational accuracy of around 96 while it reaches around 93 in Table 1). The difference, at least from the provided variability metrics, seems larger than what run-to-run variance would suggest possible. What could be the cause of this?
2. One bigger apparent weakness would seem to me that the representations you train do not have any *inherent* ability to distinguish observational cases from interventional ones. While the distributions for $X$ and $X'$ are different, the same model is used for predicting both variables, leaving the model to effectively try to model a mixture of these two distributions since the inputs for $X$ and $X'$ are, of course, still the same. Intuitively, I would expect that the model could be explicitly conditioned on an intervention taking place to facilitate full distribution shift invariance. I do understand that the information that the model is predicting interventional data is not always available, but I feel a discussion on this topic would make sense in the paper. If nothing else, for future work? This topic is important, because it also appears in your derivation of the linear model, where the non-degenerate solution S2 is effectively enforcing fully independent representations for $A$ and $B$ because the linear model is unaware it is being intervened upon, while it is expected to give reasonable performance still.
3. Top of Section 3.3: "We also randomly set the number of training epochs to use early-stopping", I do not fully understand what you mean here. Do you set the patience of early stopping with a random number of epochs?
4. Section 3.4: "Assuming zero mean for all latent variables", is this realistic? Do you have any empirical evidence that this at least *tends* to hold for linear models?
5. Proposition 1 does seem on the weaker side because of the conditions on $\beta$. The proof of the statement seems sound and I understand why these conditions arise, but from you empirical validation it does appear that the more practical setting of a small $\beta$ mostly does not satisfy those conditions. Hence, I feel this result is not saying much of practical importance, especially since the discussion does seem to favour solution S2 because of its more intuitive interpretation of "no signal from $A$ is used for $B$". A more extended discussion on the importance of this result could help place it better in the overall story of the paper.
6. Can you give an intuitive reason why the optimal regularisation parameters do not generalise between the two datasets? I can understand why in the non-resampled case, but for resampled data I would expect them to be similar.
7. Why would the dependence go down with increasing $\beta$ for resampled data (Figures 8 and 9)? Given you resample, I would expect both interventional and observational components to have a similar impact in the loss and so have a more uniform final dependence.
8. I do not think the qualitative analysis on CelebA contributes much of substance to the paper. Not only do methods like GradCam have their own problems, it is hard to grasp how representative the 5 handpicked examples are. In particular, what happens for images where ERM-resampled was correct, but RepLIn-resampled was not?

---

> ### Author Response · Authors · 2025-10-16
> **Added a new, more general analysis**
>
> Dear reviewer MhWe,
>
> We received the last of the three reviews recently, and we have further updated the PDF. The major change is **a new, more general analysis** in Sec. B.4 that is **valid for multiple latent variables, added noise, and nonlinear mixing functions**. We first show that the weights of a linear predictor trained with dependence loss can be **interpreted as adding a correction vector** to the weights of a vanilla predictor trained with only prediction loss. By visualizing this correction vector, we find that this correction vector indeed pushes the weights of the linear predictors towards those of a robust predictor, and decreases in magnitude as $\beta$ increases.
>
> We also examined the **effect of imperfect intervention**. We found that RepLIn’s performance worsens compared to a vanilla predictor trained on a training dataset with observational and perfect interventional data, especially for higher values of $\beta$. However, RepLIn still outperformed vanilla predictors trained on the same training dataset with imperfect interventions, particularly when $\beta$ was low. Although these results showed that RepLIn was robust against imperfect interventions at lower $\beta$ values, they also indicated that RepLIn performed best when perfect interventional samples were available. Therefore, we have added this limitation to RepLIn.

---

### Review · Reviewer_6NZU · 2025-09-01

**Summary Of Contributions:**

This paper proposes RepLIn, a method for improving robustness to interventional distribution shifts by enforcing statistical independence between representations of intervened nodes and their non-descendants. The motivation is from the observation that hard interventions induce independence in the causal graph. The method adds NHSIC-based penalties and self-dependence regularization, with theoretical support (in linear settings) and empirical results on synthetic and real-world datasets.

**Audience:**

Yes

**Audience Explanation:**

This paper is on some interesting and relevant areas — causality, representation learning, and robustness — so I think it would definitely catch the attention of people working in those areas. The idea of using interventions to encourage independence in learned representations is a nice one, and the method is lightweight and practical.

However, the links to prior work in domain generalization, OOD robustness, and causal disentanglement aren't clearly discussed. As a result, it's a bit hard to tell how novel the method really is or where it fits in. It would be better to include a clearer discussion and some stronger comparisons.

**Broader Impact Concerns:**

No negative social impacts

**Claims And Evidence:**

Yes

**Claims Explanation:**

The claims in the paper are supported with:
- It provides a clear theoretical explanation of why ERM fails under intervention and under what conditions enforcing independence improves robustness.
- Experimental results on both synthetic and real datasets support the effectiveness of the proposed approach.


However, several aspects weaken the overall clarity and novelty of the contribution:
- While standard domain generalization (DG) baselines (IRM, Fish, GroupDRO) are included, the conceptual connection between interventional robustness and domain generalization is not fully developed. The paper does not clearly explain how RepLIn relates in assumptions, objectives, or methodology from existing DG/OOD approaches.
- The discussion of identifiable representation learning is limited to the appendix, without elaborating on how RepLIn achieves robustness without full identifiability, or how it compares to related works such as CITRIS[1] and DMSVAE[2], which also utilize interventions in similar settings.
- The paper would benefit from discussing design choices—e.g., why the authors chose NHSIC-based independence constraints instead of full identifiability losses (like ELBO + MMD), and why this might be a lighter-weight, more scalable alternative.

Overall, while the core idea is valid and supported by evidence, the contribution feels incomplete without stronger comparative analysis and positioning.

[1] CITRIS: Causal Identifiability from Temporal Intervened Sequences, 2022

[2] Nonparametric Partial Disentanglement via Mechanism Sparsity: Sparse Actions, Interventions and Sparse Temporal Dependencies, 2024

**Requested Changes:**

Please check the review above.

---

> ### Author Response · Authors · 2025-10-16
> **Added a new, more general analysis**
>
> Dear reviewer 6NZU,
>
> We received the last of the three reviews recently, and we have further updated the PDF. The major change is **a new, more general analysis** in Sec. B.4 that is **valid for multiple latent variables, added noise, and nonlinear mixing functions**. We first show that the weights of a linear predictor trained with dependence loss can be **interpreted as adding a correction vector** to the weights of a vanilla predictor trained with only prediction loss. By visualizing this correction vector, we find that this correction vector indeed pushes the weights of the linear predictors towards those of a robust predictor, and decreases in magnitude as $\beta$ increases.
>
> We also examined the **effect of imperfect intervention**. We found that RepLIn’s performance worsens compared to a vanilla predictor trained on a training dataset with observational and perfect interventional data, especially for higher values of $\beta$. However, RepLIn still outperformed vanilla predictors trained on the same training dataset with imperfect interventions, particularly when $\beta$ was low. Although these results showed that RepLIn was robust against imperfect interventions at lower $\beta$ values, they also indicated that RepLIn performed best when perfect interventional samples were available. Therefore, we have added this limitation to RepLIn.

---

### Review · Reviewer_tJsN · 2025-10-02

**Summary Of Contributions:**

This paper addresses the challenge of learning robust representations that perform well under interventional distribution shifts. The authors identify a critical oversight in existing methods: they often fail to leverage independence arising from interventions, treating interventional data merely as samples from a different domain. The core contribution of the paper is to establish a strong correlation between this performance degradation on interventional data and the statistical dependence of learned features that ought to be independent. To address this, the authors propose a new training algorithm RepLIn, which explicitly regularizes the model to enforce these known statistical independencies in the feature space during training. The paper provides theoretical justification for this approach in a linear setting, deriving sufficient conditions under which enforcing independence provably reduces test-time error. The effectiveness of RepLIn is demonstrated empirically on a synthetic dataset as well as on image and text datasets, showing improved robustness compared to ERM and several domain generalization baselines.

**Audience:**

Yes

**Audience Explanation:**

This paper reveals that incorporating interventional independence improves robustness against interventional distribution shift, which may interest some researches in the community of trustworthy machine learning.

**Broader Impact Concerns:**

I have no concerns.

**Claims And Evidence:**

Yes

**Claims Explanation:**

This paper provides both a theoretical foundation (although limited in scope) and extensive experimental results to demonstrate its central claim: incorporating interventional independence improves robustness against interventional distribution shift.

**Requested Changes:**

There are several areas where the analysis could be expanded and clarified to further strengthen the contribution.

1. The theoretical analysis in Section 3.4, while providing valuable intuition, rests on a simplified causal model that may limit its generality.
- The causal mechanism is modeled as a deterministic assignment ($B = \omega_{AB}A$). In most literature, such relationships include an exogenous noise term (e.g., $B = \omega_{AB}A + \epsilon_B$) to account for unobserved factors. It would be beneficial to discuss or show whether the paper's conclusions still hold in this more common and realistic stochastic setting.
- The analysis considers "hard interventions" where a variable is set to a fixed value. I am curious if the analysis can be extended to the more general case of stochastic interventions, where the target variable is made to follow a new distribution independent of its causal parents. Broadening the theoretical scope to these settings would significantly strengthen the paper's claims.

2. The theoretical results, while insightful, could be developed further.
- The analysis is confined to the simplest case of a two-variable linear system. While I understand the difficulty of non-linear analysis, extending the theory to the multi-variable linear case would be a valuable addition. This would better align the theory with the scalability experiments presented in Section 6.2 and provide a more complete theoretical foundation for the method.
- Proposition 1 claims that if $\beta$ is larger than a specific threshold, the model will be more robust. But the threshold depends on the unobservable causal strength parameter $\omega_{AB}$. Moreover, It remains unclear what happens when $\beta$ is smaller than the threshold.

3. The paper proposes minimizing MHSIC to enforce independence. While the authors correctly note its relation to MMD, the paper would benefit from a more explicit justification for this choice over other common independence measures. For instance, a discussion on the pros and cons of NHSIC versus minimizing Mutual Information (MI)—a very common technique in representation learning—in terms of computational complexity, estimator variance, and suitability for gradient-based optimization would be highly valuable.


4. There are a few minor points that could improve the clarity and accuracy of the presentation.
- First, the paper refers to the experiments on CelebA and CivilComments as being on "real image and text datasets". This is slightly imprecise, as the causal graphs and the nature of the interventions are synthetically imposed onto the data. I would suggest a more accurate phrasing, such as "experiments on real-world datasets with semi-synthetic causal structures."
- Second, there are several places where citations are unnecessarily enclosed in parentheses.

---

> ### Author Response · Authors · 2025-10-16
> **Response to reviewer tjsN (part 1/2)**
>
> We thank the reviewer for their comments. We agree that a deeper analysis would make the paper more insightful. Therefore, we have added a new, more general analysis incorporating the factors mentioned by the reviewer in Sec. B.4.
>
> > A new, more general analysis in Sec. B.4
> >
>
> The new analysis in Sec. B.4 is **valid for multiple latent variables, added noise, and nonlinear mixing functions**. We first show that the weights of a linear predictor trained with dependence loss can be **interpreted as adding a correction vector** to the weights of a vanilla predictor trained with only prediction loss. By visualizing this correction vector, we find that this correction vector indeed pushes the weights of the linear predictors towards those of a robust predictor, and decreases in magnitude as $\beta$ increases.
>
> We also examined the **effect of imperfect intervention**. We found that RepLIn’s performance worsens compared to a vanilla predictor trained on a training dataset with observational and perfect interventional data, especially for higher values of $\beta$. However, RepLIn still outperformed vanilla predictors trained on the same training dataset with imperfect interventions, particularly when $\beta$ was low. Although these results showed that RepLIn was robust against imperfect interventions at lower $\beta$ values, they also indicated that RepLIn performed best when perfect interventional samples were available. Therefore, we have added this limitation to RepLIn.
>
> > Q1.1 Will the proof hold if we consider added noise terms in the causal graph
> >
>
> Yes, the proof holds when an additional noise term is added to $B$ during observations (assuming the noise does not overpower the causal relation from $A$). Since this noise term is independent of $A$ during observation, it is absorbed into $\tilde{U}$ in Eq. (11). While it will change the exact formulae that we derive, the core conclusion will not be affected. We have added the following sentence in Sec. 3.4 to clarify this point to the reader: “Eq. (4) would have taken a different form if there was an added noise term in the causal relation A → B, but would have still been non-zero.”
>
> > Q1.2 Imperfect/stochastic intervention
> >
>
> This is a very valuable question. As we mentioned earlier, in the new analysis added during this revision in Sec. B.4, we have included empirical results to investigate the effect of imperfect intervention on RepLIn. We modeled imperfect interventions by setting a probability of intervention failure, $\eta$, during the data-generation process. Here, with $\eta$ probability, we replace each interventional sample in the training dataset with an observational sample. During training, RepLIn minimizes dependence on this imperfect interventional distribution. On one hand, as we expected, RepLIn performs worse than a vanilla predictor (trained with $\eta=0$) when $\eta$ increases, particularly for larger values of $\beta$. On the other hand, RepLIn still outperformed the corresponding vanilla predictor trained with the same value of $\eta$, particularly at low values of $\beta$. While these results indicate that RepLIn is robust against imperfect interventions in those scenarios where interventional data is scarce, they also show that RepLIn works best when perfect interventional samples are available (added this to limitations). We believe these results further strengthen the practical importance of RepLIn.
>
> > Q2.1 Multi-variable causal model
> >
>
> In our approach, we enforce pairwise independence between the parent variables and the intervened variable. The primary metric is the robustness in predicting the parent variable during distribution shifts caused by the said intervention. Since our approach affects each parent variable independently, the analysis for a given parent-child pair can be extended to multiple parent-child pairs. This decision helped us to write a more concise proof in Sec. 3.4.
>
> We also agree with you that an analysis that is valid for multi-variable causal models strengthens our findings about the scalability of RepLIn. Therefore, in the revised PDF, we have added a new analysis in Sec. B.4 of the revised PDF where we simultaneously minimized dependence on multiple interventions to address your concerns.
>
> > Q2.2 When $\beta$ is lower than the threshold in Proposition 1
> >
>
> The conclusion from Proposition 1 provides the proportion of interventional data during training **sufficient** to improve robustness. Proposition 1 is not a guiding principle on whether RepLIn is suitable or not for the given value of $\beta$, since it does not imply that RepLIn cannot be applied if $\beta$ is less than this sufficient value. Proposition 1 is an affirmative theory that guarantees performance improvement under certain conditions. This can be seen in the empirical evidence where RepLIn is effective even for very small values of $\beta$.

---

> > ### Author Response · Authors · 2025-10-16
> > **Response to Reviewer tjsN (part 2/2)**
> >
> > > Q3. Discussion on mutual information
> > >
> >
> > Thank you for the suggestion. We agree that including a discussion comparing HSIC with MI will help the readers. Our choice of HSIC over MI was guided primarily by two factors: (1) **computational ease of HSIC**: Compared to HSIC, it is challenging to compute MI, especially in high dimensions, as it typically requires density estimation. Existing MI estimators also have high variance (Song and Ermon, 2020). (2) **flexibility of kernel-based measures**: Compared to MI, HSIC requires little to no assumption on the kernel. This makes HSIC an attractive component in flexible training algorithms for deep networks. We have added a discussion on MI in Sec. 3.3 with these points and references.
> >
> > > Q4.1 Adding "semi-synthetic causal structures” to description
> > >
> >
> > Thank you for this suggestion. We agree that adding “semi-synthetic causal structures” to the experiment description is more appropriate. We have revised the PDF accordingly in the abstract, introduction, experiment details, and conclusion.
> >
> > > Q4.2 Citations in parentheses
> > >
> >
> > As far as we know, the default citation format (using citet) for TMLR includes parentheses if you are referencing the paper. E.g., “This was shown previously in (XYZ, et al., 2000).” We did not use parentheses when we referenced the authors (using citep). E.g., “This was shown previously by XYZ, et al., 2000.”
> >
> >
> > J. Song and S. Ermon, “Understanding the Limitations of Variational Mutual Information Estimators”, ICLR 2020.

---

### Author Response · Authors · 2025-09-24
**Writing changes in the revised PDF**

Two reviewers have submitted their reviews so far. We thank the reviewers for their constructive feedback. Their comments and questions were productive and improved the quality of the submission. Here, we list and summarize the changes that we have made to the PDF to answer their questions (added text in the PDF is shown in green, and removed text is struck out in red):

1. **[6NZU] Differences w.r.t. DG/OOD setting**: We have added a discussion in Sec. 2 (Related works) and in detail in App. D about the differences between the setting of domain generalization/out-of-distribution and our setting. In summary, our setting has the broader objective of learning robust representations for all causal variables and has more constraints and assumptions that allow us to get more trustworthy representations.
2. **[MhWe] Hyperparameter change between experiments**: We extended the hyperparameter ablation in App. H to empirically explain the hyperparameter change. To summarize, the hyperparameters get larger when the functional relation from the parent latent variable to the observed signal becomes more difficult to learn.
3. **[MhWe] GradCAM visualization for negative results**: We now show the GradCAM visualizations for the randomly chosen samples that were incorrectly classified by RepLIn but were correctly classified by ERM in App. F. We found that despite the wrong prediction, the features still paid attention to the appropriate face regions, which suggests that the errors made by RepLIn are not due to the distribution shift.
4. **[6NZU] Positioning w.r.t. CITRIS and other ICRL works**: We had an extensive discussion about ICRL works in App. C that included CITRIS and an earlier version of Lachapelle et al. (2024) listed by 6NZU. We do not compare with ICRL methods as we cannot make the final predictions from the general attribute-identifiable representations learned by ICRL, as they are identifiable only up to permutation invariance. We have now added references to these works in Sec. 2 (Related Works), and explained why we do not compare with ICRL works in Secs. 2 and 5 (Experiments). We have also extended App. C to clarify why RepLIn does not require full identifiability, unlike ICRL works. In summary, since our objective is to learn robust discriminative representations, we can afford to forego some identifiability as long as it does not affect our discriminative performance.
5. **[6NZU] Why not MMD instead of NHSIC**? NHSIC was chosen to enforce independence due to demonstrated success in computing dependence between representations in prior work. NHSIC is essentially MMD between the joint and the product of marginals. This explanation is now included in Sec. 3.3, where we were already discussing HSIC and KCC.
6. **Minor writing changes**:
    1. [MhWe] Clarified what early stopping means in Sec. 3.3.
    2. [MhWe] Clarified what Proposition 1 entails for our work.
    3. [MhWe] Reported the same numbers in Tables 1, 2, and 7.
    4. [MhWe] Added details to the medical diagnosis example to improve clarity.

---

### Author Response · Authors · 2025-12-06
**Comments about the camera-ready version**

We thank the action editor (AE) and the reviewers for their constructive feedback. For the camera-ready version, we have incorporated their feedback during the discussion phases and AE’s comments in the decision. The changes made are listed below:

1. **Clarifications in Sec. B.4**: The vectors and the matrices used in the proof are now described. Intermediate computation steps are also detailed.
2. **DG/OOD vs RepLIn**: As recommended by the AE, we now use less polarizing language to compare our setting with DG/OOD. We have also included pointers in Sec. 2 to the detailed review of the related works in the appendix. We also moved the short discussion on group imbalanced training to the appendix section on DG/OOD, since learning from imbalanced groups is studied in DG/OOD literature (e.g., GroupDRO).
3. **Consistency in terminology and results**: The terminology is now consistent throughout the paper. We use the term “semi-synthetic causal model” to describe our real dataset settings in the abstract, introduction, experiments section, and conclusion. The numbers in the main tables are also consistent with those in the rebuttal.
4. **Details for practical utility**: We have included a detailed discussion on the kernel choice and bias considerations for empirical HSIC estimation in Secs. 3 and 4, along with a description of how kernel design affects the nature of dependence it captures. The discussion on MI v HSIC is in Sec. 3.
5. **Limitations and future works**: We have stated the limitations of RepLIn that we identified during the discussion phases in Sec. 6.
6. **Minor writing changes**: To improve the overall readability of the paper, we have:
    1. shortened some sentences, especially in the abstract and the introduction,
    2. added bold/underlined descriptions for highlighting key aspects in paragraphs,
    3. reordered some paragraphs for a smoother flow of ideas, and
    4. rewrote some sentences to remove hanging words at the end of paragraphs.

As a result of these changes, the main paper is now 15 pages long, compared to 16 pages at the end of the discussion phase.

---

### Decision · Action_Editor_ZXU7 · 2025-11-19

**Recommendation:** Accept with minor revision

**Additional Comments:**

The paper is generally well-written, empirically solid. Reviewers provide some useful suggestions, and it requires minior revision.

1. Clarify and complete Sec. B.4 derivations. Define all matrices/vectors used (e.g., those in Eq. (16)), provide the missing algebraic steps (the equality below Eq. (18) and the gradient-to-zero step in Eq. (19)), and ensure symbols are consistent with the main text. Since B.4 is already in the appendix, fuller detail is preferable.

2. Temper the positioning vs. DG/OOD. Please present RepLIn as a specialized interventional OOD setting rather than a fundamentally distinct paradigm. The current App. D language reads overly polarizing; tighten the comparison with standard DG/OOD assumptions/objectives, and keep the (helpful) summary table while avoiding overselling. A brief pointer to how RepLIn differs from/relates to ICRL works (e.g., CITRIS/MSR) in the main text would help orient readers.

3. Consistent terminology and reporting. Use “semi-synthetic causal structures” uniformly when describing CelebA/CivilComments. Ensure all tables (incl. Table 1) report mean ± std and that cross-table numbers are reconciled as per your response.

4. Independence penalty choice and practical guidance. Keep the brief HSIC vs. MI discussion in Sec. 3.3 and add 2–3 sentences of practical guidance (e.g., kernel choice, estimator stability, batch size implications). This will aid reproducibility and adoption.

5. Limitations & future work. In the Limitations section, explicitly (i) restate that RepLIn works best with accurate/perfect interventions and known causal graphs, (ii) note open questions around intervention-aware predictors (when a flag is available at test time), and (iii) summarize your observations on regularization transferability across datasets (why optimal β/λ differ) as current best-practice guidance.

**Audience:**

Yes

**Audience Explanation:**

This work sits at a timely intersection of causal representation learning and robustness. The idea of explicitly enforcing intervention-induced independences in learned features is simple, practical, and broadly relevant to researchers working on domain shift, causal structure-aware learning, and trustworthy ML.

**Claims And Evidence:**

Yes

**Claims Explanation:**

The paper substantiates its main claim that enforcing interventional independence improves robustness to interventional distribution shift through (i) a clear causal motivation, (ii) a linear-theory analysis giving sufficient conditions for improvement, and (iii) experiments on synthetic data and real-world datasets with semi-synthetic causal structures (CelebA, CivilComments).